# The Direction of Landscape Erosion

Colin P. Stark[1] and Gavin J. Stark[2]

[1]BlueMarbleSoft
[2]University of Cambridge

**Correspondence:** C. P. Stark (cstarkjp@gmail.com)

**Abstract.**

The rate of erosion of a landscape depends largely on local gradient and material fluxes. Since both quantities are functions of the shape of the catchment surface, this dependence constitutes a mathematical straitjacket, in the sense that—subject to simplifying assumptions about the erosion process, and absent variations in external forcing and erodibility—the rate of change of surface geometry is solely a function of surface geometry. Here we demonstrate how to use this geometric self-constraint to convert a gradient-dependent erosion model into its equivalent Hamiltonian, and explore the implications of having a Hamiltonian description of the erosion process. To achieve this conversion, we recognize that the rate of erosion defines the velocity of surface motion in its orthogonal direction, and we express this rate in its reciprocal form as the surface-normal slowness. By rewriting surface tilt in terms of normal slowness components, and by deploying a substitution developed in geometric mechanics, we extract what is known as the fundamental metric function of the model phase space; its square is the Hamiltonian. Such a Hamiltonian provides several new ways to solve for the evolution of an erosion surface: here we use it to derive Hamilton's ray tracing equations, which describe both the velocity of a surface point and the rate of change of the surface-normal slowness at that point. In this context, gradient-dependent erosion involves two distinct directions: (i) the surface-normal direction, which points subvertically downwards, and (ii) the erosion ray direction, which points upstream at a generally small angle to horizontal with a sign controlled by the scaling of erosion with slope. If the model erosion rate scales faster than linearly with gradient, the rays point obliquely upwards; but if erosion scales sublinearly with gradient, the rays point obliquely downwards. This dependence of erosional anisotropy on gradient scaling explains why, as previous studies have shown, model knickpoints behave in two distinct ways depending on the gradient exponent. Analysis of the Hamiltonian shows that the erosion rays carry boundary-condition information upstream, and that they are geodesics, meaning that surface evolution takes the path of least erosion time. Correspondingly, the time it takes for external changes to propagate into and change a landscape is set by the velocity of these rays. The Hamiltonian also reveals that gradient-dependent erosion surfaces have a critical tilt, given by a simple function of the gradient scaling exponent, at which ray propagation behaviour changes. Channel profiles generated from the non-dimensionalized Hamiltonian have a shape entirely determined by the scaling exponents and by a dimensionless erosion rate expressed as the surface tilt at the downstream boundary.

# 1 Introduction

When geomorphologists describe the evolution of a landform, a direction of erosion is often invoked: for example, we speak of a bank cutting laterally, or a cliff retreating, or a knickpoint eroding upstream, or a river channel incising down into bedrock. Generally, such statements are taken at face value, and the erosion direction in each case is understood from context, e.g., erosion in a bedrock channel is broadly considered to take place sub-vertically downwards, hewing closely to gravity, except at knickpoints where it occurs sub-horizontally upstream, and along the channel walls where it acts sub-horizontally and roughly orthogonal to streamflow. At the same time, we recognize that the geomorphic processes driving or mediating erosion are associated with particular directions relative to the geometry of the surface, which presumably has consequences for the direction in which that surface erodes: weathering acts roughly normal to an exposed surface, mechanical abrasion involves obliquely streamwise impacts that can be resolved into normal and tangential components, as can frictional wear by sliding ice or debris, and so on. There are obviously many directions involved in driving the evolution of a landscape, so what can we say about the direction of motion of the erosion surface itself? Our goal here is to answer this question using some concepts and tools of differential geometry and classical mechanics.

## 1.1 Tracking points on an erosion surface

Tracking the motion of a solid object is easy if the surface of the object is not eroded during motion: all that's needed is to tag the surface with markers and monitor their displacements. This isn't possible for a surface undergoing erosion, because all such markers are destroyed by the erosion process itself. We can nevertheless describe, in a mathematical sense, how points on an erosion surface move—if we know something about the process of erosion. The purpose of this section is to preview how this task can be achieved and to provide some conceptual context. The ideas outlined here are developed in full in the main body of the paper.

A moving erosion surface has only one *intrinsic* direction available at each surface point: the local normal to the surface. Describing motion in any other way entails the supply of extra information—through the choice of an additional direction as a reference. Since gravity acts downwards, the usual choice is to assign vertical as the reference axis, and to express erosion rate as a vertical velocity. On the other hand, for problems such as sea cliff retreat or river bank erosion it can be more convenient to pick horizontal as the reference. Whatever the choice, basic trigonometry makes it easy to transform an erosion function between any of these geometries (but with a complication: see Sect. 3.1).

The minimal approach therefore avoids supplying a reference direction and treats surface erosion as acting intrinsically in the local normal direction. In light of this, we may be tempted to infer that *points* on an erosion surface move in the normal direction: in general, however, they do not.

To see why, let's examine a surface evolving by some unknown mechanism. Let's assume for simplicity that the surface is an always-smooth 1D line in 2D $x$–$z$ space (Fig. 1). Mark the surface at time $T_a$, and again at a very small time interval later $T_b = T_a + \Delta T$. Each surface can be considered as a set of points: $T_a = \{a\}$ and $T_b = \{b\}$, where $a$ and $b$ are 2D vectors.

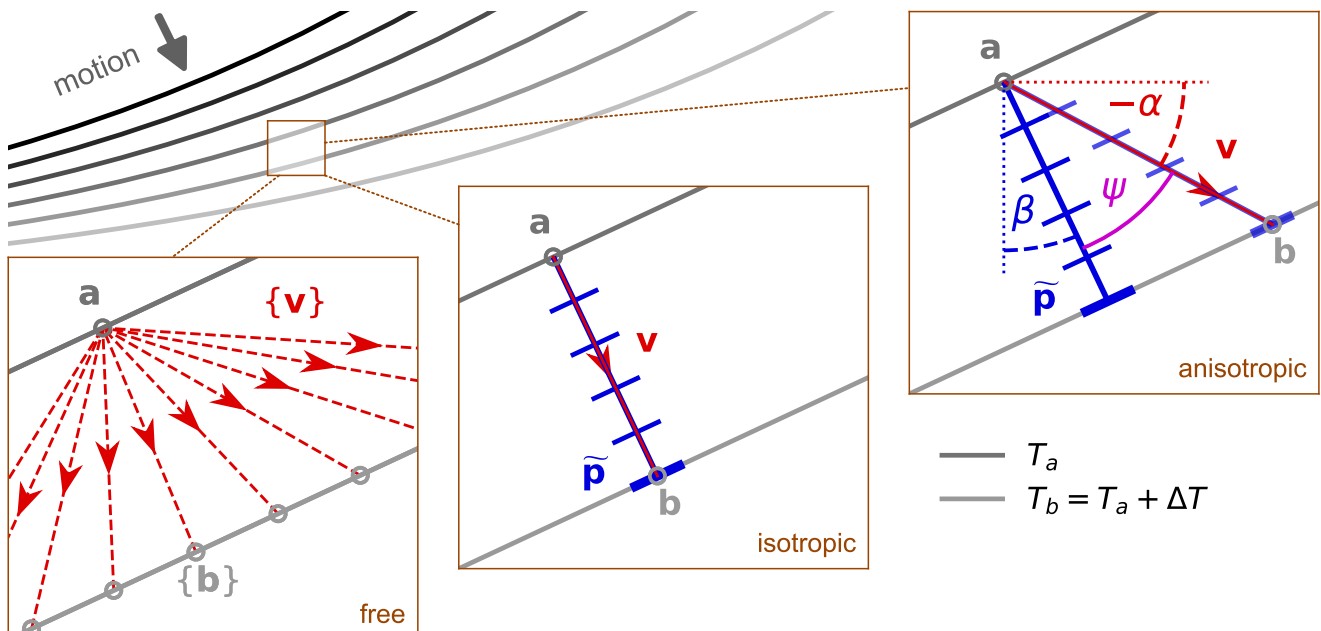

**Figure 1.** Illustration of how points map from one erosion surface (grey curves) at time $t = T_a$ to the next at time $t = T_b = T_a + \Delta T$. If the erosion mechanism is not specified ("free" inset), each point $a \in T_a$ (solid grey circle) can in principle be mapped to any of the points $\{b\} \in T_b$ (empty grey circles). However, if the erosion process is known, the point mapping is constrained (albeit indirectly) as follows. The erosion function can be converted into a metric function that tells us how far apart the surfaces are after the time interval $\Delta T$. We gauge this spacing using a slowness covector $\widetilde{p}$ (blue ladder symbols) oriented normal to $T_a$ and at an angle $\beta$ from vertical. If we convert the metric into a Hamiltonian, we get evolution equations both for $\widetilde{p}$ and for point velocity $v$ (red arrows; at angle $\alpha$ clockwise from horizontal). The point velocity determines the point pairing. If the erosion process is independent of gradient ("isotropic" inset), the metric is Euclidean, the point velocity is colinear with normal slowness, and the point $a$ on $T_a$ maps to its nearest neighbour $b$ on $T_b$. If instead the erosion process is gradient-dependent ("anisotropic" inset), the metric it generates is non-Euclidean, $\widetilde{p}$ and $v$ are not colinear, and the mapping of point $a$ to point $b$ is oriented at an angle $\psi = \alpha - \beta + 90°$ to the surface normal. The angle $\psi$ is therefore a measure of erosional anisotropy.

 In the absence of an equation of motion, we are free to pair each point $a \in T_a$ with any otherwise unpaired point $b \in T_b$
60 (Fig. 1, "free" inset). We could enforce a strict order to the pairings, but we would still have a very large number of choices. What matters is that the motion of surface points from $T_a$ to $T_b$ is defined by our choice of mappings, and for the moment this choice is arbitrary.

 Now, there are two ways to assess the rate of surface motion: one familiar, the other much less so. The familiar quantity is the velocity vector, which we get by measuring the distance between, and direction defined by, each pair of points $v :=$
65 $(b - a)/\Delta T$. The unfamiliar quantity is the normal-slowness covector $\widetilde{p}$ (Sect. 3.1), which get by measuring the time $\Delta T$ it takes for the surface to move a given distance in its normal (intrinsic) direction. We visualize $\widetilde{p}$ as a series of small planes

emanating from $\boldsymbol{a}$, parallel to the local tangent to $T_a$ and approaching $T_b$. The term "slowness" is used because the units of $\widetilde{\boldsymbol{p}}$ are reciprocal speed.

This brings us to our key premise: when we specify an erosion function, we are explicitly defining the behaviour of $\widetilde{\boldsymbol{p}}$, but
only indirectly obtain the behaviour of $\boldsymbol{v}$. That's because an erosion rate function measures the time it takes for the *surface* to move a given distance, not the travel time for *points* on the surface. If the process of erosion is isotropic, this subtle distinction is moot; if, however, the erosion rate depends on gradient, the distinction is fundamentally important (Fig. 1, "isotropic" and "anisotropic" insets).

We can understand why if we realize that by quantifying the elapsed time between successive erosion surfaces, the erosion
rate function actually defines a *metric*, aka a tool for measuring the "length" of the covector $\widetilde{\boldsymbol{p}}$. If the erosion rate depends only on location, meaning that it's independent of surface tilt and thus isotropic, the corresponding metric is Euclidean, which makes $\widetilde{\boldsymbol{p}}$ and $\boldsymbol{v}$ point in the same direction, and leads each point on $T_a$ to pair with its nearest neighbour (in a Euclidean sense of the term) on $T_b$. This is the most intuitive way of linking points on one surface to another, but is not correct for erosion in general. That's because, if the erosion rate is also a function of gradient, the resulting metric will be anisotropic and non-Euclidean,
$\widetilde{\boldsymbol{p}}$ and $\boldsymbol{v}$ will point in different directions, and the way the metric measures the shortest distance between successive erosion surfaces will no longer be a simple use of Pythagorean geometry.

Metrics of this kind—that depend on position and orientation—are called *Finsler metrics*. They constitute a way to measure travel time between two points when resistance to motion varies with direction in a non-trivial way. Physical analogues include measuring travel time when walking over hills or navigating a boat in a wind. In special cases they may reduce to, or at least
incorporate, a Riemannian form.

Transformation of the erosion equation into a metric function takes a few steps. The first is to reparameterize the directional parts of the erosion equation using the components of the slowness covector $\widetilde{\boldsymbol{p}} = [p_x, p_z]$ while leaving any spatial dependence untouched. For example, if the erosion function depends explicitly on surface gradient $\tan\beta$, where $\beta$ is the angle of the surface-normal relative to vertical, we can use the substitution $\tan\beta = |p_x/p_z|$. The normal erosion speed is replaced with the
reciprocal magnitude of the slowness covector $\xi^\perp = 1/p = 1/\sqrt{p_x^2 + p_z^2}$.

If this reparameterization is possible, we get an equation that can be rearranged into the form $\mathcal{F}_*(\boldsymbol{a}, \widetilde{\boldsymbol{p}}) = 1$. This $\mathcal{F}_*$ is a fundamental metric function, which measures the shortest time interval for the surface to erode a unit distance in a given direction. Among several special properties exhibited by this function, the crucial one is its order-1 Euler homogeneity in $\widetilde{\boldsymbol{p}}$, which means that $\mathcal{F}_*(\boldsymbol{a}, \lambda\widetilde{\boldsymbol{p}}) = \lambda\mathcal{F}_*(\boldsymbol{a}, \widetilde{\boldsymbol{p}})$.

Squaring and scaling the metric function defines a quadratic Hamiltonian $\mathcal{H}(\boldsymbol{a}, \widetilde{\boldsymbol{p}}) := \mathcal{F}_*^2/2 = 1/2$, which is the key result of this study. This "geomorphic" Hamiltonian provides us with equations of surface motion in the form of Hamilton's equations, which allow ray tracing and other methods to be used to solve for landscape evolution. And it tells us not just that surface points move according to a Hamiltonian flow, but also that they follow geodesic path aka paths of shortest erosion time.

Point velocities, and therefore point pairings $\{\boldsymbol{a}, \boldsymbol{b}\}$, are given by one half of Hamilton's equations: differentiating the
Hamiltonian by each of the erosion slowness covector components $p_x$ and $p_z$ in turn, we get a vector expressing the change of

point position with time: $\boldsymbol{v} = \partial\mathcal{H}/\partial\widetilde{\boldsymbol{p}}$. It follows from order-1 homogeneity of the metric function $\mathcal{F}_*$ that the surface-normal slowness covector and this point velocity vector must always be conjugate, $\widetilde{\boldsymbol{p}} \cdot \boldsymbol{v} = 1$.

Earlier, we asserted that surface points do not, in general, move in the surface-normal direction: now we have proof. Exploiting conjugacy, we can measure the angle $\psi$ between the surface-normal and the point velocity using their dot product $\cos\psi = \widetilde{\boldsymbol{p}} \cdot \boldsymbol{v}/(p\,v)$. If the rate of erosion depends on surface tilt $\beta$, the corresponding metric function and Hamiltonian will both depend, in some nonlinear fashion, on the normal slowness components $p_x$ and $p_z$, and so $\partial\mathcal{H}/\partial\widetilde{\boldsymbol{p}}$ and point velocity $\boldsymbol{v}$ will not in general be colinear with the surface normal. A gradient-dependent erosion process is therefore *anisotropic*, and its degree of anisotropy is measured by the angle $\psi$.

The practical consequence of erosion driving anisotropic Hamiltonian flow lies in how it controls the propagation of information, in the sense of initial and boundary conditions, into a landscape. Each element of this Hamiltonian flow has both a point position $\boldsymbol{a}$ and a normal slowness $\widetilde{\boldsymbol{p}}$, i.e., each element contains information about the location and orientation of the surface and its reciprocal rate of erosion. Progression along the Hamiltonian flow occurs along successive point pairings; each pairing translates an element in space while carrying (and to some extent modifying) the surface information along with it. The angular disparity between the direction of information transfer (aka point velocity) and the intrinsic direction of surface-normal motion, is the anisotropy $\psi$.

## 1.2 Structure of the paper

The paper is organized into eight sections and a set of appendices. Section 2 summarizes how erosion in three dimensions (3D) can be tracked using implicit surfaces and level sets, makes a connection with the Hamilton-Jacobi equation, and demonstrates the natural link with Hamiltonian methods. Section 3 combines these concepts with those introduced in Section 1.1 and formulates a Hamiltonian theory of gradient-driven erosion (for a 2D slice of 3D space). It explores this theory using geometric mechanics and differential geometry, and reveals how strong anisotropy lies at the heart of landscape surface evolution. Section 4 implements the geomorphic surface Hamiltonian using a particular model of gradient-driven erosion—an adaptation of the stream-power incision model to handling erosion in the surface-normal direction—and presents a non-dimensionalization of the Hamiltonian and Hamilton's ray tracing equations, and a simple means of model solution. Section 5 shows how to use Hamiltonian ray tracing to obtain model surface solutions for a domain whose boundaries are subject to a constant vertical erosion rate. Section 6 discusses these numerical solutions and examines what they have to tell us about erosional anisotropy and the notion of two distinct directions of landscape erosion. It also relates model scales to real-world landscape time, space and velocity scales. Section 7 looks at the broader implications of the Hamiltonian approach to erosion, and Sect. 8 draws some conclusions. The appendices A–F draw on disparate literature sources linked together here for the first time, and use them to shed light on the theory presented in this paper.

## 2 Core principles

### 2.1 Landscape as an implicit surface

In almost every model treatment of landscape erosion (Coulthard, 2001; Dietrich et al., 2003; Fowler, 2011; Pazzaglia, 2003; Tucker and Hancock, 2010; Tucker, 2015; van der Beek, 2013; Willgoose, 2005), the shape of the land surface in 3D space is written mathematically as a function of elevation $h$ parameterized by the 2D horizontal coordinates $\{x, y\}$ of points on the surface, and by the time $t$ at which the point elevations are assessed. In other words, the landscape is described by an explicit surface function $h(x, y; t)$.

An explicit surface description has advantages and disadvantages. On the plus side, theoretical development is relatively simple, because it effectively involves only two spatial dimensions, and numerical solution can be carried out on a 2D grid. On the negative side, the rate of erosion is *only* tracked in the vertical direction, through the partial derivative of elevation with time $\partial h / \partial t$: if there is any horizontal component of erosion it is not tracked directly, and has to be calculated indirectly using the lateral variation in elevation $\nabla h$. Problems arise when the surface gradient becomes very steep, for example at knickpoints or channel banks, and any development of overhangs is obviously impossible.

If we instead describe the landscape using an implicit surface, many of these issues are eliminated. The price is greater complexity in the mathematics needed to formulate surface motion and to resolve it numerically. The extra cost is worth paying if it leads to greater insights into how landscapes form.

### 2.2 Landscape as the 2D zero contour of a 3D function

An implicit surface in 3D space is the set of points $\{x(t), y(t), z(t)\}$ that define the 2D "contour" or level-set surface of a function $\phi$:

$$\phi(x, y, z; \ldots) = \phi_0 \tag{1}$$

where $\phi$ is a nonlinear function defined at all points across the 3D domain of interest, that varies with time, and is non-local—in the sense that it can be a function of curvature, or of values of itself at a distance, etc. Put more simply, $\phi$ is a very flexible function that can be tailored to induce whatever surface motion is desired.

The term "implicit" is used because surface positions are not specified directly; instead, a surface is defined by "slicing" the function $\phi(x, y, z)$ at some chosen value $\phi_0$ and finding positions $\{x, y, z\}$ for which $\phi(x, y, z) = \phi_0$. Think of how a visualization tool for a 3D scalar field, such as temperature, works: sequential slicing across a range of temperatures provides an animated view of its variation throughout a volume. This variation can be complex, revealing isolated blobs of high (or low) values that connect in topologically complicated ways as the slicing threshold temperature is changed. In this way, an implicit description of a surface can represent complex, multivalued geometry and topology without extra mathematical work.

Landscape evolution can be modelled with an implicit surface by writing an equation to drive evolution of the function $\phi$, and watching how its zero level-set $\phi = \phi_0 = 0$ implicitly prescribes changes in surface positions $\{x(t), y(t), z(t)\}$ over time.

## 2.3 The level-set equation

Implicit surface motion in its most general form is described by the level-set equation (Gibou et al., 2018; Giga, 2006; Osher and Fedkiw, 2001, 2003; Sethian, 1999; Vladimirsky, 2001), in which $\phi(x, y, z; t)$ is a 3D function constructed so as to evolve over time $t$ with a velocity $\boldsymbol{\xi}$, a vector function that in general varies with position and time, is possibly non-local, and only need be defined where $\phi = 0$:

$$\frac{\partial \phi}{\partial t} + \boldsymbol{\xi} \cdot \boldsymbol{\nabla} \phi = 0 \tag{2}$$

This is equivalent to holding the material derivative of the scalar field $\phi$—driven to move by the vector field $\boldsymbol{\xi}$—at zero along the zero contour of $\phi$, but otherwise allowing it to vary unconstrained.

Only the normal component $\xi^\perp$ of the implicit surface velocity plays any role in driving motion: in the geomorphic case, this would be the surface-normal erosion rate. So we can write

$$\frac{\partial \phi}{\partial t} + \xi^\perp |\boldsymbol{\nabla} \phi| = 0 \tag{3}$$

The notation $\xi^\perp$ is adapted from Osher and Merriman (1997).

This equation provides a very generic description of how a 2D surface evolves in 3D space, in the sense that it defers all description of processes into the formulation of the surface-change rate function $\xi^\perp$. This function can readily treat topographic gradient and curvature, and substrate erodibility; suitably provided with coupled process equations, it could also incorporate water flow depth and velocity, intermittent sediment cover, development of a vegetation layer, spatiotemporal precipitation, tectonic displacement, and so on. Such flexibility, however, is not our goal here. Instead, we seek geometric insights into the process of landscape erosion, which we can achieve if we limit the scope of this equation, and make $\xi^\perp$ a simplified function of local gradient and accumulated flow. A geomorphic level-set equation in this form makes it easier to tease out its fundamental behaviour.

## 2.4 Motion described by the Hamilton-Jacobi equation

If we restrict the surface velocity $\boldsymbol{\xi}$ to be a *local* function of position and time, Eq. (3) becomes the Hamilton-Jacobi equation, or HJE:

$$\mathcal{H}(\boldsymbol{r}, \boldsymbol{\nabla} \phi; t) = \frac{\partial \phi}{\partial t} \tag{4}$$

where each vector $\boldsymbol{r}$ tracks a point as it moves from one zero level-set of $\phi$ to another with velocity $\dot{\boldsymbol{r}} = \mathrm{d}\boldsymbol{r}/\mathrm{d}t$, while the front itself at that point moves in the direction $\boldsymbol{\nabla}\phi$. These directions are not necessarily the same.

The HJE is a first-order partial differential equation that plays a central role in classical mechanics (Arnold, 1989; Goldstein et al., 2000; Houchmandzadeh, 2020; Small and Lam, 2011; Whitham, 1999). Its driver is the Hamiltonian $\mathcal{H}$, which combines the surface velocity $\boldsymbol{\xi}$ with the gradient $\boldsymbol{\nabla}\phi$ in a way that lends it special properties.

The Hamiltonian in the HJE is required to be a local function—in the sense that it can depend on position $\boldsymbol{r}$ and instantaneous time $t$, but cannot depend on the shape of the propagating surface at some distance away, or on any history-dependent quantities.

Diffusive and quasi-diffusive processes are not allowed either. However, viscosity solutions of the HJE (Crandall and Lions, 1981), which are the standard means of resolving profound mathematical challenges with this equation, ironically involve the addition of a weak, ultimately vanishing, second-order term that can be considered a diffusive process at the sub-grid scale.

## 2.5 Landscape as an erosion arrival-time surface

If we wish to use the HJE to treat landscape evolution in terms of an implicit function, we need to consider how to write a Hamiltonian form of the erosion function driving that evolution. If this Hamiltonian is independent of (i.e., does not change with) time, it simplifies into a static HJE or eikonal equation $\mathcal{H}(\boldsymbol{r}, \boldsymbol{\nabla}\phi)$. The implicit surface function $\phi$ that solves this static HJE is a single-valued, 3D function that defines the position and shape of arrival time surfaces. In other words, $\phi$ can be thought of as a first arrival time function $T(x, y, z) - t$, where $T$ defines the locus of surface points $\{x, y, z\}$ that satisfy at each time step $t$ the equation

$$T(x, y, z) - t = 0 \tag{5}$$

Another way to express this is to say that the contours of $T$ are 2D isochrone surfaces embedded in 3D space that define the shape of the landscape as it changes.

In the eikonal equation, the Hamiltonian is a constant function of surface point position $\boldsymbol{r}$ and the gradient of the arrival time $\boldsymbol{\nabla}T$ with the simple form:

$$\mathcal{H}(\boldsymbol{r}, \boldsymbol{\nabla}T) = \text{const} \tag{6}$$

Points on the surface move with velocity vector $\boldsymbol{v} = \dot{\boldsymbol{r}}$, while the surface itself moves with a *slowness covector* given by $\widetilde{\boldsymbol{p}} = \boldsymbol{\nabla}T)$. It is important to emphasize that $\widetilde{\boldsymbol{p}}$ is not a vector: Sect. 3.1 goes into more detail as to what is meant by the term "covector" and why the distinction is consequential.

Although both $\boldsymbol{v}$ and $\widetilde{\boldsymbol{p}}$ are both directional quantities describing surface motion, they only point in the same direction if the motion mechanism is isotropic. Measuring their angular disparity is the key to assessing the anisotropy. One of the aims of this study is use this measure to reveal the strong anisotropy of landscape erosion processes (see Sect. 3.18).

## 3 Theory

In this section, we formalize the ideas presented above into a Hamiltonian theory of erosion front motion. First, we provide a gentle introduction to the pivotal concept of a covector (Sect. 3.1), and show how useful it is for treating the direction and reciprocal speed of the propagating front. Then we show that the gradient of the surface arrival time is itself a covector (Sect. 3.2). Next, we make the case that the geomorphic processes driving erosional motion of a topographic surface can be represented by local functions (Sect. 3.3) parameterized by the surface-normal covector, and how they constitute, broadly speaking, a form of geometric self-constraint (Sect. 3.4). After imposing a gradient-dependent form on the erosion function (Sect. 3.5), we show how the above ingredients lead, via the fundamental metric function, to a Hamiltonian description of

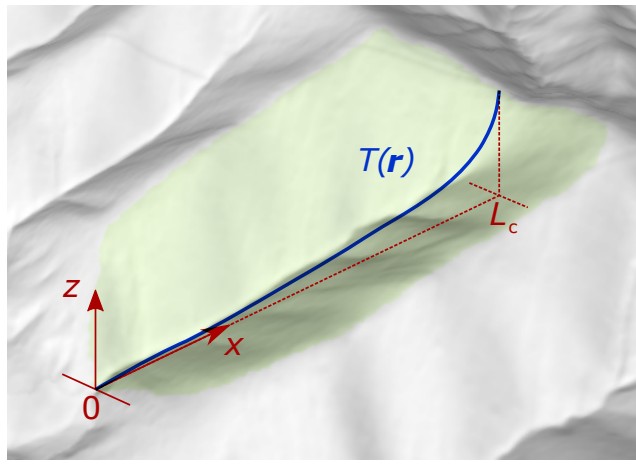

**Figure 2.** Model context and geometry. Theoretical treatment in the current study is limited to 2D. The model domain is a vertical transect following a stream profile, with vertical axis $z$ and horizontal axis $x$, spanning a fixed distance from catchment exit at $x = 0$ to drainage divide at $x = L_c$. The locus of points $\boldsymbol{r}$ along the profile at time $t = T$, aka the surface isochrone, is defined as $T(\boldsymbol{r})$.

erosion (Sects. 3.6–3.9). Next we delve into the connections between the fundamental function and erosional wavelets, and use them to provide a graphic explanation of Huygens' principle as applied to erosion surface propagation (Sect. 3.10). We then express the equivalent Fermat's principle in terms of the variational path of least action (Sect. 3.11) to show that a point on the surface follows the path of least erosion time. This leads on to derivation of Hamilton's ray tracing equations (Sects. 3.12–3.13) and a discussion of some of their properties (Sects. 3.14–3.15). A verification that the Lagrangian is constant (Sect. 3.16) follows. Then we discuss ray angles, their behaviour relative to surface tilt, and the existence of a critical tilt at which ray propagation behaviour changes (Sect. 3.17). This leads to an exploration of how the disparity between the two directions of erosion is a measure of erosional anisotropy (Sect. 3.18). Finally (Sect. 3.19), we look at the various ways the evolving surface tilt can be tracked in the model. Non-dimensionalization is undertaken in Section 4.

Note: we use superscripts for contravariant tensor components (e.g., $r^x$), and subscripts for covariant tensor components (e.g., $p_z$); the Einstein summation convention (summing over similar tensor components) is adopted for brevity. Symbol usage is summarized in Table A1.

## 3.1 Tracking erosion with covectors

Imagine a locally planar surface undergoing constant erosion (Fig. 3), where the surface tilt angle is $\beta$ and the vector $\boldsymbol{r}$ takes values that lie along the erosion surface at a given time $T(\boldsymbol{r})$. As time passes, erosion moves the surface progressively further into the substrate. Taking snapshots at regular intervals $\Delta T$ generates a uniformly spaced sequence of surfaces which we call erosional isochrones. These isochrones are level sets or contours of the arrival time function $T$. In Fig. 3, the time interval is chosen to be $\Delta T = 1\,\mathrm{y}$ and isochrones have been plotted for $T(\boldsymbol{r}) = \{0, 1, 2, 3, 4, 5\}\,\mathrm{y}$.

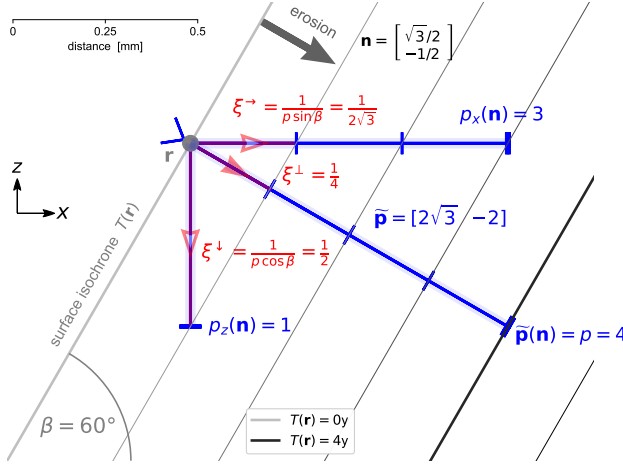

**Figure 3.** Tracking surface motion at a point $r$ using a slowness covector $\widetilde{p}$ normal to the erosion front $T(r)$, which points in the direction $n$. Normal slowness here is $p = \widetilde{p}(n) = 4\,\mathrm{y\,mm^{-1}}$ (Eqs. 13, 18) for a surface tilted at $\beta = 60°$ corresponding to a surface-normal erosion rate of $\xi^\perp = 1/4\,\mathrm{mm\,y^{-1}}$ (Eq. 8). Simple trigonometry applied to $p$ gives the vertical and horizontal slownesses (Eq. 17), and their reciprocals are the vertical $\xi^\downarrow$ and horizontal $\xi^\rightarrow$ erosion rates (Eqs. 9, 10, and 17). The front covector is also the gradient of the arrival times, or isochrone density, given by $\widetilde{p} = \nabla T$, which counts the number of isochrones crossed in unit time in the front-normal direction (Eq. 22).

Let's fix the point of interest $r$ at the location shown in Fig. 3. Here the surface-normal rate or speed of erosion is $\xi^\perp = 0.25\,\mathrm{mm\,y^{-1}}$ and surface tilt is $\beta = 60°$. Written as a vector, the erosion rate is:

$$\boldsymbol{\xi} := \begin{bmatrix} \xi^x \\ \xi^z \end{bmatrix} = \xi^\perp \begin{bmatrix} \sin\beta \\ -\cos\beta \end{bmatrix} = \begin{bmatrix} \sqrt{3}/8 \\ -1/8 \end{bmatrix} \tag{7}$$

with a direction normal to the surface and an angle $\beta = 60°$ to the vertical; its length or magnitude is the surface-normal erosion rate:

$$\xi^\perp = |\boldsymbol{\xi}| = \frac{1}{4}\,\mathrm{mm\,y^{-1}} \tag{8}$$

Ideally, we should only have to compute the sine and cosine components to the erosion velocity vector $\boldsymbol{\xi}$ to get the horizontal and vertical rates of erosion. However, the vertical trigonometric component $\xi^z$ does not equal the (negated) vertical rate of erosion $\xi^\downarrow$ (eq. 10), nor does the horizontal trigonometric component $\xi^x$ equal the horizontal rate of erosion $\xi^\rightarrow$:

$$\xi^x = \xi^\perp \sin\beta \quad \neq \quad \xi^\rightarrow = \frac{\xi^\perp}{\sin\beta} \tag{9}$$

$$-\xi^z = \xi^\perp \cos\beta \quad \neq \quad \xi^\downarrow = \frac{\xi^\perp}{\cos\beta} \tag{10}$$

It seems almost too trivial to ask, but why does naive application of trigonometry let us down here? The answer lies in the fact that we have written the erosion rate as a vector: we should instead express it as a *covector*.

Consider $\widetilde{p}$ in Fig. 3, which can be written as a function with single-row matrix form

$$\widetilde{p}(\cdot) = \begin{bmatrix} p_x & p_z \end{bmatrix}(\cdot) = \begin{bmatrix} 2\sqrt{3} & -2 \end{bmatrix}(\cdot) \tag{11}$$

This scalar function takes as input a vector such as $n$ and returns the number of isochrones crossed by that vector. Here $n$ is the surface-normal unit vector

$$n = \begin{bmatrix} n^x \\ n^z \end{bmatrix} = \begin{bmatrix} \sqrt{3}/2 \\ -1/2 \end{bmatrix} \tag{12}$$

Because we employ here units of millimetres and years, $n$ has a length of $|n| = 1\,\text{mm}$. Over this distance $n$ crosses four 1-year isochrones, so we obtain

$$\widetilde{p}(n) = \widetilde{p}\left( \begin{bmatrix} \sqrt{3}/2 \\ -1/2 \end{bmatrix} \right) = \begin{bmatrix} 2\sqrt{3} & -2 \end{bmatrix} \begin{bmatrix} \sqrt{3}/2 \\ -1/2 \end{bmatrix} = 4 \text{ y mm}^{-1} \tag{13}$$

Now consider the vertical component of $\widetilde{p}$ (which is negative here) acting on $n$: counting downwards over a distance $n_z = 1/2\,\text{mm}$, we find one isochrone crossing, so:

$$p_z(n) = \widetilde{p}\left( \begin{bmatrix} 0 \\ -1/2 \end{bmatrix} \right) = \begin{bmatrix} 2\sqrt{3} & -2 \end{bmatrix} \begin{bmatrix} 0 \\ -1/2 \end{bmatrix} = 1 \text{ y mm}^{-1} \tag{14}$$

The horizontal component of $\widetilde{p}$ counts three isochrone crossings by the unit normal vector counting rightwards over a distance $n_x = \sqrt{3}/2\,\text{mm}$:

$$p_x(n) = \widetilde{p}\left( \begin{bmatrix} \sqrt{3}/2 \\ 0 \end{bmatrix} \right) = \begin{bmatrix} 2\sqrt{3} & -2 \end{bmatrix} \begin{bmatrix} \sqrt{3}/2 \\ 0 \end{bmatrix} = 3 \text{ y mm}^{-1} \tag{15}$$

These components can be added together because $\widetilde{p}$ is a linear function; this summation gives

$$p_x(n) + p_z(n) = \widetilde{p}(n^x) + \widetilde{p}(n^z) = 3 + 1 = 4 = \widetilde{p}(n) \tag{16}$$

which is the count of four we found by measuring along $n$ directly. The count can of course take any real (fractional) value: for clarity, the example here has been constructed so as to yield round numbers.

The function $\widetilde{p}$ is called a *one-form* in the terminology of differential geometry, and instances of $\widetilde{p}$ are called covectors. In general, a one-form operates on a vector and returns a scalar. Here, $\widetilde{p}$ takes in a unit vector and returns the slowness of erosion in the direction of that vector. In optics and seismology, $\widetilde{p}$ is known as the normal slowness; in classical mechanics it is called the generalized momentum. In a geomorphic context, this normal slowness can be interpreted as the maximum isochrone density, and $\widetilde{p}$ the isochrone density covector, in that when applied to the unit normal vector $n$ it calculates the maximum number of isochrones to be found in any direction from that point.

The slowness covector $\widetilde{\boldsymbol{p}}$ is a more convenient measure of erosion rate because its sine and cosine components are the horizontal and vertical slownesses, which are (respectively) the reciprocal rates of erosion horizontally and vertically:

$$\widetilde{\boldsymbol{p}} = \begin{bmatrix} p_x & p_z \end{bmatrix} \tag{17}$$

$$= p \begin{bmatrix} \sin\beta & -\cos\beta \end{bmatrix}$$

$$= \begin{bmatrix} 1/\xi^{\rightarrow} & -1/\xi^{\downarrow} \end{bmatrix}$$

$$= \begin{bmatrix} \sin\beta/\xi^{\perp} & -\cos\beta/\xi^{\perp} \end{bmatrix}$$

The magnitude of the covector here is the normal erosion slowness aka the reciprocal erosion rate, and is given by

$$p = |\widetilde{\boldsymbol{p}}| = \sqrt{p_x^2 + p_z^2} = \frac{1}{\xi^{\perp}} \tag{18}$$

and surface slope is

$$\tan\beta = -\frac{p_x}{p_z} \tag{19}$$

In other words, by describing the rate of surface motion with an erosion slowness covector, instead of an erosion velocity vector, we can assess its variation with direction much more easily. Fundamentally, a covector is the correct way to represent motion *of a surface at a given point*, and a vector is the appropriate way to represent the position and motion *of that point*. See Appendix B for more details.

### 3.2   Gradient is a covector

The erosion slowness covector $\widetilde{\boldsymbol{p}}$ has another facet: it is also the gradient of the arrival-time function $T$. To see why, consider again Fig. 3 and its level sets of $T$ at discrete intervals. These level sets are isochrones or contour surfaces of equal arrival time $T(\boldsymbol{r}) = \{0, 1, \dots\}$, which are represented schematically as simple straight lines in this figure. They successively increase in the direction of the normal vector $\boldsymbol{n}$.

If we measure (in Fig. 3) the change in $T$ in the $x$ direction over a distance $n^x = \sqrt{3}/2$, we find that $n^x \mathrm{d}T_x = 3$. Similarly, if we measure the change in the $-z$ direction over a distance $n^z = 1/2$, we get $n^z \mathrm{d}T_z = 1$. In general terms,

$$\mathrm{d}T(\boldsymbol{n}) = n^x \mathrm{d}T_x + n^z \mathrm{d}T_z = \begin{bmatrix} p_x & p_z \end{bmatrix} \begin{bmatrix} n^x \\ n^z \end{bmatrix} = \widetilde{\boldsymbol{p}}(\boldsymbol{n}) \tag{20}$$

and in this example we find

$$\mathrm{d}T(\boldsymbol{n}) = 4\,\mathrm{y}\,\mathrm{mm}^{-1} \tag{21}$$

which is the normal slowness obtained in Eq. (13) written as a differential one-form. In other words, the rate of change $\mathrm{d}T(\cdot)$ over a unit distance in the isochrone-normal direction $\boldsymbol{n}$ is given by $\mathrm{d}T(\boldsymbol{n})$, and the isochrone or contour density $\mathrm{d}T(\boldsymbol{n})$ in the contour-normal direction is the same as the covector magnitude $p$. We can now invoke the gradient operator $\boldsymbol{\nabla}$ and have

$$\boldsymbol{\nabla}T := \begin{bmatrix} \dfrac{\partial T}{\partial x} & \dfrac{\partial T}{\partial z} \end{bmatrix} = \begin{bmatrix} p_x & p_z \end{bmatrix} = \widetilde{\boldsymbol{p}} \tag{22}$$

which says that the Euclidean gradient of the arrival time $T$ of the erosion surface is the normal slowness covector $\widetilde{\boldsymbol{p}}$.

## 3.3 Modelling erosion in the surface-normal direction

If we wish to frame a model of landscape evolution in terms of geometric mechanics, we need to employ the following three elements: (i) an implicit function to track the evolving landscape surface geometry; (ii) a surface-normal erosion slowness covector, corresponding to the gradient of the implicit function, that encodes the reciprocal rate of motion of the surface; and (iii) an erosion model for the surface-normal speed of erosion that can be parameterized using the slowness covector.

To supply the third element, we can write a generic model for the surface-normal speed of geomorphic erosion that is a solely function of local fluxes and gradient:

$$\text{surface-normal erosion rate} \sim \text{func}(\textit{flow}, \textit{gradient}) \tag{23}$$

Some erosion phenomena, such as quasi-diffusive processes like rain splash, cannot be modelled under this local restriction, but this is a minor loss. Henceforth the only flow we will consider is kinematic water flow resulting from spatially uniform rainfall-runoff, and we will ignore complexities such as storm hydrograph cycles and the effects of sediment supply, transport, and cover.

A model in this form is not unambiguously local: its dependence on accumulated water flow presupposes a dependence on upstream catchment geometry; any change in catchment geometry, through motion of drainage divides, acts to change flow at distant points downstream. A fundamentally important assumption here is that divide motion is slow enough for the erosion equation to be considered effectively local. The validity of this assumption is discussed at the end of Sect. 7.1.

## 3.4 Erosion imposes a geometric self-constraint

The process of landscape evolution represented by Eq. 23 is a kind of geometric straitjacket, or geometric self-constraint—in the sense that it essentially says the landscape obeys:

$$\text{changes in geometry} \sim \text{geometry} \tag{24}$$

In other words, the shape of the landscape determines the patterns of surface flow and thereby the fluxes of material over the surface, and it mediates the effectiveness of these fluxes through its control of the gradients; these effects combine to set the rate at which the shape of the landscape changes: in short, change in landscape geometry is controlled by landscape geometry. This conclusion applies even if the erosion process is not spatially local.

The consequence of this geometric self-constraint is that, at its heart, geomorphic erosion is driven by a particular kind of Hamiltonian. This Hamiltonian arises from how points on an erosion surface "see" (for want of a better term) their shortest path of erosion to the next set of surface points at little time later. The sections below explore this assertion in detail.

## 3.5 Separable, gradient-dependent erosion rate model

The Hamiltonian approach developed here can in principle be applied to any erosion rate model, with the proviso that the bedrock surface can only undergo erosion, meaning that its motion must always be positive $\xi^\perp > 0$. If transient sediment deposition and bed cover are to be modelled, meaning that topographic elevation (in the bedrock reference frame) can rise as well as fall, alluvial geometry needs to be tracked as an additional model variable along with bedrock surface position. The resulting Hamiltonian would not be static and the dimensionality of its phase space would be comparatively large. Such sophistication will eventually be needed, as models of this kind become the standard (e.g., Dietrich et al., 2003; Sklar and Dietrich, 2006; Zhang et al., 2015). However, in this introduction of geometric mechanics to the task of modelling erosion, we choose to avoid such complexity, and instead settle on an erosion equation that: (1) is a nonlinear (power) function of (space-time variable) rock surface gradient $\tan\beta(\boldsymbol{x},\boldsymbol{t})$; (2) has a separable form, with spatial variables (constant in time) such as flow velocity and depth, sediment concentration, substrate erodibility, and the abrasion process itself aggregated into a separate multiplicative term $\varphi(x)$; (3) describes the speed of erosion $\xi^\perp(\boldsymbol{x},t)$ in the rock-surface normal direction:

$$\xi^\perp(\boldsymbol{x},t) \; := \; \varphi(\boldsymbol{x})\,|\sin\beta(\boldsymbol{x},t)|^\eta \tag{25}$$

Note that surface tilt relative to vertical is expressed as $\sin\beta$ rather than $\tan\beta$, because erosion rate is measured in the normal rather than the vertical direction. In a further simplification, we restrict the model to a 2D transect (Fig. 2).

## 3.6 The erosion equation in Hamiltonian coordinates

Covectors are an essential ingredient in the construction of a Hamiltonian framework for surface erosion. As we will show in the coming sections, the Hamiltonian endows each point on the surface at position $\boldsymbol{r}$ with an associated tangent covector $\widetilde{\boldsymbol{p}}$ that represents the normal slowness of the surface at that point. The components of $\boldsymbol{r}$ and $\widetilde{\boldsymbol{p}}$ correspond to the axes of the phase space inhabited by the Hamiltonian.

Since our model is restricted here to a 2D transect of 3D Euclidean space, this Hamiltonian phase space is 4D; two of its four axes are spanned by the two components of the position vector, and the remaining two by the slowness covector components:

$$\boldsymbol{r} \; := \; \begin{bmatrix} r^x \\ r^z \end{bmatrix} \quad , \quad \widetilde{\boldsymbol{p}} \; := \; \begin{bmatrix} p_x & p_z \end{bmatrix} \tag{26}$$

The Hamiltonian parameters $(\boldsymbol{r},\widetilde{\boldsymbol{p}})$ are coordinates in what, in mechanics, is usually called momentum phase space, and in differential geometry is called a cotangent bundle; we henceforth refer to this as the *slowness* phase space since momentum has no meaning in the current context. It has a dual, called the velocity space, or tangent bundle, where the Lagrangian corresponding to the geomorphic surface Hamiltonian is defined.

Reiterating Eq. (18), and reducing it to express the surface tilt angle $\beta$ explicitly, we have:

$$\frac{1}{\xi^\perp} = p = |\widetilde{\boldsymbol{p}}| = \sqrt{p_x^2 + p_z^2} \quad , \quad \sin\beta = \frac{p_x}{\sqrt{p_x^2 + p_z^2}} \tag{27}$$

noting that $p_x > 0$ and $p_z < 0$ for the half-domain shown in Fig. 2. Each point in phase space acts entirely independently.

The erosion equation (Eq. 25) is now easy to convert into a form parameterized by the components of $\boldsymbol{r}$ and $\widetilde{\boldsymbol{p}}$:

$$\sqrt{p_x^2 + p_z^2} = \frac{1}{\varphi(r^x)} \left| \frac{\sqrt{p_x^2 + p_z^2}}{p_x} \right|^\eta \tag{28}$$

This equation defines the surface-normal reciprocal rate of erosion along a 2D profile, written in a form that neatly expresses the geometric self-constraint inherent to the geomorphic erosion process. This self-constraint is parameterized by vector position $(r^x, r^z)$ and covector normal-slowness $(p_x, p_z)$, which respectively locate a particular point on the surface and encode the reciprocal speed of erosion orthogonal to the surface at that point.

## 3.7 The fundamental function

What we need to do now is reparameterize Eq. (28) to express the degree to which a coordinate $(\boldsymbol{r}, \widetilde{\boldsymbol{p}})$ satisfies the geometric self-constraint imposed by this equation. This is easily achieved using Okubo's technique (Antonelli et al., 1993; Bao et al., 2000; Shimada and Sabau, 2005; Yajima and Nagahama, 2009; Yajima et al., 2011), in which the covector parameter is scaled by a positive function $\mathcal{F}_*(\boldsymbol{r}, \widetilde{\boldsymbol{p}})$:

$$p_x, p_z \quad \rightarrow \quad \frac{p_x}{\mathcal{F}_*}, \frac{p_z}{\mathcal{F}_*} \tag{29}$$

and substituted back in, rearranging to make $\mathcal{F}_*$ the subject:

$$\mathcal{F}_*(\boldsymbol{r}, \widetilde{\boldsymbol{p}}) = \varphi(r^x)\, p_x^\eta \left( p_x^2 + p_z^2 \right)^{(1-\eta)/2} \tag{30}$$

The function $\mathcal{F}_*$ is known as the *fundamental (metric) function* (see Appendix C; note that an asterisk in used in $\mathcal{F}_*$ for reasons that will become clear in Section 3.9). It is also a Hamiltonian, and as such it is associated with a phase space defined by the four coordinate components $(r^x, r^z, p_x, p_z)$. The subset of this 4D space whose locations satisfy the erosion equation given by Eq. (28) must meet the condition:

$$\mathcal{F}_*(\boldsymbol{r}, \widetilde{\boldsymbol{p}}) = 1 \tag{31}$$

The power of a Hamiltonian comes from being able to trace a sequence of $(\boldsymbol{r}, \widetilde{\boldsymbol{p}})$ across phase space for which this criterion holds continuously—a procedure otherwise known as solving Hamilton's equations—which yields the evolution over time of a single point on an erosion surface. However, for technical reasons (Sect. 3.8) it is best not to use $\mathcal{F}_*$ directly as the geomorphic surface Hamiltonian; a little more work is needed.

To clarify the behaviour of $\mathcal{F}_*$, consider the combined meaning of Eqs. (30) and (31). The value of $\mathcal{F}_*$ at a location in phase space with coordinates $(\boldsymbol{r}, \widetilde{\boldsymbol{p}})$ is equal to the normal slowness $\sqrt{p_x^2 + p_z^2}$ implied by that coordinate, aka its reciprocal erosion rate, multiplied by the erosion rate determined by the erosion process $\varphi(r^x)\,p_x{}^\eta / (p_x^2 + p_z^2)^{\eta/2}$ acting at that coordinate. This product—of speed times slowness—is obviously equal to one for locations in phase space that represent geomorphically valid surface points in real space. All other locations of phase space are unphysical, because at these values of $(\boldsymbol{r}, \widetilde{\boldsymbol{p}})$ the erosion rate is not reciprocal to the erosion slowness, and this product is not equal to one.

## 3.8 The geomorphic surface Hamiltonian

The problem with using $\mathcal{F}_*$ as a Hamiltonian is its order-1 Euler homogeneity: functions of this type generate a metric tensor whose determinant is singular, meaning that the tensor cannot be inverted (e.g., Červený, 2002). This puts the Legendre transform, and the Lagrangian, out of reach. Fortunately there is a simple solution: just use the fundamental function in its squared form, and define the geomorphic surface Hamiltonian as:

$$\mathcal{H}(\boldsymbol{r}, \widetilde{\boldsymbol{p}}) \ := \ \frac{1}{2}\mathcal{F}_*^2 = \frac{1}{2}\varphi^2(r^x)\, p_x^{2\eta}\left(p_x^2 + p_z^2\right)^{1-\eta} \tag{32}$$

A prefactor of $\frac{1}{2}$ is included to make subsequent derivations tidier.

This quadratic-form Hamiltonian has the advantage that it is order-2 Euler homogeneous:

$$\mathcal{H}(\boldsymbol{r}, \lambda\widetilde{\boldsymbol{p}}) = \lambda^2 \mathcal{H}(\boldsymbol{r}, \widetilde{\boldsymbol{p}}) \qquad \text{for } \lambda > 0 \tag{33}$$

which makes its metric tensor non-singular (if $\eta \neq 1$) and the Legendre transform feasible.

We know from Eq. (31) that $\mathcal{F}_* = 1$ for trajectories across slowness phase space that correspond to physically viable behavior of surface points. So we can assert that the Hamiltonian is static and has the value

$$\mathcal{H}(\boldsymbol{r}, \widetilde{\boldsymbol{p}}) = \frac{1}{2} \tag{34}$$

for solutions of the erosion equation. In more concrete terms, we can say that an arbitrary surface point located at $\boldsymbol{r}$ can only represent a point on an eroding surface if its associated orientation and slowness $\widetilde{\boldsymbol{p}}$ satisfies this equation.

## 3.9 The geomorphic surface Lagrangian

The quadratic Hamiltonian $\mathcal{H}(\boldsymbol{r}, \widetilde{\boldsymbol{p}})$ has a dual quantity called the Lagrangian $\mathcal{L}(\boldsymbol{r}, \boldsymbol{v})$, which operates in a counterpart space spanned by coordinates giving the position $\boldsymbol{r}$ and velocity $\boldsymbol{v}$ of evolving points on the erosion surface. By symmetry, the Lagrangian is also the quadratic of a fundamental function, denoted $\mathcal{F}$. This function $\mathcal{F}$ is the dual of $\mathcal{F}_*$, and is similarly order-1 homogeneous. Its quadratic $\mathcal{L}$ is similarly order-2 homogeneous:

$$\mathcal{L} \ := \ \frac{1}{2}\mathcal{F}^2 \tag{35}$$

To make the link between the spaces of $\mathcal{H}$ and $\mathcal{L}$, we recognize that the normal slowness covector can be defined as the derivative of the Lagrangian with respect to the velocity coordinate

$$\widetilde{\boldsymbol{p}} = \frac{\partial \mathcal{L}}{\partial \boldsymbol{v}} \quad \Leftrightarrow \quad p_i = \frac{\partial L}{\partial v^i} = \frac{\partial\left(\frac{1}{2}\mathcal{F}^2\right)}{\partial v^i} \tag{36}$$

This is known as the "fibre derivative".

Mapping from the Hamiltonian $\mathcal{H}$ to the Lagrangian $\mathcal{L}$ (and vice versa) exploits this property and is achieved with the Legendre transform:

$$\mathcal{L} = \widetilde{\boldsymbol{p}}(\boldsymbol{v}) - \mathcal{H} = p_i v^i - \mathcal{H} \tag{37}$$

A closed form for $\mathcal{L}$ requires several more pieces of the puzzle before it can be derived, and the eventual equation is unwieldy. The contrasting simplicity of $\mathcal{H}$ (Eq. 32) is why we prioritize the Hamiltonian over the Lagrangian in this paper.

In due course we will show that the dual fundamental function and the corresponding Lagrangian have constant values $\mathcal{F} = 1$ and $\mathcal{L} = \frac{1}{2}$, in symmetry with $\mathcal{F}_* = 1$ and $\mathcal{H} = \frac{1}{2}$. Such constancy means that the Lagrangian does not vary with time, and that the mutual variation of position $r$ and erosion velocity $v$ is tightly constrained.

### 3.10 Erosional wavelets and Huygens' Principle

Geometric optics provides a way to visualize the Lagrangian and its relationship to the Hamiltonian (Figures 4 and 5). Motion of an erosion front obeys Huygens' principle: we can imagine each point on the front generating a tiny erosional wavelet, and the coalescence of these wavelets forming the next erosion front. The shape of each erosional wavelet is defined by $\mathcal{F}$. Each shape is a velocity indicatrix giving the radial variation of ray velocity $v$ at a point $r$, or equivalently giving the distance that a point on the surface will erode in an infinitesimal interval.

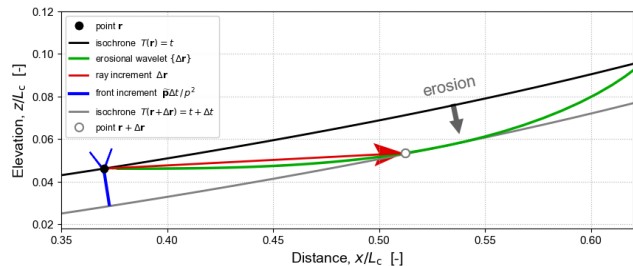

**Figure 4.** Incremental erosion (for $\eta = \frac{3}{2}$) described by $\mathcal{H}$ and $\mathcal{L}$, with $\mathcal{L}$ visualized as an erosional wavelet (green curve) aka a velocity indicatrix; point-motion ray vector in red and front-normal-motion covector in blue.

Figure 4 visualizes a single erosional wavelet, its relationship both to the current erosion front at $T = t$ and to the next at $T = t + \Delta t$, the particular ray increment vector for which $\mathcal{H} = \mathcal{L} = \frac{1}{2}$, and the conjugate relationship of this vector to the front normal covector (see Sect. 3.15). Motion of the surface $T(r) = t$ at point $r$ over the interval $\Delta t$ can be viewed in two mutually consistent ways: (i) the front moves a distance $\Delta t/p$ in the surface-normal direction given by $\widetilde{p}$; (ii) the point moves a distance $\Delta r = v \Delta t$ in the ray direction $r$. These directions are quite different, because the erosion process is strongly anisotropic (Sect. 3.18).

Unconstrained, the point at $r$ could be displaced onto any of the points along the erosional wavelet $\{\Delta r\}$. However, the only valid motion is onto the point $r + \Delta r$ where the tangent to the wavelet curve is orthogonal to the front increment $\widetilde{p}\Delta t/p^2$, i.e., the ray and front increments are conjugate to each other (Sect. 3.15).

When erosional wavelets at points along the surface are aggregated, moving $T(r)$ onto $T(r + \Delta r)$ as shown in Fig. 5, the result is anisotropic front motion that obeys Huygens' principle. The new front can also be found simply by propagating the old front a distance $\Delta t/p$ in the direction $\widetilde{p}$ at each point $r$.

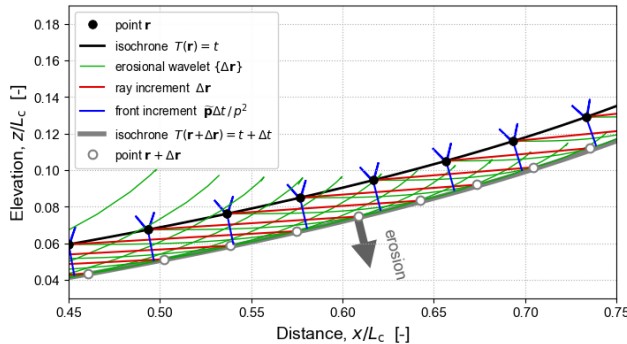

**Figure 5.** Huygens' principle visualized as the coalescence of erosional wavelets (green curves; for $\eta = \frac{3}{2}$) at their mutual tangent envelope (pale grey isochrone).

### 3.11 Fermat's principle as a least action integral

Huygens' principle emphasizes HJE solution in terms of propagation of a front; Fermat's principle, on the other hand, emphasizes solution in terms of tracing the trajectories of points along that front. These two principles are equivalent or dual (Holm, 2011; Houchmandzadeh, 2020; Small and Lam, 2011). Fermat's principle says that these trajectories are paths of stationary travel time: each trajectory obeys a variational principle which ensures its travel time is extremized; this extremal is almost always a minimum. The geomorphic equivalent is the principle that the path of erosion through a substrate from one point to

another is the shortest route given the erodibility of the material and its anisotropy and inhomogeneity.

This principle is expressed mathematically by writing an action functional $S_\gamma$, in terms of the static Lagrangian $\mathcal{L}$, for the set of all possible paths $\{\gamma(t)\}$ that a point on the erosion surface might take between two fixed points $\boldsymbol{a} = \boldsymbol{\gamma}(t_a)$ and $\boldsymbol{b} = \boldsymbol{\gamma}(t_b)$:

$$S_\gamma \; := \; \int_a^b \mathcal{L}\left(\boldsymbol{\gamma}(t), \dot{\boldsymbol{\gamma}}(t)\right) \, \mathrm{d}t \tag{38}$$

Note that the integrand $\mathcal{L}$ is independent of time $t$ and is a parametric function of positions along $\gamma$ only. The path actually

taken $\gamma_0$ is the path for which the variation of the action is stationary:

$$\boldsymbol{\gamma}_0 = \boldsymbol{\gamma}: \quad \delta S_\gamma = \delta \int_a^b \mathcal{L}\left(\boldsymbol{\gamma}(t), \dot{\boldsymbol{\gamma}}(t)\right) \, \mathrm{d}t = 0 \tag{39}$$

For paths traced across the velocity space to which the geomorphic surface Lagrangian $\mathcal{L}$ belongs, we can be sure that the action is minimized. Since $\mathcal{L}$ is independent of $t$, we can deduce that $\gamma_0$ is the path of (locally) least erosion time. *Such paths are known as geodesics.*

In summary: by expressing a local erosion equation as a geomorphic surface Hamiltonian, converting it into its dual Lagrangian form, and writing the consequent variational principle as the minimization of an action functional for paths across velocity space, we can conclude that points on an erosion surface follow the shortest (in terms of erosion time) possible paths

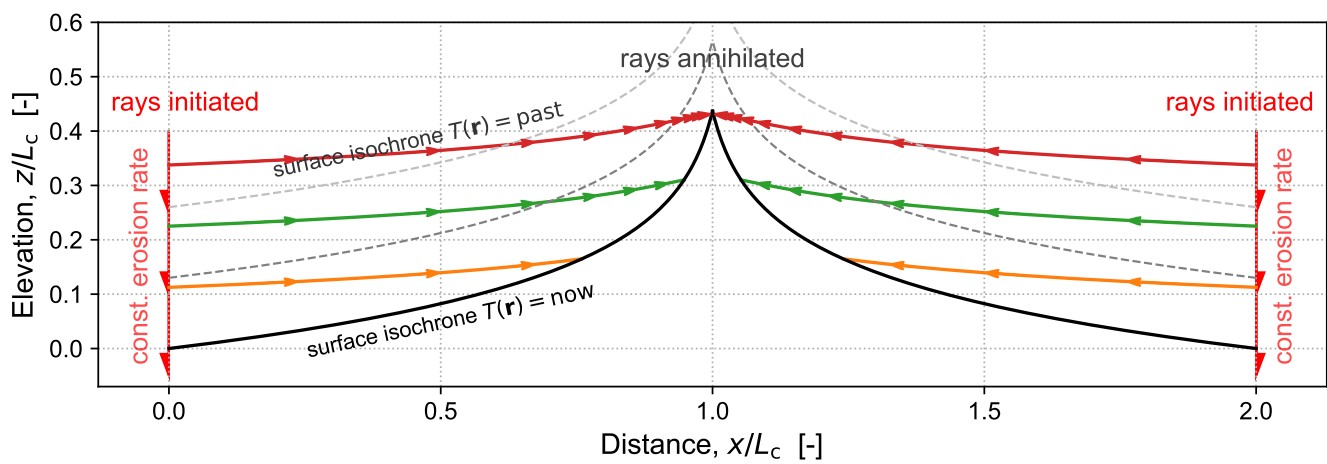

**Figure 6.** Ray tracing of erosion using Hamilton's equations (Sects. 3.12 & 4.2), illustrated here for a 2D landscape transect. The geomorphic surface Hamiltonian is solved over the left-hand half-domain, ranging from an exit boundary at $x = 0$ up to a drainage divide at $x = L_c$ (see Fig. 2). A fixed divide is enforced by mirroring this profile over the right-hand half-domain (for $L_c \leq x \leq 2L_c$, such that symmetrically generated rays annihilate each other at a cusp formed at $x = L_c$. The boundary condition imposed at $x = 0$ (and mirrored at $x = 2L_c$) is a constant vertical erosion rate $\xi^{\downarrow 0}$, mimicking the behavior of a vertical normal fault slipping at a constant rate $\xi^{\downarrow 0}$ at the boundary. The initial value of the front slowness covector $\widetilde{p}$ at $x = 0$ is chosen such that the surface tilt $\beta_0$ and vertical slowness $p_{z_0}$ are consistent with this rate. The model therefore simulates a horst block undergoing constant uplift and consequent erosion. Model topography is obtained by constructing surface isochrones $\{T(\boldsymbol{r})\}$ from the rays. Since rays are traced only from the boundary, and none from an initial surface, the isochrones are time-invariant. The standard term for such topography is "steady state", but the term is somewhat misleading here because the Hamiltonian dynamical system has no stable point.

in real space. The next section derives Hamilton's ray tracing equations from the Hamiltonian: integration of these rays across slowness phase space generates identical paths of shortest erosion time in real space.

### 3.12 Derivation of Hamilton's ray tracing equations

The fundamental function $\mathcal{F}_*$ generates a slowness phase space spanned by $\boldsymbol{r}$ and $\widetilde{\boldsymbol{p}}$ on which the geomorphic surface Hamiltonian $\mathcal{H}(\boldsymbol{r}, \widetilde{\boldsymbol{p}})$ operates, and we have a simple expression for $\mathcal{H}$ given by Eq. (32). We inferred the existence of a dual fundamental function $\mathcal{F}$ that generates a velocity space spanned by $\boldsymbol{r}$ and $\boldsymbol{v}$ on which a Lagrangian $\mathcal{L}(\boldsymbol{r}, \boldsymbol{v})$ operates, but we have yet to obtain expressions for $\mathcal{F}$ and $\mathcal{L}$. We can nevertheless make use of the Lagrangian to derive equations of motion for the erosion surface that operate on the slowness phase space. These are called Hamilton's equations.

Our starting point is to examine the differentials of $\mathcal{H}$ and $\mathcal{L}$ and to compare them. The geomorphic surface Hamiltonian defined in Eq. (32) is static, meaning that it is constant over time, so its differential is:

$$\mathrm{d}\mathcal{H} = \frac{\partial \mathcal{H}}{\partial r^i} \, \mathrm{d}r^i + \frac{\partial \mathcal{H}}{\partial p_i} \, \mathrm{d}p_i \qquad (40)$$

The differential of its counterpart Lagrangian $\mathcal{L}(\boldsymbol{r}, \boldsymbol{v})$ is:

$$\mathrm{d}\mathcal{L} = \frac{\partial \mathcal{L}}{\partial r^i}\,\mathrm{d}r^i + \frac{\partial \mathcal{L}}{\partial v^i}\,\mathrm{d}v^i \tag{41}$$

Substituting the "fibre derivative" form of $\widetilde{\boldsymbol{p}}$ in Eq. (36) into this equation, and adapting the terms in $p_i$, gives

$$\mathrm{d}\mathcal{L} = \frac{\partial \mathcal{L}}{\partial r^i}\,\mathrm{d}r^i + p_i\mathrm{d}v^i = \frac{\partial \mathcal{L}}{\partial r^i}\,\mathrm{d}r^i + \mathrm{d}(p_i v^i) - v^i\mathrm{d}p_i \tag{42}$$

Rearranging, we have an equation that contains the Legendre transform given in Eq. (37),

$$\mathrm{d}\left(p_i v^i - \mathcal{L}\right) = -\frac{\partial \mathcal{L}}{\partial r^i}\,\mathrm{d}r^i + v^i\mathrm{d}p_i \tag{43}$$

Consequently we have a second expression for the differential of $\mathcal{H}$:

$$\mathrm{d}\mathcal{H} = -\frac{\partial \mathcal{L}}{\partial r^i}\,\mathrm{d}r^i + v^i\mathrm{d}p_i \tag{44}$$

Equating the terms in $\mathrm{d}\mathcal{H}$ defined by this equation with those in Eq. (40), we obtain:

$$\frac{\partial \mathcal{H}}{\partial r^i} = -\frac{\partial \mathcal{L}}{\partial r^i} \quad , \quad \frac{\partial \mathcal{H}}{\partial p_i} = v^i \tag{45}$$

The next step is subtle but important. Every coordinate $(\boldsymbol{r}, \boldsymbol{v})$ in velocity space is (potentially) an initial position and velocity for a point on some initial erosion surface. Similarly, every coordinate $(\boldsymbol{r}, \widetilde{\boldsymbol{p}})$ in slowness phase space is (potentially) an initial position, surface orientation, and reciprocal surface-normal erosion rate for a point on that initial erosion surface. However, most such phase space coordinates do not correspond to real-world points lying on physically viable paths $\{\gamma_0\}$ that obey the principle of least erosion time established in Eq. (39). Conversely, for the locations in phase space that *do* lie on a paths of least action, we can write:

$$\frac{\mathrm{d}r^i}{\mathrm{d}t} = v^i \quad \Rightarrow \quad \int \delta v^i\,\mathrm{d}t = \delta r^i \tag{46}$$

Returning to the variation integral in Eq. (39), we can integrate by parts and simplify using the above result to get:

$$\delta S_\gamma = \int_a^b \left(\frac{\partial \mathcal{L}}{\partial r^i}\delta r^i + \frac{\partial \mathcal{L}}{\partial v^i}\delta v^i\right)\mathrm{d}t \tag{47}$$

$$= \left[\frac{\partial \mathcal{L}}{\partial v^i}\delta r^i\right]_a^b + \int_a^b \left(\frac{\partial \mathcal{L}}{\partial r^i} - \frac{\mathrm{d}}{\mathrm{d}t}\frac{\partial \mathcal{L}}{\partial v^i}\right)\delta r^i\,\mathrm{d}t = 0$$

The term in brackets $[\cdot]$ vanishes because $\boldsymbol{a}$ and $\boldsymbol{b}$ are fixed points, associated with limit times $t_a$ and $t_b$, at which $\delta r^i = 0$. The remaining integral gives the Euler-Lagrange equations for erosional surface motion:

$$\frac{\mathrm{d}}{\mathrm{d}t}\frac{\partial \mathcal{L}}{\partial v^i} - \frac{\partial \mathcal{L}}{\partial r^i} = 0 \tag{48}$$

Substituting this equation, Eq. (36) and Eq. (46) into the two linking equations in Eq. (45) we obtain Hamilton's equations:

$$\frac{\mathrm{d}r^i}{\mathrm{d}t} = \frac{\partial \mathcal{H}}{\partial p_i} \quad , \quad \frac{\mathrm{d}p_i}{\mathrm{d}t} = -\frac{\partial \mathcal{H}}{\partial r^i} \tag{49}$$

## 3.13 The meaning of Hamilton's equations

Hamilton's equations are coupled first-order ordinary differential equations (ODEs) whose integration gives the motion of a single point on an erosion surface in terms of a trajectory across slowness phase space. Each point along the trajectory has phase space coordinates of position $r^i = \boldsymbol{r}$ (also the position in real space) and normal slowness covector $p_i = \widetilde{\boldsymbol{p}}$ (which encodes both the local tilt of the erosion surface and its reciprocal rate of erosion $p = 1/\xi^\perp$). If we aggregate the trajectories of a set of points from an initial surface we have the motion of the whole surface. This method of front tracking is called ray tracing.

The differential equations in Eq. (49) define the rates of change of the coordinates $(\boldsymbol{r}, \widetilde{\boldsymbol{p}})$ in terms of the gradient components of the Hamiltonian. Since the Hamiltonian is a constant $\mathcal{H} = \frac{1}{2}$ along a ray or trajectory (Eq. 34), motion across the phase space must follow coordinates $(\boldsymbol{r}, \widetilde{\boldsymbol{p}})$ that trace a contour of $\mathcal{H}$. This is achieved by moving $\boldsymbol{r}$ in the direction $\partial\mathcal{H}/\partial p_i$ and $\widetilde{\boldsymbol{p}}$ in the direction $-\partial\mathcal{H}/\partial r^i$, which is to say, orthogonal to the Hamiltonian gradient.

Hamilton's equations take concrete form if we substitute the expression for $\mathcal{H}$ in Eq. (32) into Eq. (49). Since the model is limited here to 2D we have four coupled ODEs: two for the component rates of change of position,

$$\frac{\mathrm{d}\boldsymbol{r}}{\mathrm{d}t} = \dot{\boldsymbol{r}} = \begin{bmatrix} \dot{r}^x \\ \dot{r}^z \end{bmatrix} = \begin{bmatrix} v^x \\ v^z \end{bmatrix} = \frac{\partial\mathcal{H}}{\partial\widetilde{\boldsymbol{p}}} \tag{50}$$

$$= \varphi^2(r^x) \frac{p_x^{2\eta-1}}{(p_x^2 + p_z^2)^\eta} \begin{bmatrix} (p_x^2 + \eta p_z^2) \\ -(\eta-1)\,p_x p_z \end{bmatrix}$$

and two for the component rates of change of normal slowness,

$$\frac{\mathrm{d}\widetilde{\boldsymbol{p}}}{\mathrm{d}t} = \dot{\widetilde{\boldsymbol{p}}} = \begin{bmatrix} \dot{p}_x & \dot{p}_z \end{bmatrix} = -\frac{\partial\mathcal{H}}{\partial\boldsymbol{r}} \tag{51}$$

$$= -p_x^{2\eta} \left(p_x^2 + p_z^2\right)^{1-\eta} \varphi(r^x) \frac{\partial\varphi}{\partial r^x} \begin{bmatrix} 1 & 0 \end{bmatrix}$$

Ray tracing solutions of Hamilton's equations are illustrated in Figs. 6, 12, 13, 14 and 17.

## 3.14 Constancy of the vertical erosion rate along a ray

The erosion model defined in Eq. (25) is independent of elevation. This makes the Hamiltonian $\mathcal{H}$ independent of the vertical coordinate $r^z$, which leads to the zero element in $\dot{\widetilde{\boldsymbol{p}}}$ in Eq. (51), i.e., the vertical component of erosion slowness is constant:

$$\dot{p}_z = \frac{\mathrm{d}p_z}{\mathrm{d}t} = 0 \tag{52}$$

This is a manifestation of Noether's theorem (Holm, 2011; Noether, 1971), which states that a continuous symmetry in the action implies a conservation law for the Euler-Lagrange equations. Here, we have symmetry with respect to $r^z$ in $\mathcal{H}$, and therefore in $\mathcal{L}$, which implies a *law of conservation of vertical slowness* for the ray tracing equations, i.e., that $p_z$ must be conserved along a ray. Inasmuch as normal slowness can be crudely equated with the concept of momentum in classical mechanics, we have a "law of conservation of vertical momentum". Similar conservation laws limited to particular coordinate directions arise in geometric optics (Holm, 2011).

This property simplifies the task of ray tracing by reducing the number of coupled ODEs in the numerical integration from four to three. Moreover, this constancy has the profound implication that the initial rate of vertical erosion $\xi^{\downarrow 0}$ of a point is carried unchanged along its ray trajectory as the surface to which it is attached moves:

$$\xi^{\downarrow}(t) = -\frac{1}{p_z(t)} = -\frac{1}{p_{z_0}} = \xi^{\downarrow 0} \tag{53}$$

As such, each ray propagates information about the initial surface erosion rate upstream into the landscape until such time as it is destroyed at a cusp (which includes drainage divides: e.g., Fig. 6). Meanwhile the *horizontal* erosion rate can and does change along the ray, because the horizontal component of the slowness covector $p_x$ evolves as the surface erodes (Eq. 51).

### 3.15 Conjugacy of point velocity and front slowness

Hidden in the mathematics in previous sections is a simple relationship between the tangent velocity vector and cotangent normal-slowness covector pair: they are conjugate to each other (Figs. 4 and 5), which is to say, their inner product is one. To prove this, consider the following property of an order-2 homogeneous function like $\mathcal{H}$:

$$\frac{\partial \mathcal{H}}{\partial p_i} p_i = \frac{\partial \left( \frac{1}{2} \mathcal{F}_*^2 \right)}{\partial p_i} p_i = \mathcal{F}_*^2 \tag{54}$$

Combining Hamilton's equation for $\partial \mathcal{H} / \partial p_i$ (Eq. 49) with the definition of ray velocity $v^i = \mathrm{d} r^i / \mathrm{d} t$, and given the constant value of $\mathcal{F}_* = 1$ known from Eq. (30), this gives

$$\widetilde{\boldsymbol{p}}(\boldsymbol{v}) = p_i v^i = 1 \tag{55}$$

which is the definition of conjugacy.

If the process of erosion were isotropic, conjugacy would obviously be true: erosion velocity and normal slowness would be colinear, and since their magnitudes are mutually reciprocal, their product would be unity. However, the erosion process is manifestly not isotropic (see Sect. 3.18), which means that conjugacy also constrains the angular disparity between the ray and front-normal directions.

### 3.16 Constancy of the Lagrangian

We can exploit conjugacy to reveal important behaviour of the fundamental function $\mathcal{F}$ and the related Lagrangian $\mathcal{L}$. Since $\mathcal{L}$ is (like $\mathcal{H}$) order-2 homogeneous, it has the property

$$\frac{\partial \mathcal{L}}{\partial v^i} v^i = \frac{\partial \left( \frac{1}{2} \mathcal{F}^2 \right)}{\partial v^i} v^i = \mathcal{F}^2 \tag{56}$$

Using the fibre derivative form of $\widetilde{\boldsymbol{p}}$ in Eq. (36) and the definition of the Lagrangian in Eq. (35), we can deduce that, for physically valid ray trajectories,

$$\mathcal{F}^2 = \widetilde{\boldsymbol{p}}(\boldsymbol{v}) = p_i v^i = 1 \quad \Rightarrow \quad \mathcal{L}(\boldsymbol{r}, \boldsymbol{v}) = \frac{1}{2} \tag{57}$$

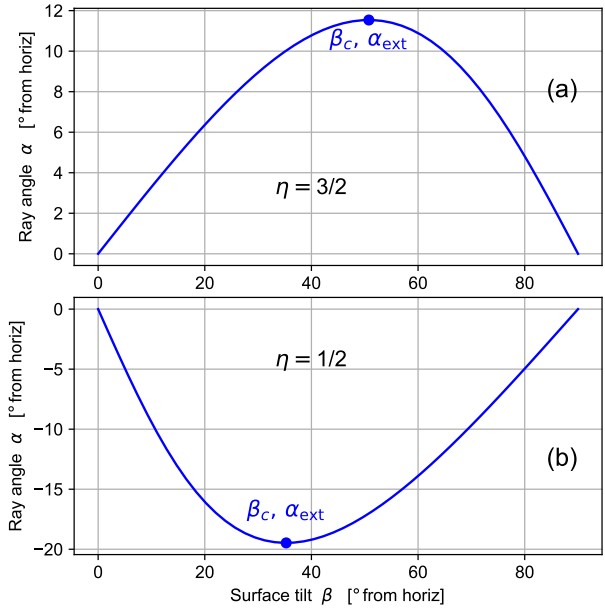

**Figure 7.** Variation of ray dip $\alpha$ with surface tilt $\beta$ for $\eta = \frac{3}{2}, \frac{1}{2}$.

In other words, the Lagrangian has the constant value of $\frac{1}{2}$—just like the Hamiltonian (Eq. 34)—meaning that is is only those points with positions $r$ and velocities $v$ satisfying this equation that represent points on a moving erosion surface.

This shows that the geomorphic surface Lagrangian and Hamiltonian are both static given the model assumptions made here, such as constant external forcing and domain symmetry (Fig. 6): a more general theory that relaxes these restrictions would lead to non-constancy of $\mathcal{L}$ and $\mathcal{H}$.

### 3.17 Ray angle

An essential measure of ray direction is the angle $\alpha$ of the velocity vector $v$ defined relative to horizontal:

$$\tan \alpha := \frac{v^z}{v^x} \tag{58}$$

This definition, along with that for $\beta$ given in Eq. (27), allow us to manipulate Hamilton's equations for the components of $\dot{r}$ (see Eq. 50) for which

$$\frac{v^z}{v^x} = -\frac{p_x^{2\eta} p_x^{1-2\eta} p_z (\eta - 1)}{\eta p_z^2 + p_x^2} \tag{59}$$

into a relationship between the two angles (Fig. 7):

$$\tan \alpha = \frac{(\eta - 1) \tan \beta}{\eta + \tan^2 \beta} \tag{60}$$

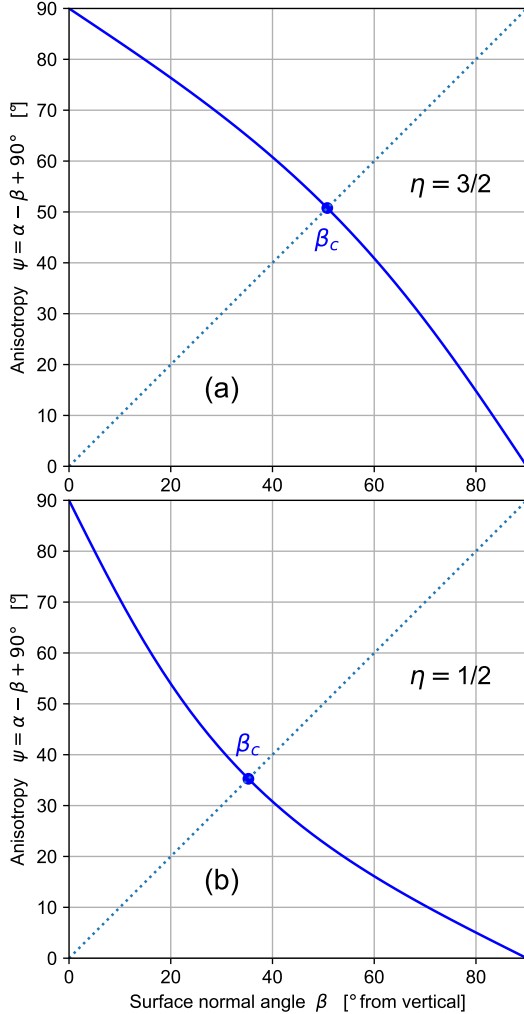

**Figure 8.** Erosional anisotropy measured using ray vs. normal angular disparity $\psi = \alpha - \beta + 90°$: variation with surface tilt $\beta$ shown for **(a)** $\eta = \frac{3}{2}$ and **(b)** $\eta = \frac{1}{2}$, and with $\mu/\eta = \frac{1}{2}$.

which inverts to give

$$\tan\beta = \frac{\eta \pm \sqrt{\eta^2 - 4\eta\tan^2\alpha - 2\eta + 1} - 1}{2\tan\alpha} \tag{61}$$

where the choice of root depends on how far the point is along the ray trajectory (see below). By comparing $\alpha$ and $\beta$ we can measure erosional anisotropy (see Sect. 3.18).

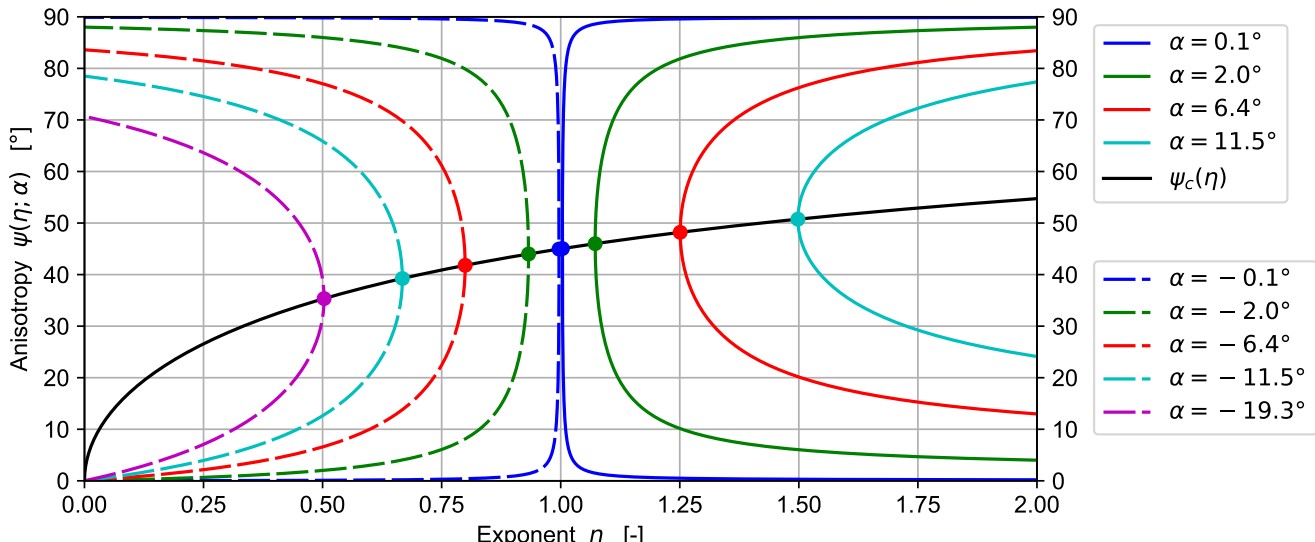

**Figure 9.** Ray anisotropy $\psi(\eta;\alpha)$ (colour curves) as a function of gradient exponent $\eta$, and its value $\psi_c$ (black line and solid circles) at the ray angle extremum $\alpha_{\text{ext}}$, for a selection of ray angles $\alpha \in \{\pm0.1°, \pm2°, \pm6.4°, \pm11.5°, -19.3°\}$.

Examination of Eqs. (58) and (59) reveals an important property of the vertical motion of erosion rays and its dependence on $\eta$. Since $p_x > 0$ and $p_z < 0$ in the model half-space, and because $v_x > 0$,

$$\alpha > 0 \quad \Leftrightarrow \quad \text{rays point up} \qquad\qquad \text{for} \quad \eta > 1$$
$$\alpha = 0 \quad \Leftrightarrow \quad \text{rays are horizontal} \qquad \text{for} \quad \eta = 1$$
$$\alpha < 0 \quad \Leftrightarrow \quad \text{rays point down} \qquad\quad \text{for} \quad \eta < 1$$

This switch in ray orientation as a function of slope scaling exponent $\eta$, which is illustrated in Fig. 12, echoes the observations in 1+1D of Weissel and Seidl (1998) and Royden and Perron (2013) of a change in upstream propagation with their gradient scaling exponent $n$. As their work has shown, this switch has important consequences for how and when knickpoints form (Stark and Stark, 2022).

The ray angle function (Eq. 60) has an extremum whose value is given by:

$$\tan\alpha_{\text{ext}} = \frac{\eta - 1}{2\sqrt{\eta}} \tag{62}$$

This extremum represents a bound on permissible values of ray angle $\alpha$. For $\eta > 1$, the extremum is positive $\alpha_{\text{ext}} > 0$ and rays cannot point up more steeply than $\alpha < \alpha_{\text{ext}}$, while for $\eta < 1$, the extremum is negative $\alpha_{\text{ext}} < 0$ and rays point down at negative angles limited by $\alpha > \alpha_{\text{ext}}$. The extremum is located at a critical value of $\beta$:

$$\tan\beta_c = \sqrt{\eta} \tag{63}$$

For $\eta = \frac{3}{2}$, the critical surface tilt is $\beta_c = 50.77°$, while for $\eta = \frac{1}{2}$ the critical tilt is $\beta_c = 35.26°$ (see Fig. 7). At this critical angle the Lagrangian and the metric tensor are singular, which means that if the surface tilt reaches this angle, the link between $\mathcal{H}$ and $\mathcal{L}$ is broken, $\mathcal{F}_*$ and $\mathcal{F}$ are no longer metric functions, and the model space is no longer (pseudo) Finsler. What this means in practice is not yet clear; the critical angle may manifest as a transition in landscape geometric behaviour, but we can only speculate at this stage: further study is needed.

### 3.18 Erosional anisotropy

The difference between the erosion ray angle $\alpha$ and the erosion front-normal angle $\beta$ (rotated by $90°$ such that both angles are measured relative to horizontal) quantifies the anisotropy of the erosion process:

$$\psi := \alpha - \beta + 90° \tag{64}$$

Defined in this way, $\psi = 0°$ for isotropic motion and $\psi = 90°$ when anisotropy is so strong that rays and surface normal are orthogonal.

Figure 8 shows how $\psi$ varies with surface tilt $\beta$ when computed along a time-invariant profile for $\eta = \frac{3}{2}$ and $\eta = \frac{1}{2}$. As these plots demonstrate, the gradient-dependent erosion process described by Eq. (25) is strongly anisotropic.

Figure 9 illustrates how anisotropy varies as a function of gradient-scaling exponent $\eta$ for a selection of ray angles $\alpha$. As predicted in the previous section, the rays all point upwards (positive $\alpha$) for $\eta > 1$ and downwards (negative $\alpha$) for $\eta < 1$. Broadly speaking, anisotropy $\psi$ reaches greater extremes for larger absolute values of $|\eta - 1|$.

The physical relevance of anisotropy $\psi$ is revealed by the following. The surface-normal erosion rate can be computed from ray velocity by exploiting ray-front conjugacy (Eq. 55), which is equivalent to a dot product between ray vector and surface-normal slowness

$$\widetilde{p}(v) = pv\cos\psi = pv\cos(\alpha - \beta + 90°) = 1 \tag{65}$$

and by using the reciprocal relationship between erosion slowness and erosion speed $p = 1/\xi^\perp$ (Eq. 18), to get

$$v = \frac{\xi^\perp}{\cos\psi} \tag{66}$$

While surface erosion takes place at a speed $\xi^\perp$, changes in external boundary conditions propagate much faster into the landscape along an erosion ray trajectory with a speed $\xi^\perp \sec\psi$. The two are related by projecting the ray vector $v$ onto the local unit surface-normal vector, which lies at an relative angle $\psi$ relative to the ray.

### 3.19 Measuring slope along the erosion front

Since the Hamiltonian tracks motion of the erosion front in a phase space spanned in part by the surface-normal covector, solutions of front motion have the surface gradient encoded into them. Therefore the gradient along the evolving topographic surface can be tracked in three distinct ways. One method is to take the ratio of the covector components:

$$\tan\beta_p := \tan\beta = -\frac{p_x}{p_z} \tag{67}$$

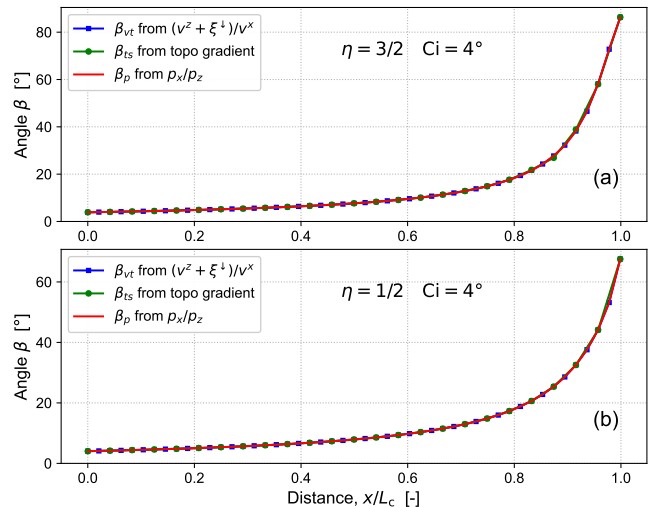

**Figure 10.** Estimation of the surface-normal angle from vertical $\beta$, aka the angle of the surface from horizontal for **(a)** $\eta = \frac{3}{2}$ and **(b)** $\eta = \frac{1}{2}$, and with $\mu/\eta = \frac{1}{2}$ and $\mathsf{Ci} = 4°$. This angle can be computed in three ways; their mutual consistency shown here provides a partial validation of the ray tracing method.

A second method is to compute the topographic gradient:

$$\tan\beta_{ts} := \frac{\mathrm{d}z}{\mathrm{d}x} \qquad \text{for} \quad \{x, z\} \in T(x, z) \tag{68}$$

In a numerical solution, this entails making a finite-difference approximation using values at nearest neighbour points. A third method is to construct a velocity triangle from the ray velocity components and the reciprocal covector slowness in the vertical direction, aka the vertical erosion rate:

$$\tan\beta_{vt} := \frac{v^z - 1/p_z}{v^x} = \frac{v^z + \xi^{\downarrow}}{v^x} \tag{69}$$

Ideally, all three measurements of the topographic gradient should be equal. In practice, $\beta_{ts}$ is computed non-locally while $\beta_p$ and $\beta_{vt}$ are strictly local but numerically different computations; we therefore expect the three estimates to be equal to within a precision set by choices such as ray density, time step and interpolation method. A comparison of the methods is given in Fig. 10.

## 4 Implementation

To keep development of a geomorphic Hamiltonian theory as simple as possible, the treatment so far (Sect. 3) has employed a somewhat abstract erosion model: it has assumed the erosion rate can be written as some combination of a power function of surface tilt and a spatially variable (but constant in time) function that encompasses flow rate, flow geometry, substrate erodibility, and so on. If we want to probe any further the behaviour of the geomorphic surface Hamiltonian and its implications

for landscape erosion, we need to choose a particular form for the flow function component and to parameterize this spatial dependence. Bear in mind, though, that more general erosion models could also be transformed into Hamiltonian form and subjected to the analyses presented below.

## 4.1 A modified stream-power incision model

Previous studies related to our work (Luke, 1972; Royden and Perron, 2013; Weissel and Seidl, 1998) have focused on the stream power incision model (SPIM) (e.g., Lague, 2014). In order maintain a clear conceptual link with these studies, and because SPIM can be adapted to satisfy the simplifying criteria adopted in Section 3.5, we use it here in a modified form. SPIM asserts that, in channels,

$$\text{vertical erosion rate} \propto (\text{area})^m (\text{slope})^n \tag{70}$$

where "slope" is the channel gradient $\tan\beta$, and where upstream area, suitably scaled, is assumed to be a good composite proxy for the volumetric flow of water per contour width and its contributions to channel geometry, boundary flow, sediment transport, and rock surface abrasion. We modify this equation so that it instead tracks

$$\text{surface-normal erosion rate} \sim (\text{area})^\mu (\text{slope})^\eta \tag{71}$$

where "slope" is now $\sin\beta$. This model and classic SPIM coincide if $\eta = 1$, since $\xi^\downarrow = \xi^\perp / \cos\beta$ (Eq. 10), although they differ
somewhat otherwise. Given this similarity, we can treat as roughly equivalent the slope and area exponents $\eta \Leftrightarrow n$ and $\mu \Leftrightarrow m$.

Our model domain is a 2D transect along a channel, which means we have to parameterize out catchment geometry and drainage accumulation into a function of distance downstream. If we consider upstream area to scale with an offset distance from the divide $L_c - (x - \varepsilon)$, where $\varepsilon$ is a very small regularization term, we can wrap this scaling into a power function form for the flow component of the erosion model:

$$650 \quad \varphi(x) \; := \; \varphi_0 \left(L_c - x + \varepsilon\right)^{2\mu} \sim \varphi_0 (\text{upstream area})^\mu \tag{72}$$

In the numerical solutions presented in Section 6, the regularization term $\varepsilon$ is given a non-zero value, but in the equations below it is ignored.

The surface-normal channel erosion rate is then

$$\xi^\perp \; := \; \varphi_0 \left(L_c - x\right)^{2\mu} (\sin\beta)^\eta \tag{73}$$

In a similar manner to steady (constant erosion rate) solutions of SPIM (e.g., Lague, 2014), this model will generate channel profiles with the asymptotic slope-area scaling

$$\text{slope} \sim \text{area}^{-\mu/\eta} \tag{74}$$

assuming low-to-moderate slope angles where $\tan\beta \approx \sin\beta$. To ensure that our numerical simulations all yield slope-area scaling consistent with that typically observed (e.g., Beeson and McCoy, 2020; Flint, 1974; Lague, 2014; Royden and Perron,
2013), we fix the exponent ratio (aka "concavity index") at a constant $\mu/\eta = \frac{1}{2}$.

## 4.2 Non-dimensionalization

Before embarking on numerical solutions of the model, we non-dimensionalize it. This is helpful in two ways: (i) it requires us to identify the characteristic length, time, erosion rate and slowness scales, which makes it easier to relate the model to real-world landscapes; (ii) it makes generalization of model behaviour and solution geometries simpler.

An obvious length scale is the horizontal channel length $L_c$, aka the distance from the drainage divide $x = L_c$ to the channel terminus $x = 0$. The horizontal and vertical erosion rates at the terminus are

$$\xi^{\to 0} = \varphi_0 L_c{}^{2\mu} \left(\sin\beta_0\right)^{\eta-1}, \quad \xi^{\downarrow 0} = \frac{\varphi_0 L_c{}^{2\mu} \left(\sin\beta_0\right)^{\eta}}{\cos\beta_0} \tag{75}$$

where $\xi^{\to 0}/\xi^{\downarrow 0} = \tan\beta_0$, and where the channel tilt angle at the terminus is

$$\beta_0 := \beta|_{x=0} \tag{76}$$

We choose $\xi^{\to 0}$ as the characteristic velocity scale. The horizontal time scale is therefore

$$t^{\to 0} := \frac{L_c}{\xi^{\to 0}} = \frac{L_c{}^{1-2\mu}}{\varphi_0} \left(\sin\beta_0\right)^{1-\eta} \tag{77}$$

The vertical time scale is given by $t_{\downarrow 0} = t^{\to 0} \cot\beta_0$.

    Now we can non-dimensionalize the primary model variables:

$$\hat{t} := \frac{t}{t^{\to 0}}, \qquad \hat{\boldsymbol{r}} := \frac{\boldsymbol{r}}{L_c}, \qquad \hat{\boldsymbol{p}} := \xi^{\to 0} \widetilde{\boldsymbol{p}} \tag{78}$$

and the coordinate axes

$$\hat{x} := \frac{x}{L_c}, \qquad \hat{z} := \frac{z}{L_c} \tag{79}$$

Using them to rewrite the Hamiltonian we get

$$\mathcal{H}(\hat{\boldsymbol{r}}, \hat{\boldsymbol{p}}) = \frac{\left(1-\hat{r}^x\right)^{4\mu} \hat{p}_x^{2\eta}}{2\left(\sin^2\mathsf{Ci}\right)^{\eta-1}\left(\hat{p}_x^2 + \hat{p}_z^2\right)^{\eta-1}} \tag{80}$$

where we have defined the dimensionless number

$\mathsf{Ci} := \arcsin\left(\left(\dfrac{\varphi_0 L_c{}^{2\mu}}{\xi^{\to 0}}\right)^{\frac{1}{1-\eta}}\right) = \beta_0$       (81)

We can think of $\mathsf{Ci}$ as both an angle and a dimensionless erosion rate because, when we non-dimensionalize the vertical rate of erosion imposed at the boundary $\xi^{\downarrow 0}$, we get this:

$$\xi^{\downarrow 0}/\xi^{\to 0} = \tan\beta_0 = \tan\mathsf{Ci} \tag{82}$$

Note that we can write

$\varphi(\hat{r}^x) = \varphi_0 L_c{}^{2\mu} \left(1-\hat{r}^x\right)^{2\mu} = \dfrac{\xi^{\to 0} \left(1-\hat{r}^x\right)^{2\mu}}{\left(\sin\mathsf{Ci}\right)^{\eta-1}}$       (83)

We can now rewrite Hamilton's equations in dimensionless form by rederiving them from Eq. (80). Or we can just substitute the non-dimensionalized variables into Eqs. (50) and (51):

$$\frac{\mathrm{d}\hat{\boldsymbol{r}}}{\mathrm{d}\hat{t}} := \frac{t^{\to o}}{L_c} \frac{\mathrm{d}\boldsymbol{r}}{\mathrm{d}t} = \frac{1}{\xi^{\to o}} \frac{\mathrm{d}\boldsymbol{r}}{\mathrm{d}t} \tag{84}$$

$$\frac{\mathrm{d}\hat{\boldsymbol{p}}}{\mathrm{d}\hat{t}} := \xi^{\to o} t^{\to o} \frac{\mathrm{d}\widetilde{\boldsymbol{p}}}{\mathrm{d}t} = L_c \frac{\mathrm{d}\widetilde{\boldsymbol{p}}}{\mathrm{d}t} \tag{85}$$

and so we get:

$$\frac{\mathrm{d}\hat{\boldsymbol{r}}}{\mathrm{d}\hat{t}} = \frac{\partial \mathcal{H}}{\partial \hat{\boldsymbol{p}}} = \frac{1}{\xi^{\to o}} \frac{\partial \mathcal{H}}{\partial \widetilde{\boldsymbol{p}}} \tag{86}$$

$$= \frac{(1-\hat{r}^x)^{4\mu}}{(\sin^2 \mathsf{Ci})^{\eta-1}} \frac{\hat{p}_x^{2\eta-1}}{(\hat{p}_x^2 + \hat{p}_z^2)^\eta} \begin{bmatrix} (\hat{p}_x^2 + \eta \hat{p}_z^2) \\ (1-\eta)\hat{p}_x \hat{p}_z \end{bmatrix}$$

and

$$\frac{\mathrm{d}\hat{\boldsymbol{p}}}{\mathrm{d}\hat{t}} = -\frac{\partial \mathcal{H}}{\partial \hat{\boldsymbol{r}}} = -L_c \frac{\partial \mathcal{H}}{\partial \boldsymbol{r}} \tag{87}$$

$$= \frac{2\mu (1-\hat{r}^x)^{4\mu-1}}{(\sin^2 \mathsf{Ci})^{\eta-1}} \frac{\hat{p}_x^{2\eta}}{(\hat{p}_x^2 + \hat{p}_z^2)^{\eta-1}} \begin{bmatrix} 1 & 0 \end{bmatrix}$$

Figure 11 provides a comparison of time-invariant stream profiles for a selection of values of the dimensionless horizontal erosion rate $\mathsf{Ci} \in \{0.1°, 1°, 4°\}$. In all other figures illustrating numerical solutions a value of this dimensionless number is set at $\mathsf{Ci} = 4°$.

## 4.3 Direct integration

For the simple scenario of a time-invariant profile, the erosion equation (Eq. 73) can be directly integrated; more complex boundary and initial conditions do not allow it. The first step is to assume the vertical rate of erosion is constant everywhere $\xi^\downarrow = \xi^{\downarrow o}$, and thereby to manipulate Eq. (73) to expose its straightforward dependence on surface tilt $\beta$ and position $x$ (through $\varphi(x)$):

$$\xi^\downarrow = \frac{\xi^\perp}{\cos \beta} = \frac{\varphi(x) |\sin \beta|^\eta}{\cos \beta} \tag{88}$$

We can combine this equation with Eq. (68) to obtain a polynomial in surface gradient $\tan \beta = \mathrm{d}z/\mathrm{d}x$, and constrain it using the result (Eq. 53) that $-p_z = 1/\xi^\downarrow = 1/\xi^{\downarrow o}$ along the whole ray and thus everywhere along a time-invariant profile. The resulting polynomial in surface gradient, in non-dimensionalized form and for rational values of the gradient exponent such as $\eta = \frac{3}{2}$ or $\eta = \frac{1}{2}$, is

$$\frac{\mathrm{d}\hat{z}}{\mathrm{d}\hat{x}}^{4\eta} \left( \frac{\mathrm{d}\hat{z}}{\mathrm{d}\hat{x}}^2 + 1 \right)^{2-2\eta} - \frac{\sin^{4\eta}(\mathsf{Ci})}{(1-\hat{x})^{8\mu} \cos^4(\mathsf{Ci})} = 0 \tag{89}$$

We can use this function to compute the surface elevation as a 1+1D function $\hat{z}(\hat{x}; \eta, \mu, \mathsf{Ci})$ as follows: (1) pick values of $\eta$, $\mu$, and $\mathsf{Ci}$; (2) substitute these numbers into the above function to generate a polynomial in $\mathrm{d}\hat{z}/\mathrm{d}\hat{x}$ and $\hat{x}$; (3) define a set of

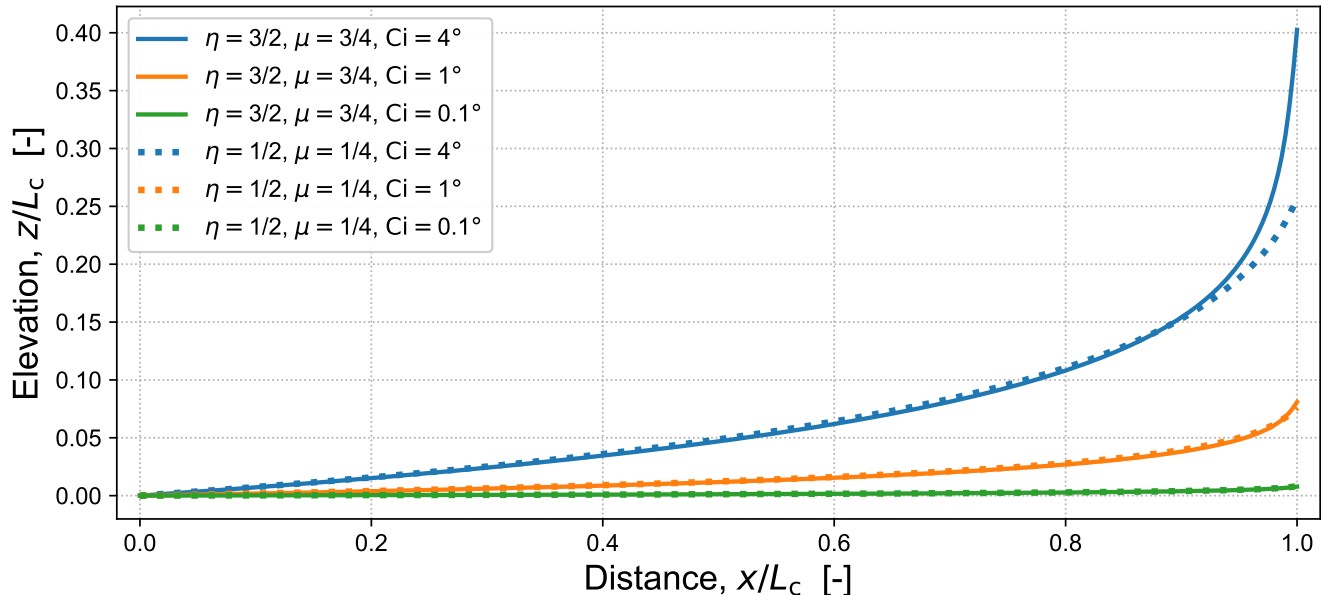

**Figure 11.** Time-invariant profiles (shown in non-dimensionalized form) obtained by direct integration of the model (Sect. 4.3) for two choices of $\eta \in \{\frac{1}{2}, \frac{3}{2}\}$ and three choices of $\text{Ci} \in \{0.1°, 1°, 4°\}$; for each value $\eta$, the flow exponent $\mu$ is chosen such that $\mu/\eta = \frac{1}{2}$. The channel incision number $\text{Ci}$ sets the overall steepness since it effectively defines the gradient at the exit $x = 0$. Given a value of $\text{Ci}$, the profiles for $\eta = \frac{3}{2}$ and $\eta = \frac{1}{2}$ are approximately the same until $x \geq 0.95L_\text{c}$.

sample positions $0 \leq \{\hat{x}\} < 1$ along the profile; (4) at each $\hat{x}$, find the positive, real root of this polynomial to infer the gradient $\text{d}\hat{z}/\text{d}\hat{x}$ at this position; (5) use quadrature to integrate the gradient values along the profile to get $\hat{z}(\hat{x})$.

Fig. 11 shows a selection of non-dimensionalized time-invariant profiles obtained in this way. Notice how the profiles for
the two different gradient exponents $\eta = \frac{3}{2}$ and $\eta = \frac{1}{2}$ are essentially colinear for $0 \leq \hat{x} = x/L_\text{c} < 0.95$. The practical upshot of this similarity is that it is unreasonable to expect to infer the scaling exponents $\eta$ and $\mu$ from topography alone.

Direct integrations like this are also useful as a validation of the ray-traced solutions: this is illustrated in Fig. 13, in which some examples of directly integrated time-invariant profiles are shown to match those obtained by ray tracing.

## 5   Ray tracing solutions

The previous sections have shown how the geometric self-constraint implicit in a broad class of erosion models can be transformed into a geomorphic surface Hamiltonian $\mathcal{H}$ (Eqs. 32, 80), and how this function can be used to derive Hamilton's equations of motion for points on an erosion surface (Eqs. 50, 51, 86, 87). In this section we solve Hamilton's equations by numerical integration and use them to construct "steady-state", time-invariant surface profiles driven by a constant erosion-rate boundary conditions. In all solutions presented below, the dimensionless horizontal erosion rate is set at $\text{Ci} = 4°$.

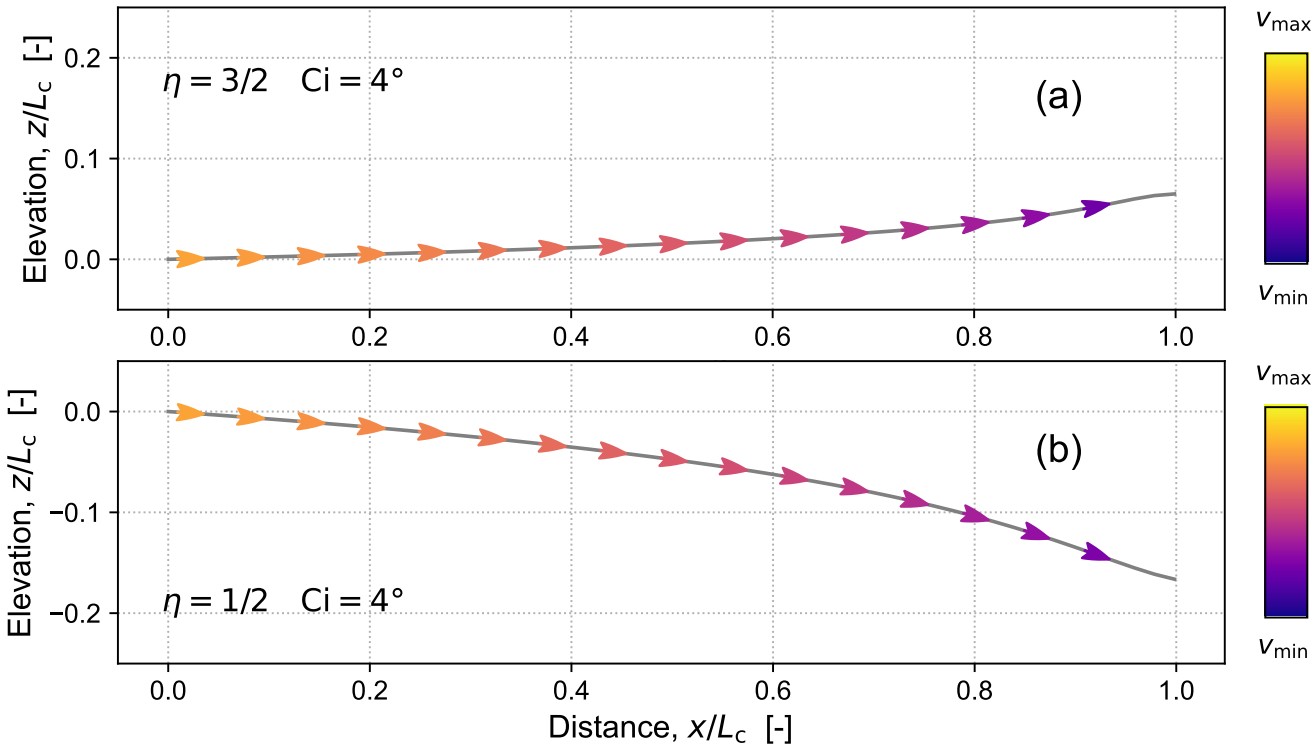

**Figure 12.** Tracing of a reference ray for **(a)** $\eta = \frac{3}{2}$ and **(b)** $\eta = \frac{1}{2}$, with $\mu/\eta = \frac{1}{2}$ and Ci $= 4°$, obtained by numerically integrating Hamilton's equations (Eqs. 50, 51) from a constant-slip boundary at $x = 0$ across the domain until termination at the divide at $x = L_c$.

## 5.1 Model domain and boundary conditions

The domain is a vertical $x$–$z$ transect (Fig. 2) along a stream profile that ranges from a drainage divide at $x = L_c$ to a flow-exit boundary at $x = 0$. Profile evolution is driven by a constant vertical erosion rate imposed at the exit, and evolution of the profile is tracked relative to the elevation of the exit. The drainage divide is pinned at a fixed horizontal position by mirroring (Fig. 6) the main profile with a symmetrical "image" profile spanning $L_c \le x \le 2L_c$; solution need only be performed over $0 \le x \le L_c$. Although there is no need to invoke tectonic processes here, note that this model is geometrically equivalent to erosion of a (half) horst block whose uniform rock uplift is driven by constant-rate vertical slip along a bounding normal fault, and whose topographic evolution is studied in the reference frame of the hanging wall.

## 5.2 Ray equations

In this model geometry, rays that initiate at $x = 0$ (Fig. 12) and propagate in the positive $x$ direction are annihilated at $x = L_c$ when a paired ray, initiated at the same time at $x = 2L_c$, arrives from the opposite direction. As such, the model induces a cusp

to form at $x = L_c$, although its formation is not explicitly modelled here—instead, rays from $x = 0$ are simply truncated at $x = L_c$.

Such ray tracing entails the numerical integration of Hamilton's equations in the form of four coupled, first-order ODEs for $\dot{r}^x$ and $\dot{r}^z$ (Eq. 50), $\dot{p}_x$ and $\dot{p}_z$ (Eq. 51). These are first-order differential equations in time alone, so for each ray we need only supply four initial conditions, i.e., $r^{x_0}, r^{z_0}, p_{x_0}, p_{z_0}$, one for each ray ODE. An oddity of ray tracing is that what would be boundary conditions in a partial differential equation (PDE) treatment become initial conditions for the rays, and what would be a separate Neumann velocity boundary condition for a PDE gets wrapped into those initial conditions.

Here we focus on obtaining the time-invariant profile generated by a constant vertical velocity boundary condition $\xi^\downarrow = \xi^{\downarrow_0}$ at $x = 0$, for which we only need to perform ray tracing from $x = 0$. We thus avoid having to generate rays along an initial topography and having to handle their transient interaction as the time-invariant profile develops (a topic to be addressed in Stark and Stark (2022)).

The initial horizontal position for all rays is fixed at the stream terminus/location of the boundary condition $r^{x_0} = x = 0$ (Fig. 2). The initial vertical position of a ray initiated at time $t = t_0$ is given by simple integration of the vertical erosion rate: $r^{z_0} = -\xi^{\downarrow_0} t_0$. The initial vertical component of the ray slowness covector must be consistent with this vertical velocity component, and so we have $p_{z_0} = -1/\xi^{\downarrow_0}$. Since $\dot{p}_z = 0$, this vertical covector component remains unchanged throughout ray propagation (see Sect. 3.14), and so the number of coupled ODEs that need to be solved is effectively reduced from four to three.

The initial horizontal component of the slowness covector can be calculated if we realize that the topographic gradient at the boundary must be consistent with the orientation of the normal slowness, i.e., $\tan\beta_0 = -p_{x_0}/p_{z_0}$. As such, the initial value of the slowness covector $\widetilde{p}$ encodes the velocity boundary condition in both its direction and magnitude.

## 5.3 Numerical integration method

After some experimentation, the most accurate quadrature or numerical integration scheme for ray tracing with Eqs. (50) and (51) was found to be an implicit Runge-Kutta method designed for stiff ODEs: specifically, an implementation of the Radau IIA family of order 5 (see Hairer and Wanner, 2013, p. 72) provided by the Python package SciPy (Virtanen et al., 2020). Simpler, and lower-order Runge-Kutta quadrature methods also work well for most choices of model parameters, as does the high-order Runge-Kutta, dense output, DOP853 method (see Hairer et al., 2008, p. 194).

All the numerical solutions presented here are reproducible using the following open source software (split into two parts, both of which are needed for full operation): (1) the `GME` package, which implements methods of geometric mechanics tailored to treating geomorphic erosion (v. 1.0: Stark, 2021a, c); and (2) a utilities library called `GMPLib` (v. 1.0: Stark, 2021b, d).

## 5.4 Reference ray construction

Computation of the trajectory of a point on an erosion surface (and its normal slowness covector) is carried out by numerically integrating the coupled set of Hamilton's equations (dimensioned: Eqs. 50 & 51; non-dimensionalized: Eqs. 86 & 87) with the boundary conditions described in Sect. 5.1. This constitutes the tracing of a single reference ray (Fig. 12), which suffices for

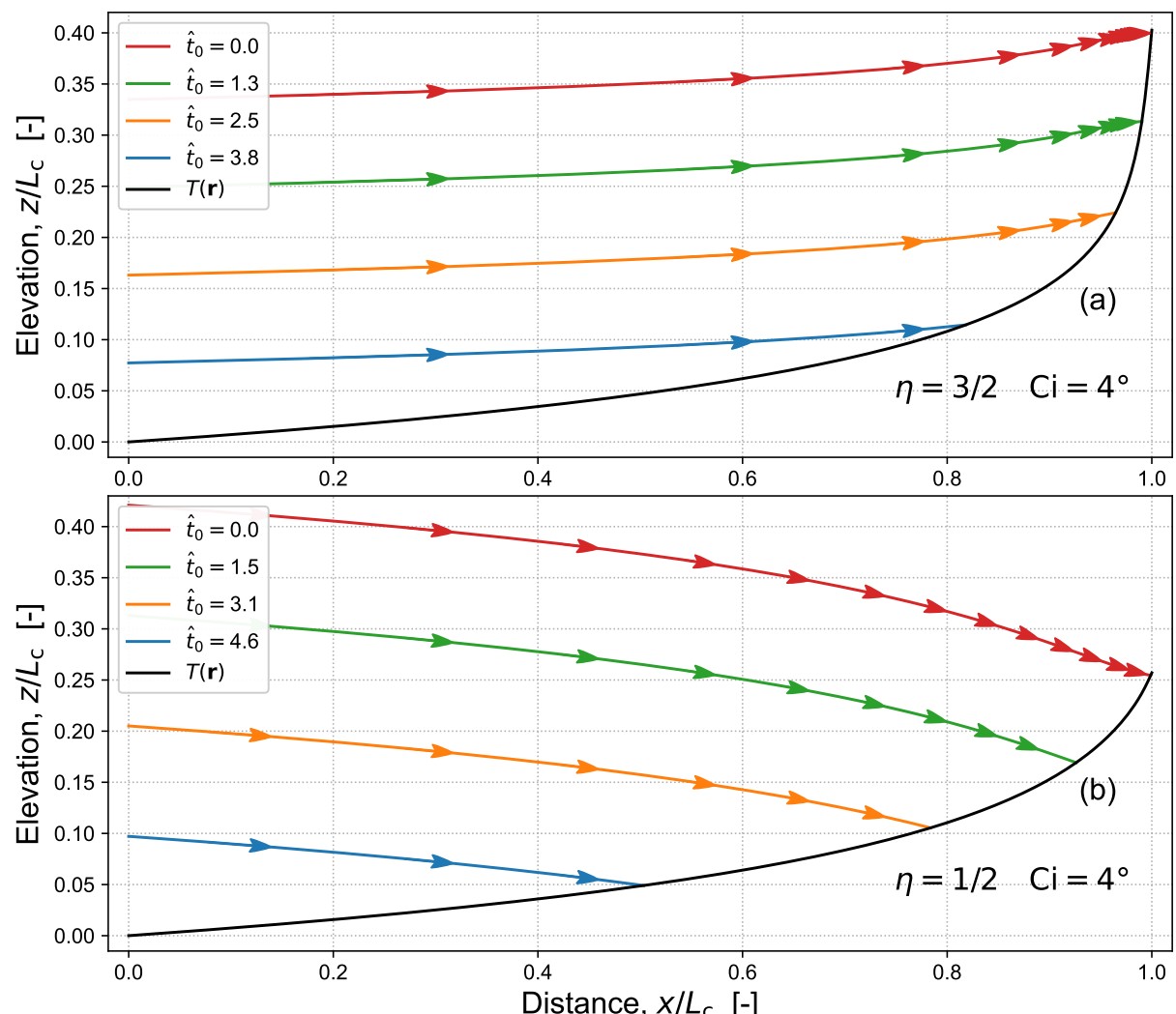

**Figure 13.** Comparison of ray-traced solutions of time-invariant profiles (black curves) for **(a)** $\eta = \frac{3}{2}$ and **(b)** $\eta = \frac{1}{2}$, and with $\mu/\eta = \frac{1}{2}$ and $Ci = 4°$. A reference ray solution was obtained (Fig. 12) by numerically integrating Hamilton's equations (Eqs. 86 & 87) from $\hat{x} = 0$ across the domain until termination at the divide at $\hat{x} = x/L_c = 1$. Successive rays were then generated with initiation times $\{\hat{t}_0\}$ and initial elevations $\{\hat{z}(\hat{t}_0)\}$ consistent with the constant vertical erosion rate imposed at $\hat{x} = 0$: four are shown here (arrowed curves). Each time-invariant profile $T(\mathbf{r})$ was generated both from the ensemble of rays and by direct integration (Sect. 4.3); the results match in each case.

construction of a time-invariant topographic profile (see below). More rays need to be traced if we want to handle time-variable boundary conditions, evolution from an initial topography, or the transition between an initial surface and a slip boundary (Stark and Stark, 2022).

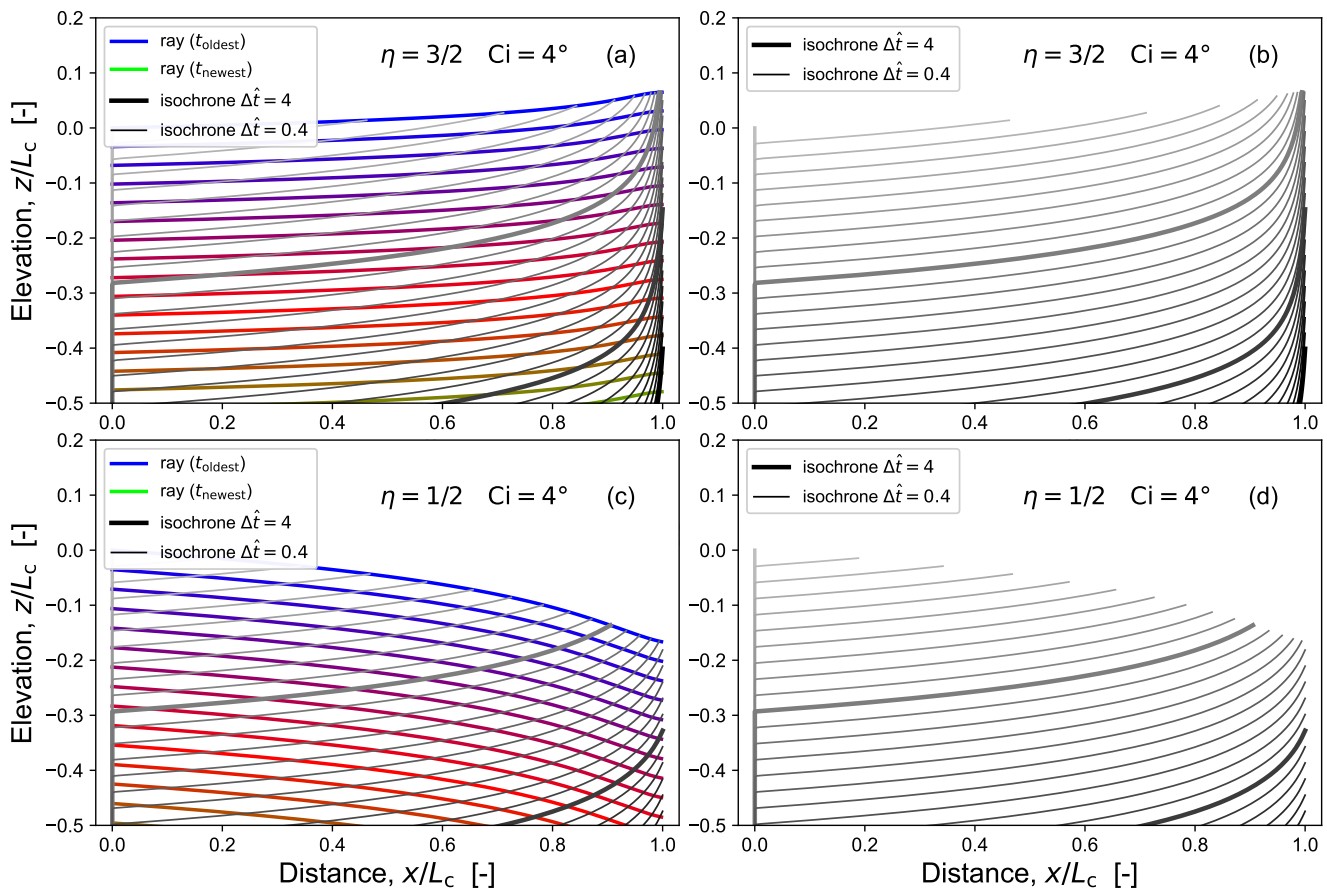

**Figure 14.** Ray tracing construction of erosion surfaces or isochrones: **(a), (b)** $\eta = \frac{3}{2}$; **(c), (d)** $\eta = \frac{1}{2}$, and with $\mu/\eta = \frac{1}{2}$ and Ci $= 4°$. Only a subset of the resolved rays and isochrones is shown.

## 5.5 Synthesis of a time-invariant profile

The following steps are required to construct a time-invariant solution of the erosion equation akin to a fault-driven steady-state solution (Figs. 6 & 13):

1. choose values for the model parameters (notably gradient-scaling exponent $\eta$ and upstream area-scaling exponent $\mu$)

2. specify the dimensionless vertical erosion rate at the boundary Ci

3. generate a reference ray $\boldsymbol{r}_{\text{ref}}(t)$ by integrating Eqs. (50) and (51) (or their non-dimensionalized equivalents Eqs. (86) and (87)) from the boundary at $(0, r_{z_0})$, and assign it an initiation time of $t_0 = 0$;

4. define the isochrone time $T$ such that $r^x_{\text{ref}}(T) = L_{\text{c}}$;

5. generate a $k^{\text{th}}$ later ray $\boldsymbol{r}_{k\Delta t_0}(t + k\Delta t_0)$ with initiation time $k\Delta t_0$ by making a copy of the reference ray, displacing it vertically by $-\xi^{\downarrow 0}k\Delta t_0$, and pasting it at $(0, r_{z_0} - \xi^{\downarrow 0}k\Delta t_0)$;

6. truncate the copied ray at the point $\boldsymbol{r}_{k\Delta t_0}(T - k\Delta t)$;

7. repeat from step 4 until $k\Delta t_0 \geq T$;

8. collate the truncation points to generate a continuous curve $T(\boldsymbol{r})$.

Some of these steps also entail interpolation and resampling.

This procedure generates the time-invariant isochrone $T(\boldsymbol{r})$ formed by the constant vertical velocity $\xi^{\downarrow 0}$ boundary condition at $x = 0$ (Figs. 6, 13). Repetition of the procedure (or a simple copying of the solution), combined with a progressive offset of the initial ray location $r_{z_0}$ at the boundary, simulates vertical normal-fault-driven erosion of a topographic profile at steady state in the reference frame of the (bedrock) substrate of the footwall bedrock (Fig. 14). Analysis of these composite results generates solutions for the along-profile variations in the component erosion rates (Figs. 15c–e, 16c–e) and their anisotropy (Figs. 15a, 16a, 17).

In all the solutions presented here, the area-scaling exponent $\mu$ is chosen such that $\mu/\eta = \frac{1}{2}$. In all but Fig. 11 the dimensionless rate of boundary erosion (Eq. 81) is fixed at $\mathsf{Ci} = 4°$.

## 6   Results

In this section we present numerical solutions of time-invariant topographic profiles in dimensionless form. These solutions help to validate the geomorphic surface Hamiltonian (Sects. 1.1–3), to test the inferences drawn from it (Sects. 3 & 4), to examine its non-dimensionalization (Sect. 4.2) and the time/length/velocity scales predicted by it (Sect. 6.1), to check how ray tracing by integrating Hamilton's equations performs as a means of modelling surface erosion and the propagation of boundary-change information (Sect. 5.4; Figs. 12–14), and to explore how erosional anisotropy $\psi$ varies across a landscape.

Although the solutions here are limited to a 2D $x$–$z$ transect, they provide a pilot test of elements needed to construct a fully 3D landscape evolution model around a geomorphic Hamiltonian: one in which (1) the denudation rate is defined as acting in the surface-normal direction, rather than purely vertically, and (2) topographic elevation is tracked as true geometric surface using an implicit "time-slice" function $T(x, y, z)$, instead of being modelled as a field using an explicit height function $h(x, y; t)$.

### 6.1   Scales

Tables 1 and 2 provide some example values of model parameters and their corresponding time, rate and vertical scales. For each example, the key choice is the dimensionless horizontal erosion rate $\mathsf{Ci}$. This dimensionless number determines the dimensionless traversal time $\hat{t}_{L_c}^{\rightarrow}$, which is defined as the time it takes for a ray to travel from $x = 0$ to $x = 0.95L_c$, and which is obtained by numerical ray tracing. Then, by choosing the domain length $L_c$ and the boundary rate of vertical erosion,

**Table 1.** Example model parameters and predicted time scales for $\eta = \frac{3}{2}$ and $\mu = \frac{3}{4}$, and for selected values of dimensionless erosion rate Ci and domain length scale $L_c$.

| Ci | grad | $\hat{t}_{L_c}^{\rightarrow}$ | $L_c$ | $\xi^{\downarrow 0}/\xi^{\rightarrow 0}$ | $t^{\rightarrow 0}$ | $t_{L_c}^{\rightarrow}$ | $h_{L_c}$ |
|---|---|---|---|---|---|---|---|
| ° | % | − | km | mm/y | My | My | m |
| 4 | 7 | 2 | 10 | 1 / 14 | 0.7 | 1.5 | 2000 |
| 4 | 7 | 2 | 5 | 1 / 14 | 0.35 | 0.7 | 1000 |
| 1 | 2 | 2 | 100 | 1 / 60 | 2 | 3 | 4800 |
| 1 | 2 | 2 | 10 | 1 / 60 | 0.2 | 0.3 | 480 |
| 0.1 | 0.2 | 2 | 100 | 1 / 600 | 0.2 | 0.3 | 480 |
| 0.1 | 0.2 | 2 | 10 | 1 / 600 | 0.02 | 0.03 | 48 |
| 4 | 7 | 2 | 10 | 10 / 140 | 0.07 | 0.15 | 2000 |
| 4 | 7 | 2 | 5 | 10 / 140 | 0.035 | 0.07 | 1000 |
| 1 | 2 | 2 | 100 | 10 / 600 | 0.2 | 0.3 | 4800 |
| 1 | 2 | 2 | 10 | 10 / 600 | 0.02 | 0.03 | 480 |
| 0.1 | 0.2 | 2 | 100 | 10/6000 | 0.02 | 0.03 | 480 |
| 0.1 | 0.2 | 2 | 10 | 10/6000 | 0.002 | 0.003 | 48 |

dimensioned quantities can be computed. The parameters $\mathrm{grad} = \tan\beta_0$, $\xi^{\rightarrow 0}$, and $t^{\rightarrow 0}$ are derived exactly; the horizontal travel time $t_{L_c}^{\rightarrow}$ and the profile height $h_{L_c}$ close to the divide (at $x = 0.95 L_c$) are obtained by numerical solution. The values shown here are all rounded to one or two significant figures for clarity.

These tables demontrate that boosting the imposed vertical erosion rate $\xi^{\downarrow 0}$ linearly increases the consequent horizontal erosion rate $\xi^{\rightarrow 0}$, and symmetrically decreases $t^{\rightarrow 0}$ and $t_{L_c}^{\rightarrow}$, but has no effect on the profile height $h_{L_c}$. The most important result here is that by calculating the dimensionless traversal time $\hat{t}_{L_c}^{\rightarrow}$ we can estimate how long it takes for boundary change information to propagate into a landscape.

## 6.2 Time-invariant solutions

Figures 13 & 14 illustrate ray-traced time-invariant solutions for two choices of the slope exponent $\eta \in \left\{\frac{3}{2}, \frac{1}{2}\right\}$ in the model equation (Eq. 73) for surface-normal erosion rate $\xi^{\perp}$. Each ray-traced isochrone $T(\boldsymbol{r})$ is compared with an isochrone obtained by directly integration (Sect. 4.3), and in each case the match is excellent. Sequences of erosion surfaces resulting from similar time-invariant solutions are shown in Fig. 14.

**Table 2.** Example model parameters and predicted time scales for $\eta = \frac{1}{2}$ and $\mu = \frac{1}{4}$, and for selected values of dimensionless erosion rate Ci and domain length scale $L_c$.

| Ci | grad | $\hat{t}_{L_c}^{\rightarrow}$ | $L_c$ | $\xi^{\downarrow 0}/\xi^{\rightarrow 0}$ | $t^{\rightarrow 0}$ | $t_{L_c}^{\rightarrow}$ | $h_{L_c}$ |
|---|---|---|---|---|---|---|---|
| ° | % | − | km | mm/y | My | My | m |
| 4 | 7 | 5 | 10 | 1 / 14 | 0.7 | 3.4 | 1900 |
| 4 | 7 | 5 | 5 | 1 / 14 | 0.35 | 1.7 | 940 |
| 1 | 2 | 6 | 100 | 1 / 60 | 2 | 10 | 5000 |
| 1 | 2 | 6 | 10 | 1 / 60 | 0.2 | 1 | 500 |
| 0.1 | 0.2 | 6 | 100 | 1 / 600 | 0.2 | 1 | 500 |
| 0.1 | 0.2 | 6 | 10 | 1 / 600 | 0.02 | 0.1 | 50 |
| 4 | 7 | 5 | 10 | 10 / 140 | 0.07 | 0.34 | 1900 |
| 4 | 7 | 5 | 5 | 10 / 140 | 0.035 | 0.17 | 940 |
| 1 | 2 | 6 | 100 | 10 / 600 | 0.2 | 1 | 5000 |
| 1 | 2 | 6 | 10 | 10 / 600 | 0.02 | 0.1 | 500 |
| 0.1 | 0.2 | 6 | 100 | 10 / 6000 | 0.02 | 0.1 | 500 |
| 0.1 | 0.2 | 6 | 10 | 10 / 6000 | 0.002 | 0.01 | 50 |

These solutions illustrate an important behaviour of the rays: for values of the slope exponent $\eta > 1$ the ray velocities always have a positive vertical component $\dot{r}^z > 0$, whereas for $\eta < 1$, the vertical component $\dot{r}^z$ always has a negative vertical component $\dot{r}^z < 0$.

## 6.3 Erosion rates

Figures 15 (for $\eta = \frac{3}{2}$) and 16 (for $\eta = \frac{1}{2}$) provide a side-by-side comparison of surface erosion rate components ($\xi^{\perp}, \xi^{\rightarrow}, \xi^{\downarrow}$) along ray-traced time-invariant profiles, together with some of the variables that contribute to their variation (anisotropy $\psi$ and ray velocity components $v^x$, $v^z$). All plotted quantities are dimensionless.

As Figs. 15a–c and 16a–c show, the progressive upstream decrease in anisotropy $\psi$ is reflected in upstream decreases ray velocity (particularly the vertical component $v^z$) and the surface-normal erosion rate $\xi^{\perp}$. The horizontal rate of erosion $\xi^{\rightarrow}$ decreases upstream in an apparently linear fashion, correlating with a similar behaviour in the horizontal component of the ray velocity $v^x$. The vertical rate of erosion $\xi^{\downarrow}$ is constant (to within the precision of the numerical solution), as expected for time-invariant ("steady-state") profiles.

Surface-normal erosion rate is computed in two ways from the ray-tracing results (Figs. 15c and 16c). One way is to simply use the fact (Eq. 18) that normal speed is the reciprocal of normal slowness $\xi^{\perp} = 1/p$. The other is to project the ray velocity

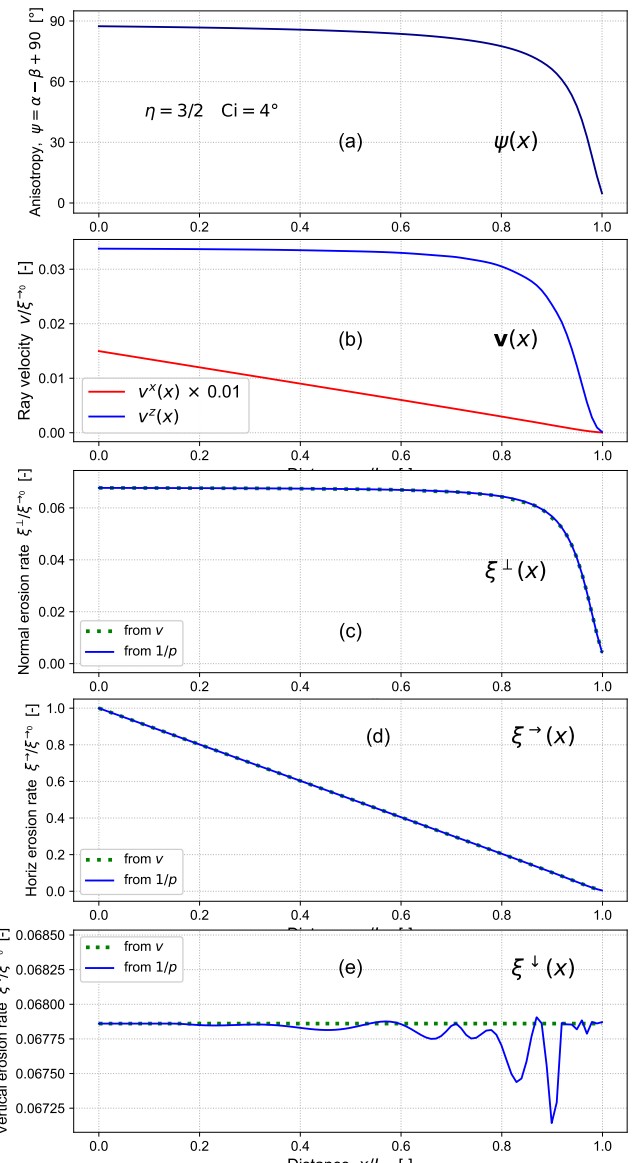

**Figure 15.** Ray and front behaviour along a time invariant profile for $\eta = \frac{3}{2}$ and $\mathrm{Ci} = 4°$: **(a)** anisotropy $\psi$, aka ray-front angular disparity $(\alpha - \beta + 90°)$; **(b)** horizontal (red) and vertical (blue) ray speeds $v^x$ and $v^z$; **(c)** surface-normal erosion rate $\xi^\perp$; **(d)** horizontal erosion rate $\xi^\rightarrow$. **(e)** vertical erosion rate $\xi^\downarrow$. All rates are normalized by the reference horizontal erosion rate $\xi^{\rightarrow 0}$, aka the rate imposed at the boundary $x = 0$.

onto the surface normal (unit) vector using Eq. (66). The horizontal $\xi^\rightarrow$ and vertical $\xi^\downarrow$ erosion rate components are computed with Eqs. (9) and (10) using either of the estimates of $\xi^\perp$. Since ray tracing involves discrete sampling, values of $\xi^\perp$ computed

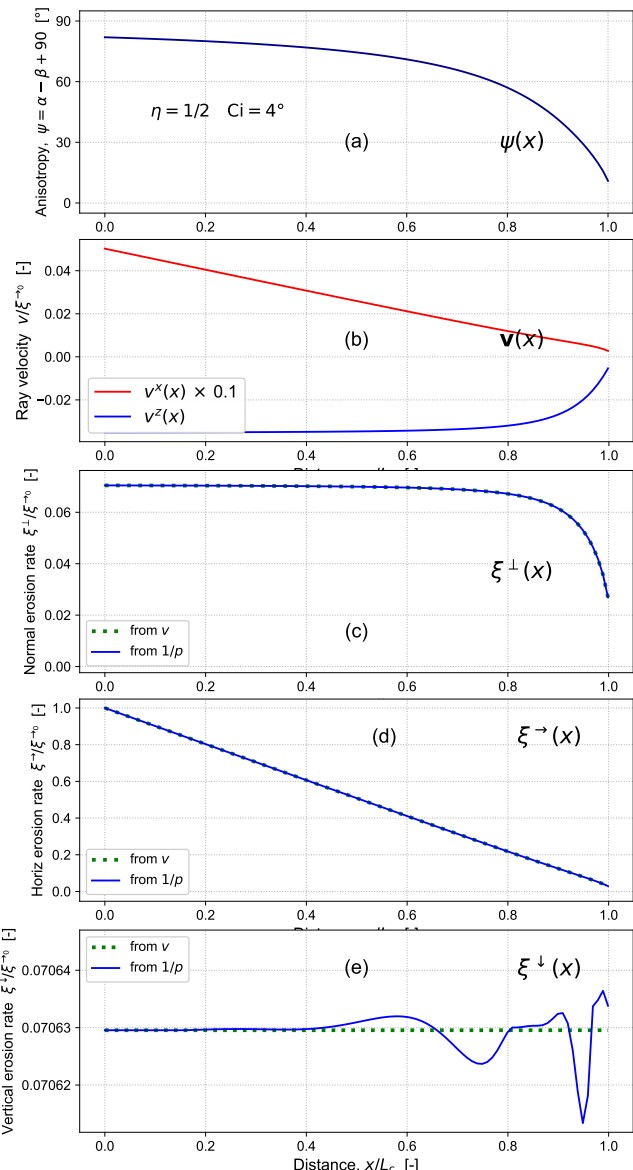

**Figure 16.** Ray and front behaviour along a time invariant profile for $\eta = \frac{1}{2}$ and $\mathrm{Ci} = 4°$: **(a)** anisotropy $\psi$, aka ray-front angular disparity $(\alpha - \beta + 90°)$; **(b)** horizontal (red) and vertical (blue) ray speeds $v^x$ and $v^z$; **(c)** surface-normal erosion rate $\xi^{\perp}$; **(d)** horizontal erosion rate $\xi^{\rightarrow}$. **(e)** vertical erosion rate $\xi^{\downarrow}$. All rates are normalized by the reference horizontal erosion rate $\xi^{\rightarrow_0}$, aka the rate imposed at the boundary $x = 0$.

in these two ways are not numerically identical. The discrete sampling also entails having to generate interpolating functions so that erosion rate values can be calculated at arbitrary positions along the profile.

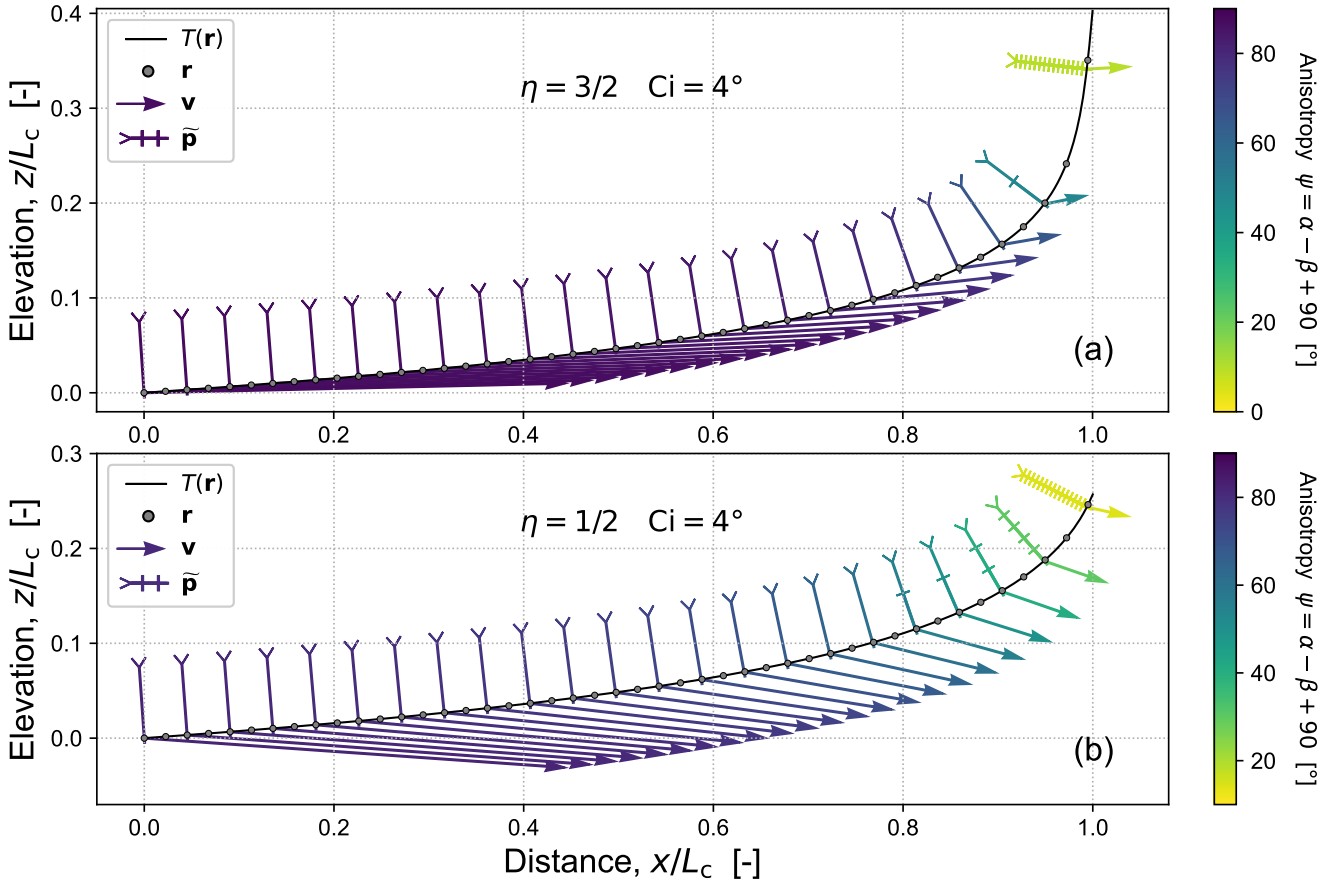

**Figure 17.** Anisotropy of erosion $\psi = \alpha - \beta + 90°$ for time-invariant profiles with $\eta = \frac{3}{2}$ and $\eta = \frac{1}{2}$, and with $\mu/\eta = \frac{1}{2}$ and $\mathrm{Ci} = 4°$. Ray velocity vectors $\boldsymbol{v}$ are represented by arrows (where arrow length provides a rough indication of speed $v$); normal-slowness covectors $\widetilde{\boldsymbol{p}}$ are represented by fishbone symbols (where the number of cross-tick "bones" approximates slowness $p$). The degree of anisotropy is evident both in the divergence of the $\boldsymbol{r}$ and $\widetilde{\boldsymbol{p}}$ directions, and in the colour attribute used to visualize their angular disparity. Anisotropy $\psi$ progressively decreases upstream.

## 6.4 Anisotropy

Figure 17 provides a striking visualization of erosional anisotropy $\psi(x)$ by plotting its variation with $x$ along time-invariant topographic profiles. The direction and magnitude of normal slowness covectors are represented with "fishbone" symbols (where the number of cross-tick "bones" approximates slowness $p$), while arrows represent the ray velocity vectors. The colour attribute of each symbol visualizes the magnitude of the angular disparity $\psi$. The degree of anisotropy is evident in the strong angular disparity of $\boldsymbol{r}$ and $\widetilde{\boldsymbol{p}}$ for the same choices of $\eta \in \left\{ \frac{3}{2}, \frac{1}{2} \right\}$. The strongest anisotropy is found downstream in the channels, where the channel tilt $\beta$ is small, the normal covector points almost vertically downwards, and the ray velocity vector points almost horizontally upstream. Anisotropy decreases monotonically upstream as the normal covector rotates towards horizontal

more rapidly than the ray vector angle. At the divide, the model erosion process is approximately isotropic; this limiting behaviour is moot, however, because the erosion model used here (Eq. 25) does not apply to steep channels.

## 7    Discussion

### 7.1    Geometry controls (almost) everything

The main aspiration of this paper is to clarify what we mean when, in the context of landscape evolution, we speak of the direction of erosion. Our central mathematical tenet has been that while gradient-driven surface erosion takes place in the surface-normal direction (Sect. 1.1; Eq. 3), points on successive erosion surfaces do not necessarily map in the same direction. Working from this premise, and with the help of geometric mechanics, we have found unexpected complexity hidden in simple erosion models.

The concept of a covector is pivotal to our theory (Sect. 3.1). Once we realize that the surface-normal erosion rate (imposed by the gradient-dependent erosion model) can be written in terms of the normal-slowness covector (which is the consequent motion of the surface), it takes only a few short steps to reach the geomorphic surface Hamiltonian. Hamilton's ray tracing equations, the geomorphic surface Lagrangian, and the adherence to Huygens' and Fermat's principles all logically follow.

The essential ingredient of the theory is the realization that, at its core, the process of erosion is a geometric self-constraint. If we disregard complexities such as sediment cover factors and external variations in forcing, a generic model of erosion is a statement about how a surface geometry (through its gradient and flow accumulation) determines the rate of change of that surface geometry. Reparameterizing this statement generates a fundamental function (and thus a Hamiltonian) that describes how to measure distance in the phase space of the erosion equation. The properties of this function reveal that landscape erosion is best described using Finsler geometry. This is important because it provides a fundamental explanation for why geomorphic erosion is anisotropic. As Fig. 17 demonstrates, this anisotropy is very strong.

Counter-intuitively, the erosion rays point (obliquely) upwards if the scaling behaviour of slope in the erosion model has an exponent $\eta > 1$: we might have expected points on an erosion surface to always move downwards since erosion is, after all, driving the surface downwards; this is indeed the case if $\eta < 1$. Remember that the topographic profiles obtained by numerical solution here are time-invariant solutions without an uplift term, as visualized as time slices in Fig. 14; upward motion of rays is therefore driven only by erosion and is not influenced by any tectonic motion.

The idea that surface erosion simultaneously drives two distinct motions—subvertically in the surface-normal direction and subhorizontally in the ray direction—is an uncomfortable and apparently very abstract notion, but it has physical consequences. It means erosion drives information about boundary conditions upstream subhorizontally while also driving motion of the whole profile downwards subvertically. The time scale on which boundary condition information propagates into the interior is the time it takes for a point to travel along a ray (the ray velocity is sometimes known as the signal velocity, which conveys the sense of information propagation well). This information may be the rate of erosion at the stream terminus, or the equivalent slip rate on a boundary fault, or the erosion rate at a point on an initial profile.

For these time-invariant solutions, all rays are identical and the boundary condition does not change. If the boundary erosion rate (or equivalent fault slip rate) were to change (e.g., Reinhardt et al., 2007), we would anticipate the ray paths to change, and we might expect them to intersect (depending on the value of $\eta$): this is one way that knickpoints form. Exploration of this topic is left for another paper (Stark and Stark, 2022). The only ray intersections presented here are those that implicitly take place at the drainage divide, as rays are imagined to approach symmetrically from a right-hand half-domain (Fig. 6). The crucial difference is that intersection at divides occurs when rays approach from opposite directions; knickpoints form where rays move in the same direction at different speeds, one overtaking the other.

Previous studies have considered knickpoint formation as the propagation and intersection of ray-like characteristics (Luke, 1972; Royden and Perron, 2013; Weissel and Seidl, 1998), but always in terms of an explicit surface function and a one-dimensional Hamilton-Jacobi equation describing elevation as a function of distance upstream and time. The parameter space traversed by these characteristics has no concept of the surface-normal or of erosion slowness covectors, which prevents a direct comparison of the results of these studies with those presented here. However, they are broadly in agreement.

Perhaps the oddest outcome of the Hamiltonian theory, but one that is not surprising in retrospect, is that the vertical component of the erosion slowness covector is constant (Sect. 3.14; Eqs. 52–53). To be precise: as a surface point initiates at the boundary and moves along a ray into the interior, its vertical component of surface slowness is invariant, $\dot{p}_z = 0$, and thus the vertical component of the surface erosion rate is constant, $\xi^\downarrow(t) = -1/p_z(t) = -1/p_{z_0} = \xi^{\downarrow 0}$. For a time-invariant (steady-state) profile, all rays are identical, meaning that there is only one ray solution; therefore, the surface at every point along the ray must be moving vertically at the same speed; all rays are independent; therefore, all rays must maintain constancy of the vertical component of the surface erosion rate $1/p_z$. In this sense, a ray carries information of the boundary condition—the vertical slip rate—into the landscape until it is destroyed at a cusp.

The time scale of this information transfer is of crucial importance. If it is small relative to the time scale on which drainage divides move laterally and significantly change accumulation areas and flows, then the assumption made in Sect. 3.5 is valid: namely, that the flow—at every point on the surface where flow influences erosion—can be parameterized by its surface geometry, aka its upstream area, in a manner constant with time (for the lifetime of a ray). This requirement can be weakened to allow for slow variation of the parameterization with time, in which case the Hamiltonian field would need to be recalculated periodically. This is not to say that the geomorphic surface Hamiltonian theory is invalidated if the time scale requirement is not met; rather, the theory would become nonlocal and more complicated. The degree to which such a step is necessary is a topic for future research.

On a side note, bear in mind that the following are all different ways of saying the same thing: (1) the directional pace (reciprocal rate) of erosion-driven surface motion; (2) the surface-normal slowness covector; (3) the gradient of the erosion-front arrival-time function; (4) the directional density of erosion-surface isochrones; (5) the gradient of the geomorphic Hamilton action.

## 7.2 A geomorphic surface Hamiltonian in 3D

While the geomorphic surface Hamiltonian developed here is limited to erosion-driven motion of a linear front in 2D space, the goal is to construct a theory for surface evolution in 3D space. Several conceptual as well as computational hurdles will need to be overcome if this goal is to be met.

The main challenge for a 3D theory will be to find a way to treat channel formation that is consistent with the Hamiltonian methodology. It is tempting to want to resolve the channel shape itself, but this would entail having to add hydrodynamics, sediment transport and abrasion processes to the mix; such a change would not only make the theory inordinately complex, it would run counter to our core premise that what matters is the geometric self-constraint imposed by geomorphic processes, not the details of those processes. A parameterization of flow focusing in channels will be required: one that encapsulates channel cross-sectional geometry without describing it explicitly.

Another challenge hinges on the assumption of locality. A first-cut 3D model can probably be framed with fixed catchment perimeters and static drainage divides; however, there will be a pressing need to generalize and allow for divide motion so that catchment shapes can self-form. The question of time scales raised in Sect. 7.1 will still apply in 3D: if the time scale of divide motion and catchment area change is large relative to the time it takes for erosion rays to traverse the catchment, we will probably be able to treat the flow component of the geomorphic surface Hamiltonian as approximately static, which will make it possible to derive Hamilton's equations for 3D ray tracing.

Numerical solution may require a change of approach, because ray tracing of a surface in 3D is much more cumbersome than for a line in 2D, particularly when dealing with ray intersections and cusp formation. An obvious alternative approach lies in the fact that the theory employs an implicit surface function to describe landscape geometry (Sect. 2): we can resolve erosion front motion on a regular grid and use a level-set method to solve the geomorphic HJE (Adalsteinsson and Sethian, 1995c; Mosaliganti et al., 2013; Sethian and Adalsteinsson, 1997). This will have to be done with some care, however, because of the Finsler nature of the geomorphic surface Hamiltonian and its inherent anisotropy. One-pass fast marching will not be possible, because even the most advanced algorithms for fast marching (Mirebeau, 2014a, 2019; Mirebeau and Portegies, 2019) are currently limited to metrics whose anisotropy is Riemannian (velocity-independent) or Randers (velocity-dependent of a different type to that of the geomorphic Hamiltonian) (Mirebeau, 2014b). A further issue will be the non-convexity of the geomorphic Hamiltonian for certain ranges of $\eta$ and $\beta$ (Appendix C): non-convex Hamiltonians were addressed in the early literature on the level-set method (e.g., Adalsteinsson and Sethian, 1995b, a, 1997) and have been encountered in applications to non-geomorphic erosion (e.g., Radjenović et al., 2006a, b; Radjenović and Radmilović-Radjenović, 2009; Radjenović et al., 2010); recent methodological developments (Chow et al., 2018, 2019; Evans, 2014; Pinezich, 2019) may help. Methods developed in the field of seismology may also prove useful (e.g., Moser, 1991; Qian et al., 2003; Rawlinson et al., 2008; Wang et al., 2006).

,

### 7.3 The variational principle is not energy minimization

There is a substantial body of work founded on the idea that landscapes self-organize in order to minimize energy dissipation across their flow networks. Most of the literature developing this idea—broadly known as optimal channel network (OCN) theory—dates from the 1990s (Ijjasz-Vasquez et al., 1993; Rigon et al., 1993; Rinaldo et al., 1992; Rodriguez-Iturbe et al., 1992a, b; Rinaldo et al., 1998) and is comprehensively reviewed in the book by Rodriguez-Iturbe and Rinaldo (2001). In this section we compare and contrast OCN theory with our theory of the geometric mechanics of erosion.

OCN theory is framed in terms of the self-optimization of a cost function. It identifies this cost function as the total rate of dissipation of mechanical potential energy released by water flowing down channels across the whole landscape. Initial development of the theory focused on the planform geometry and topology of channel networks; it was only later work that addressed the consequent formation of topography. Hillslopes were assumed to play no role. The strong geometric similarity between OCNs and natural stream networks, in particular their similar scaling behaviour, is often presented as a vindication of the theory.

Optimality is a commonly used concept in engineering; its cousin in physics is the notion that system behaviour arises through a variational principle that guarantees minimization of a key quantity. The most fundamental difference is that in engineering the optimization criterion is invoked as a design choice, whereas the variational principle arises as an expression of the underlying physics. It is from this difference that the following criticisms of OCN theory spring.

OCN theory arbitrarily requires minimization of the energy dissipated across the whole channel network; this stipulation is justified on the basis that many physical systems exhibit similar behaviour. Such a requirement implies the existence of a variational principle (Sinclair and Ball, 1996) guiding landscape evolution towards this optimal state, but OCN theory does not articulate this principle in words or mathematics. A corollary issue was the initial omission of a Hamiltonian, which was remedied to some extent in Rinaldo et al. (1998). The weakness of their Hamiltonian is that it cannot be used to derive equations for the time-evolution of the landscape: it constrains what shape the landscape must take, but it cannot explain how that shape comes about.

By framing an alternative theory in terms of geometric mechanics, these issues are avoided. The guiding variational principle is clearly articulated (sect. 3.11): *topographic evolution obeys the principle of least erosion time.* Adherence to this principle is not imposed; it arises geometrically from the way that geomorphic erosion propagates a topographic front and modifies the pattern of erosion rates. The correlative Hamiltonian (Sect. 3.8) generates equations that describe landscape evolution both in the form of Hamilton's equations and in their equivalent form of a Hamilton-Jacobi equation. Solution of these equations, for appropriate boundary conditions, evolves the topography to a time-invariant shape, but this shape is the outcome of geometric interaction rather than a mechanism of energy-dissipation minimization.

Comparison of the two theories is a little premature, because our theory needs further development if we want it to describe the evolution of a whole channel network. The current model also pins the drainage divide at a fixed position: as a result, the degrees of freedom present in a landscape evolving in 3D, notably those that permit different flow topologies and geometries, are absent from our model. It is these degrees of freedom that lead to the existence of many possible states of energy dissipation

aka many possible drainage network configurations; in our 2D theory, only one time-invariant state (a simple linear profile), imposed by the model erosion (Eq. 25), is possible. Nevertheless, it will be interesting to see if a full 3D theory can throw light on what drives landscape self-organization and channel network formation: whether these phenomena arise primarily from the geometric self-constraint imposed by geomorphic erosion, and if so, the extent to which the process of energy minimization is complimentary.

## 8 Conclusions

When we say that the rate of erosion of a geomorphic surface is a function both of its tilt and of the fluxes passing over it, we are in essence saying that the rate of change of landscape geometry is a function of that geometry. Here we have shown how to express this geometric statement as a Hamiltonian, and how to use this Hamiltonian to understand the meaning of the phrase "the direction of landscape erosion".

Our foundational premise is that motion of an erosion surface *intrinsically* acts in the surface-normal direction. On this basis, we can convert a gradient-dependent erosion rate (aka speed) model into a model for the *normal slowness* (aka pace) of surface erosion, parameterized by surface tilt and upstream area and expressed as a covector. Using a simple mathematical trick (a scaling substitution), and by writing tilt in terms of slowness covector components, the model equation can be rearranged into what is called the fundamental function of its metric space; the square of this function is the geomorphic surface Hamiltonian.

This Hamiltonian is parameterized in terms of (i) the position of a single point on the surface, and (ii) the corresponding orientation and slowness of the surface at that point. The Hamiltonian thus occupies a six-dimensional phase space (which reduces to four if the model domain is restricted to a 2D slice). Although such extra dimensionality may seem to be just a mathematical abstraction, it provides real insight.

Study of the Hamiltonian and its phase space reveals that surface evolution simultaneously involves *two distinct* directions of motion: the surface (at a given point) moves in the surface-normal direction, while the point itself moves in what may be an entirely different direction. The disparity between these two directions is a measure of the anisotropy of the process governing motion, and for the class of erosion models studied here such anisotropy is very strong.

This phenomenon is best explored using Hamilton's ray tracing equations—derived from the Hamiltonian by simple differentiation— which express the motion of a surface point and its allied surface-normal slowness in terms of ordinary differential equations (ODEs). They show that while changes in the surface erosion rate and direction are encoded in the normal slowness ODEs, information about boundary conditions and external changes is carried upstream by the ray ODEs.

There is an important dependence of ray tracing on surface tilt: if the model erosion rate faster than linearly with gradient ($\eta > 1$), such rays always have a positive vertical component, i.e., they point upstream and obliquely *upwards*. However, if the model erosion rate scales sublinearly with gradient ($\eta < 1$), erosion rays always have a negative vertical component, i.e., they point upstream but obliquely *downwards* along their trajectories.

We have shown how the phase space occupied by the geomorphic surface Hamiltonian is a metric space, and how this leads us to deduce that the erosion rays traced by surface points are *geodesics*, In other words, they follow paths of locally shortest

erosion time: this is the variational principle that guides geomorphic surface erosion. It appears that energy dissipation need not be invoked, and that instead all that matters is geometry.

*Code availability.* Software to solve and visualize the model has been developed as a pair of platform-independent Python3 packages (built around the SymPy, NumPy and SciPy libraries (Harris et al., 2020; Meurer et al., 2017; Virtanen et al., 2020)) and a set of allied IPython/Jupyter notebooks (Kluyver et al., 2016). The base utilities package is available at the "Geomorphysics Python library" (`GMPLib`) repository on GitHub at Stark (2021a) and archived on Zenodo at Stark (2021d) . The allied code and notebooks to solve the equations presented in this paper, both algebraically and numerically, is available at the "Geometric Mechanics of Erosion" (`GME`) repository on GitHub (Stark, 2021b) and archived on Zenodo at Stark (2021c). `GMPLib` release version 1.0 and `GME` release version 1.0 were used to generate the results presented in this paper.

## Appendix A:  Related studies

### A1    Geoscience applications of the HJE

The HJE has seen only sporadic use in the geosciences—except in the field of seismology, where its static or eikonal form has been found to be particularly useful. The eikonal equation is a good approximation for seismic wave propagation in the so-called "high frequency limit" at which seismic wavelengths are very small compared to the scale of wave propagation (e.g., Červený, 1989, 2005, 2002; Dellinger, 1997; Mensch and Farra, 1999; Rawlinson et al., 2008; Slawinski, 2014; Virieux and Lambaré, 2007; Woodhouse and Deuss, 2007). From this approximation arises the convenient fiction of seismic rays, which are both the characteristics of the HJE and solutions of Hamilton's equations. Although there are disadvantages to its use in treating seismic wave propagation, e.g., dynamic interactions are not modelled and spectral information is lost, the Hamiltonian approach has proven insightful, particularly when dealing with anisotropic media (Antonelli et al., 2003a, b; Bóna and Slawinski, 2002, 2003; Bucataru and Slawinski, 2005; Červený, 2002; Klimeš, 2002; Yajima and Nagahama, 2009; Yajima et al., 2011).

An analogous form of the seismic Hamiltonian approach has been applied to studying the effects of anisotropy on fluid flow in porous media (Sieniutycz, 2000, 2007; Yajima and Nagahama, 2015).

### A2    Applications of the HJE to geomorphology

In geomorphology, Luke (1972, 1974, 1976) pioneered application of the HJE to the modelling of fluvial knickpoints as shocks formed by kinematic waves (Lighthill and Whitham, 1955a, b; Whitham, 1999). Weissel and Seidl (1998) and Royden and Perron (2013) built on this approach to further understand the conditions under which knickpoints form and how they propagate. In all these studies, the HJE was deployed in an explicit-surface form, and its ability to model implicit-surface motion was not considered.

## A3 Use of the eikonal equation in geomorphology

To our knowledge, only one previous study has attempted to model landscape evolution as an implicit surface moving according to an eikonal equation. Aronsson and Lindé (1982) did so in a treatment of weathering-limited denudation of a rock cliff incised at its base by a river. By integrating the eikonal equation representing this erosion process, and by presenting level-set solutions as isochrones of the cliff transect, they demonstrated how variations in rock erodibility can lead to highly irregular surface geometry such as overhangs.

## A4 Non-geomorphic erosion modelled with the HJE

There is a literature on erosion driven by non-geomorphic processes, and much of it is unfamiliar to the geomorphology community. The methods employed in some of these papers provide a partial foundation for our Hamiltonian-based approach. For example, both implicit surface motion and the HJE have been the basis for modelling erosion at microscopic scales in an engineering context.

Frank (1958) employed the concept of surface-motion slowness as a means to model the anisotropic dissolution of crystal surfaces in 2D (although neither the HJE nor the concept of a covector were explicitly invoked). He later extended this approach to handling dissolution in 3D (Frank and Ives, 1960). His technique is widely cited in the crystallography literature (e.g., Frank and Ives, 1960; Ives, 1961; Osher and Merriman, 1997; Shemenski et al., 1965).

In materials science, the Frank method has been adapted to treat surface erosion at the micron scale driven by ion beam bombardment. Early work (Barber et al., 1973; Carter et al., 1971; Nobes et al., 1969) focused on amorphous substrates and isotropic erosion without mentioning the HJE. Subsequent advances introduced the HJE (Carter et al., 1984; Katardjiev, 1989; Katardjiev et al., 1989; Nobes et al., 1987; Smith et al., 1986; Witcomb, 1975) and the eikonal equation (Carter, 2001), and used them to address the issue of anisotropic erosion. Perhaps most relevant to our theoretical development is the review article by Smith et al. (1986), which is also notable for its invocation of an erosional Hamiltonian, and the papers by Carter et al. (1984), Katardjiev (1989) and Katardjiev et al. (1989), which connect the HJE and its Hamiltonian to Huygens' principle and the concept of erosional wavelets (see also: Adalsteinsson and Sethian, 1995a, b, 1997; Sethian and Adalsteinsson, 1997).

## A5 Front motion obeys Huygens' principle

Central to the ideas in the previous sections is Huygens' principle, one of the founding contributions to the field of optics. Using a graphical construction, the principle explains how a wavefront bends as it passes through media of varying resistance to motion (e.g., Arnold, 1989; Holm, 2011; Miller, 1991). At every instant, it pictures the front peppered with tiny wavelets. Each wavelet represents how far, if it were spreading in isolation, a point on the front would expand in the next instant to form its own microfront. Since the points are not isolated, they interfere to form a mutually tangential envelope, with each point moving to the location of its wavelet tangent. The set of successive of tangential envelopes constitutes the progressive motion of the front.

In wave propagation terms, the wavelet represents the unit envelope of group velocity at the point of interest: its shape is called an indicatrix. There is a corresponding structure for phase velocity, known as the figuratrix, which is typically used in its reciprocal speed or slowness form. The velocity indicatrix and slowness figuratrix are linked through mutual conjugacy: as such, they contain the same information about front propagation, but in different forms (Carathéodory, 1999; Perlick, 2000; Rider, 1926; Rund, 1959).

In other words, wavefront propagation can be tracked using either group information or phase information. For front propagation in general this equivalence translates into tracking using either (i) point velocities and their trajectories (ray paths), or (ii) point-wise front-normal slownesses and their ensemble motions.

Huygens' principle is best known for explaining wave propagation in inhomogeneous but isotropic media, where the indicatrices and figuratrices are spherical but vary in size from place to place; isotropy ensures that the group and phase propagation directions are the same. The principle is also often used to explain propagation in media whose anisotropy is symmetric but ellipsoidal (Arnold, 1989), where the group and phase propagation directions are different. Recent efforts have further proved that the principle extends to asymmetric, non-ellipsoidal indicatrices and figuratrices representing a generalized form of anisotropy (e.g., Dehkordi and Saa, 2019; Innami, 1995; Javaloyes et al., 2021; Markvorsen, 2016; Palmer, 2015) expressed in terms of something called Finsler geometry (see Appendices C and D).

## A6    Wildfire spread and Finsler geometry

Several of the ideas discussed in previous sections have seen application in a totally different field—that of wildfire prediction—in the envelope model of fire spread. The earliest form of this 2D model was very simple (Van Wagner, 1969), postulating that wind-driven fire growth can be approximated as a burn ellipse elongated and offset in the wind direction. Anderson et al. (1982) extended the model, and deployed Huygens' principle to propagate a wildfire using elementary burn ellipses scattered along the fire front, each scaled and shaped according to the local fuel availability and wind direction.

These early efforts were purely graphical constructions (Sullivan, 2009). Subsequently, Richards (1990, 1995) formalized the fire front propagation process as a form of the HJE (without explicitly mentioning the equation by name). The model has subsequently evolved, and its most sophisticated version (Markvorsen, 2016) recognizes the elementary burn shapes as non-elliptical velocity indicatrices and frames the anisotropic motion in terms of Finsler structures. Finsler geometry is useful because it provides a convenient mathematical context in which to express the time it takes for a fire front to cover a given distance under the directional influence of wind and terrain. It is for similar reasons that Finsler geometry is important for understanding the anisotropy of geomorphic erosion, as Appendix D shows.

## A7    Ray tracing the motion of a front

The rays of seismology and geometric optics are paths of least time, and they can be traced in two distinct ways: (i) by integrating Hamilton's equations, which are derived from the Hamiltonian contained in the HJE; or (ii) by transforming the Hamiltonian into (or writing directly) the corresponding Lagrangian, converting into the Euler-Lagrange equations, and integrating them (see Appendix E). In both cases, the essential step is to write a Hamiltonian version of the process governing

motion. For simplicity, the derivation presented in this paper is limited to a 2D vertical slice of a landscape. A fully 3D treatment is the subject of ongoing research.

There is a connection between the Hamiltonian ray tracing method developed here and the work of Luke (1972), Royden and Perron (2013), and Weissel and Seidl (1998). These previous approaches deployed the method of characteristics to solve a 1+1D form of HJE in which a 2D topographic profile is represented in an explicit fashion, and their results have some resemblance to those we obtain by full ray tracing (see Sect. 3.12). The main difference is the explicitly 1+1D form of the governing equation in these studies, which forces elevation to be a single-valued function, and which coerces ray tracing into resolving horizontal motion only. If one were to write the Hamiltonian phase space covector coordinate (the direction and reciprocal speed of the surface at a point on the front) for these problems, it would take the reduced form of the slope patch variable of Royden and Perron (2013); this variable contains explicit information about horizontal motion of a surface patch (through its position), but vertical motion is implicit (see Royden and Perron, 2013, Eq. 15). As a result, the inherent anisotropy of the erosion process is hidden.

## Appendix B: Phase spaces and tensors

Slowness covectors and velocity vectors are different mathematical objects, and they live on different spaces, where "space" is meant in the abstract sense used in differential geometry. For each point $r$ in the physical, Euclidean world we can create an allied tangent space that contains all the possible tangent velocity vectors (like $\xi$) at that point; we can also envisage a corresponding cotangent space to contain all the possible slowness covectors (like $\widetilde{p}$) at that point. Bundled together, the tangent spaces for all points in real space constitute a tangent bundle or velocity space, while the union of cotangent spaces forms a cotangent bundle or slowness (classically called "momentum") phase space. These two spaces are indispensable tools of geometric mechanics.

One way to see that vectors and covectors are different is to look at their tensor form: vectors are rank (1,0) contravariant tensors, whereas covectors are rank (0,1) covariant tensors (which is where the "co-" prefix comes from). Tensors of different rank cannot be combined arithmetically; instead, operations such as contraction are needed to combine them. For example, in Eq. (13), the action of covector $\widetilde{p}$ on the unit vector $n$ is a tensor contraction:

$$\widetilde{p}(n) = p_i n^i = \sum_{i \in \{x,z\}} p_i n^i = p_x n^x + p_z n^z \tag{B1}$$

The expression $p_i n^i$ here employs the Einstein summation convention: when an index (such as $i$) is shared by several terms, summation is automatically performed for those terms over all index elements (in this case, over $i \in \{x, z\}$). Upper indexes are used for contravariant tensor components; lower indexes are used for covariant tensor components.

## Appendix C: $\mathcal{F}_*$ is a metric function

The fundamental function $\mathcal{F}_*$ has three key properties that are valid for a domain $D$ of $\{r^x, r^z, p_x, p_x\}$ phase space corresponding to physically reasonable values of surface tilt and erosion rate:

1. Positive, order-1 Euler homogeneity in the parameter $\widetilde{\boldsymbol{p}}$: if the covector $\widetilde{\boldsymbol{p}}$ in $\mathcal{F}_*$ is scaled by a positive scalar $\lambda > 0$, the reparameterized function equals the original function scaled by $\lambda$,

$$\mathcal{F}_*(\boldsymbol{r}, \lambda\widetilde{\boldsymbol{p}}) = \lambda\mathcal{F}_*(\boldsymbol{r}, \widetilde{\boldsymbol{p}}) \qquad \text{for } \lambda > 0 \tag{C1}$$

where the 1 in "order-1" refers to the exponent in $\lambda$ on the right hand side of this equation

2. Regularity: $\mathcal{F}_*$ is smooth, in that it can be differentiated infinitely many times without encountering a discontinuity or undefined value.

3. The Hessian of $\mathcal{F}_*^2$, i.e., the Hessian of the Hamiltonian $\mathcal{H}$, is:

$$g_*^{ij} := \frac{1}{2}\frac{\partial\mathcal{F}_*^2}{\partial p_x \partial p_z} \tag{C2}$$

For $\eta > 1$ and $-p_x/p_z = \tan\beta_c \neq \sqrt{\eta}$, both eigenvalues of $g_*^{ij}$ are real and positive, making $g_*^{ij}$ positive-definite and $\mathcal{F}_*$ strongly convex.

Given these properties, $\mathcal{F}_*$ constitutes a type of Finsler metric (Bao et al., 2000; Shimada and Sabau, 2000). This means that $\mathcal{F}_*$ provides a means of measuring distance and travel time between points in slowness phase space that is dependent on both position and direction of motion (Sect. A7 and Appendix D; see Bao, 2007). In other words, the shortest time path between two points in the corresponding real space may not be a straight line.

Strictly speaking, $\mathcal{F}_*$ is a *co-Finsler* metric on the cotangent space, and thus we are dealing with a co-Finsler or Cartan geometric space (e.g., Miron et al., 2002; Yajima et al., 2011). The term "Finsler" is reserved for the counterpart tangent space and for the fundamental function $\mathcal{F}$, the dual of $\mathcal{F}_*$. Nevertheless, for brevity we use the term "Finsler" to apply to both spaces and metrics.

We also need to be cautious in generalizing about Finsler properties of $\mathcal{F}_*$ and the geomorphic surface Hamiltonian. For $\eta < 1$ and $\beta < \beta_c$, the Hessian of $\mathcal{H}$ is not regular, $g$ is indefinite with mixed signature (the eigenvalues are both positive and negative), $\mathcal{F}_*$ is not convex (Beem, 1971; Červený, 2002; Giaquinta and Hildebrandt, 2004), and the Hamiltonian $\mathcal{H}$ is non-convex. For $\eta > 1$ but $\beta > \beta_c$, the Hamiltonian is similarly non-convex. Under these conditions, it is more appropriate to use the term *pseudo-Finsler* for the metric and its phase space (see Asanov, 1985, pp. 21, 44, 266)

Having a Finsler, or at least pseudo-Finsler, geometry is important for several reasons. The most immediate is the need to adopt a quadratic form of $\mathcal{F}_*$ as a Hamiltonian (Sect. 3.8), because $\mathcal{F}_*$ cannot be Legendre transformed directly (e.g., Červený, 2002; Giaquinta and Hildebrandt, 2004, p. 16). It also means that if we wish to solve erosion front motion as an HJE, we need to find an alternative to the fast marching method, because this algorithm is limited to Riemannian anisotropic metrics (Mirebeau, 2014a, 2019; Mirebeau and Portegies, 2019) and to a small subset of Finsler metrics whose velocity-dependent, Randers type anisotropy (Mirebeau, 2014b) differs from that of the geomorphic Hamiltonian.

## Appendix D: Finsler geometry and curved space

In geomorphology, we are used to dealing with equations that operate in a flat Euclidean geometry where the space is spanned by Cartesian $\{x, y, z\}$ coordinates. In such a space, distances are measured directly using Pythagoras' theorem and the topology of curves across it is straightforward. Working in a flat space like this is fine for studies at the catchment scale and is a good approximation even at the orogen scale.

    There are scales, however, where use of such a flat space is inadequate. For example, what if we are interested in processes
on a global scale, and need to account for the spherical geometry of Earth's surface? Switching to spheroidal coordinates is only half the battle, because transport on a sphere is topologically different to that on a flat space: particles in locally straight motion follow looping paths; these paths are great circles; sets of great circles always converge and intersect; and so on. In this example, we need to understand the consequences of working in a curved space and its consequences if we want to understand physical phenomena acting at such scales.

The concept of curved spaces is relevant not just to processes on objects with topological curvature; in an abstract way, it can also apply to the space in which the governing equations operate. For some types of process, the governing equations can be mapped from Euclidean space into a non-flat phase space that both simplifies their solution and exposes their fundamental properties and behaviour.

    The geomorphic surface Hamiltonian $\mathcal{H}$, which arises from the transformation of an erosion equation, operates in such
a non-flat space. Distance and travel time are not Euclidean measures on this phase space, because the fundamental metric function $\mathcal{F}_*$ that defines $\mathcal{H}$ has the properties described in Sect. 3.7.

    The curved nature of non-Euclidean spaces lies in how distance is measured on them. The measurement of distance always requires a yardstick of some kind (on a metric space, this is a tensor derived from $\mathcal{F}$ or $\mathcal{F}_*$, and an associated inner product), but the yardstick used in Finsler geometry is not the equivalent of a simple ruler. It is not an isotropic constant as it would be in
a flat Euclidean space (where the metric tensor is a simple Kronecker delta). Nor is it an anisotropic, possibly spatially variable, but otherwise static quantity as it would be in a curved Riemannian space (with a metric tensor whose variable elements are a function of position $r$ alone). Instead, the yardstick is a function both of position and of the direction and magnitude of motion at that position, i.e., for $\mathcal{F}_*$, the metric tensor elements vary with both $r$ and $\widetilde{p}$, not just with $r$. Instead of measuring distance with a single inner product at each point, there is a family of inner products associated with each point (e.g., Shen, 2001).

This directional dependence of the "yardstick" or metric tensor is the defining characteristic of Finsler geometry (Bao et al., 2000; Chern, 1996; Holm, 2011; Shimada and Sabau, 2000). To be precise, $\mathcal{F}_*$ specifies that the slowness phase space is a co-Finsler or Cartan space, and its dual $\mathcal{F}$ specifies that the velocity space is a Finsler space. The practical consequence is that the time taken to travel an infinitesimal distance across the space (Bao, 2007) at unit speed in a given direction is a function of that travel direction. Adding up such incremental times allows us to find the shortest path across the space, but the directional
dependence of erosion time measurement makes this calculation non-trivial.

The fact that the geomorphic surface Hamiltonian operates in a Finsler geometry has profound consequences for the construction of erosion-driven equations of motion, for the variational principle that underlies how landscape shape evolves, and for the concept of erosional anisotropy. These consequences are explored in the next sections.

For further information on Finsler geometry, a good introduction is the non-technical discussion in Gibbons and Warnick (2011), which also touches on several other topics important to this paper. A more mathematical but surprisingly approachable introduction can be found in Bao (2007), while more comprehensive treatments are provided by Antonelli et al. (1993); Antonelli (2000); Bao et al. (2000); Chern (1996); Giaquinta and Hildebrandt (2004); Mo (2006); Miron et al. (2002); Shen (2001, 2013); Shimada and Sabau (2000).

## Appendix E: Lagrangian and geodesics

The Lagrangian method of ray tracing the motion of an erosion front is taken up in detail in Stark et al. (2022). This alternate approach is important, because it demonstrates in practical terms how erosion rays are also *geodesics*, i.e., that they are solutions of the geodesic equation (Misner et al., 1973; Nolte, 2019) corresponding to the geomorphic surface Hamiltonian. Although geodesics have cropped up before in geomorphology as a means of delineating drainage on DEMs (Passalacqua et al., 2010a, b), their use in that context was a pragmatic means to an end, rather than a reflection of any underlying physics. In the our theory, however, the geodesic equation has physical meaning in that it is synonymous with the Euler-Lagrange equation of the geomorphic HJE; solutions to the geodesic equation follow the same paths of least time as solutions to Hamilton's ray tracing equations derived from the geomorphic surface Hamiltonian. This assertion is proved in Stark et al. (2022).

## Appendix F: HJE and Hamilton action

Ray tracing through integration of Hamilton's equations is not the only way to solve for surface motion. In principle, we could instead use the geomorphic surface Hamiltonian $\mathcal{H}(\boldsymbol{r}, \widetilde{\boldsymbol{p}})$ in its HJE form and solve erosion front propagation using grid-based methods. In practice, numerical solution of this kind of eikonal equation is not straightforward (see Sect. 7.2 and Appendix C). The HJE is nevertheless instructive if we examine it in the context of some important concepts of classical mechanics. For example, Hamilton's principal function $S(\boldsymbol{r}, t)$, which is the Hamilton action $S_\gamma$ (see Eq. 38) plus a constant, is

$$S = \int \mathcal{L}\, \mathrm{d}t \quad \Leftrightarrow \quad \frac{\mathrm{d}S}{\mathrm{d}t} = \mathcal{L} \tag{F1}$$

Use of the Legendre transform (Eq. 37) yields

$$\frac{\mathrm{d}S}{\mathrm{d}t} = p_i v^i - \mathcal{H} \tag{F2}$$

The total derivative of $S(\boldsymbol{r}, t)$ with respect to time $t$ has

$$\frac{\mathrm{d}S}{\mathrm{d}t} = \frac{\partial S}{\partial r^i}\frac{\partial r^i}{\partial t} + \frac{\partial S}{\partial t} \tag{F3}$$

Assuming the points $\{r\}$ all lie on a path $\gamma_0$ of least erosion time, we can write

$$\frac{\mathrm{d}S}{\mathrm{d}t} = \frac{\partial S}{\partial r^i} v^i + \frac{\partial S}{\partial t} \tag{F4}$$

Comparing this equation with Eq. (F2) leads to

$$p_i = \frac{\partial S}{\partial r^i} \quad , \quad -\mathcal{H} = \frac{\partial S}{\partial t} \tag{F5}$$

such that the Hamiltonian $\mathcal{H}(r, \widetilde{p})$ can be written as

$$\mathcal{H}\left(r, \frac{\partial S}{\partial r}\right) = -\frac{\partial S}{\partial t} \tag{F6}$$

which is the standard form for the HJE. Now consider the arrival time function $T(r)$, whose total time derivative is, given Eqs. (22), (55), and $\partial T / \partial t = 0$:

$$\frac{\mathrm{d}T}{\mathrm{d}t} = \frac{\partial T}{\partial r^i} \frac{\partial r^i}{\partial t} = \frac{\partial T}{\partial r^i} v^i = p_i v^i = 1 \tag{F7}$$

Integration here gives the *abbreviated action*; by choosing to integrate along a path of least action $\gamma_0$ we obtain the shortest erosion time $T(r)$:

$$\int p_i v^i \, \mathrm{d}t = \int p_i \, \mathrm{d}r^i = \int \mathrm{d}t = T(r) \tag{F8}$$

Use of Eq. (34) and Eq. (F2) connects $S(r,t)$ with $T(r)$, $\mathcal{H}(r, \widetilde{p})$ and time $t$:

$$S = T - \mathcal{H}t = T - \tfrac{1}{2}t \tag{F9}$$

Differentiation gives

$$\frac{\partial S}{\partial r} = \frac{\partial T}{\partial r} = \nabla T \quad , \quad \frac{\partial S}{\partial t} = -\frac{1}{2} \tag{F10}$$

Substitution into the standard HJE in Eq. (F6) leads to

$$\mathcal{H}(r, \nabla T) = \frac{1}{2} \tag{F11}$$

In this form, the HJE prescribes how the erosion front $T(r)$ has a locus (a set of points $\{r\}$) that propagates such that the gradient $\nabla T$ (the directional density of $T$ isochrones) satisfies the static Hamiltonian $\mathcal{H} = \frac{1}{2}$.

**Table A1.** Notation

| | |
|---|---|
| $t$ | time |
| $L_\mathrm{c}$ | distance from stream terminus to the drainage divide |
| $x$ | horizontal coordinate $0 \leq x \leq L_\mathrm{c}$ measured from stream terminus |
| $y$ | out-of-section horizontal coordinate |
| $z$ | vertical coordinate: distance above the terminus |
| $\{\boldsymbol{a}\}, \{\boldsymbol{b}\}$ | sets of points defining successive erosion front surfaces |
| $T_a, T_b$ | corresponding loci of erosion surfaces at successive times |
| $T(\boldsymbol{r})$ | isochrone of erosion surface at point $\boldsymbol{r} \Leftrightarrow$ surface locus at $T = t$ |
| $\boldsymbol{r}$ | point vector on erosion front surface, i.e., point on erosion ray |
| $\boldsymbol{v} = \dot{\boldsymbol{r}}$ | tangent velocity vector of point moving along erosion ray |
| $\widetilde{\boldsymbol{p}}$ | covector of normal slowness of erosion front |
| $r^x, r^z$ | horizontal, vertical components of ray point vector $\boldsymbol{r}$ |
| $r^{x0}, r^{z0}$ | boundary values of components of $\boldsymbol{r}$ |
| $v^x, v^z$ | horizontal, vertical components of tangent ray velocity vector $\boldsymbol{v}$ |
| $p_x, p_z$ | horizontal, vertical components of $\widetilde{\boldsymbol{p}}$ |
| $p = |\widetilde{\boldsymbol{p}}|$ | surface normal slowness aka reciprocal erosion rate |
| $p_{x_0}, p_{z_0}$ | boundary values of components of $\widetilde{\boldsymbol{p}}$ |
| $\alpha$ | ray angle, aka angle of $\boldsymbol{v}$ from horizontal |
| $\alpha_\mathrm{ext}$ | limit ray angle (maximum for $\eta < 1$, minimum for $\eta > 1$) |
| $\beta$ | angle of $\widetilde{\boldsymbol{p}}$ from vertical; also surface slope angle from horizontal |
| $\beta_c$ | critical surface slope angle |
| $\beta_0$ | boundary value of surface slope angle |
| $\mathcal{F}(\boldsymbol{r}, \boldsymbol{v})$ | 1-homogeneous Finsler fundamental function |
| $\mathcal{F}_*(\boldsymbol{r}, \widetilde{\boldsymbol{p}})$ | 1-homogeneous co-Finsler (Cartan) fundamental function |
| $\mathcal{H}(\boldsymbol{r}, \widetilde{\boldsymbol{p}})$ | 2-homogeneous Hamiltonian |
| $\psi$ | erosional anisotropy $= \alpha - \beta + 90°$ |
| $h(x)$ | elevation as a 1+1D function of horizontal distance upstream |
| $\phi(x, y, z; t)$ | level-set function |
| $\boldsymbol{\xi}$ | erosion velocity vector; generic velocity function in level-set equation |
| $\xi^x, \xi^z$ | horizontal, vertical components of erosion velocity vector $\boldsymbol{\xi}$ |
| $\xi^\perp$ | surface-normal erosion rate (speed) |
| $\xi^\rightarrow$ | horizontal (positive right) erosion rate |
| $\xi^\downarrow$ | vertical (positive down) erosion rate |
| $\xi^{\downarrow 0}$ | boundary value of vertical (positive down) erosion rate |
| $\boldsymbol{n}$ | surface-normal unit vector |
| $n^x, n^z$ | horizontal, vertical components of surface-normal unit vector $\boldsymbol{n}$ |
| $\eta$ | gradient-scaling exponent in surface-normal erosion model |
| $\mu$ | upstream area-scaling exponent in surface-normal erosion model |
| $\lambda$ | a real scalar |
| $\mathcal{L}(\boldsymbol{r}, \boldsymbol{v})$ | 2-homogeneous Lagrangian |
| $i, j \in \{x, z\}$ | covariant (lower) or contravariant (upper) indices |
| $\gamma(t)$ | potential erosion ray path (parameterized by time $t$ |
| $\gamma_0(t)$ | erosion ray path of least time |
| $S_\gamma$ | action functional for erosion ray paths |
| $S_{\gamma_0}$ | least action $\Leftrightarrow$ (half) least erosion time |
| $n$ | gradient-scaling exponent in vertical erosion model (SPIM) |
| $m$ | upstream area-scaling exponent in vertical erosion model (SPIM) |
| $\varphi(x)$ | spatial component of rate of erosion at distance $x$ upstream |
| $\varphi_0$ | base rate in flow component of erosion model |
| $\varepsilon$ | relative flow component rate at zero upstream area |
| $t^{\rightarrow 0}$ | horizontal erosion time scale for (computed from boundary rates) |
| $\hat{t}, \hat{x}, \hat{z}$ | non-dimensionalized coordinates |
| $\hat{\boldsymbol{r}}, \hat{r}^x, \hat{r}^z$ | non-dimensionalized position variables |
| $\hat{\boldsymbol{p}}, \hat{p}_x, \hat{p}_z$ | non-dimensionalized slowness variables |
| Ci | dimensionless boundary erosion rate |
| $k$ | ray index |
| $t^\rightarrow_{L_\mathrm{c}}$ | time scale for erosion to traverse the domain |
| $h_{L_\mathrm{c}}$ | height scale of time-invariant topographic profile |
| $g_*^{ij}$ | co-metric tensor |
| $S$ | Hamilton's principal function |

*Author contributions.* CPS led the research, wrote the code, and co-wrote the paper. GJS collaborated on the research and in writing the paper.

*Competing interests.* None

*Acknowledgements.* Discussions with Petrus dos Anjos, Liam Reinhardt and Mike Ellis are greatly appreciated.

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
