# Peer review of "The Direction of Landscape Erosion"

_Earth Surface Dynamics, 2021_

## Referee Comment (RC2)

Review of:

**The Direction of Erosion**

Colin P. Stark and Gavin J. Stark

**General Comments**

I am supportive of this interesting paper. It definitely should be added to the conversation regarding how we conceptualize and formally describe the motions of eroding surfaces/lines. Please know that, whereas I am mostly familiar with the concepts and some of the mathematics presented, parts of the presentation are unfamiliar to me. I have attempted to work through the logical progressions to my satisfaction at numerous points in the text (there are many moving parts in the analyses), but I must acknowledge that I have not carefully checked all elements of the work. That said, my experience suggests that the authors are careful in their work, and things seem to be in order.

The authors are asking the readers to absorb the essentials of level sets, the idea of a Hamiltonian, Hugyens's principle, geodesics, etc. Those who have a background in these topics might be fine with the presentation. But I suspect some of this material will be quite unfamiliar to many *ESD* readers. This is a deep read, and it likely will require more than one sitting for many readers (as it did in my case). For this reason, my recommendations starting with the next paragraph are aimed at helping make the material more accessible to a broader audience — assuming this is the intention rather than just being aimed at a restricted group of readers.

The Abstract reasonably describes the key elements of the paper, whereas the information content of the Introduction (Section 1) is sparse. Following this, I suspect that some if not many will stop reading, with eyes glazed over, somewhere within Sections 2.4 through 2.14. The material in these sections comes fast, and although the authors attempt to make connections with descriptions of Earth surfaces/lines, my reading suggests that these sections risk confusing readers without offering a clear idea of why it is important for readers to grasp these relatively unfamiliar concepts and techniques. As one who repeatedly struggles with the question of how to best present (oftentimes) unfamiliar mathematical material in papers, I suggest the following possibility.

Offer an example (or examples), with clear diagram(s), right up front in the Introduction. Show the elements of what the analysis is describing about the motion of a surface/line and what the motion implies. Describe what is happening with reference to qualitative descriptions of the level set(s), normal motion, asymmetry, etc. Such diagrams could be simplified versions of material contained in diagrams that appear later. The relatively familiar problem presented in Figure 8 might be a good candidate, notable given that the rays computed from

the Hamiltonian description are directly compared and explained relative to what is obtained from the conventional approach (Section 4.3). Then — and importantly — state the implications of the analysis relative to what is normally envisioned/modeled, including what the analysis reveals that otherwise is not accessible from conventional analyses, including key material selected from the Discussion section. In other words, explain at the outset what the specific merits of a mostly unfamiliar style of analysis are. I suspect that such a "preview" might well help readers with the subsequent primer that unfolds the technical material. I further suggest that this sort of introductory material deserves to be reasonably thorough if it is to be effective. That said, please know that I will not be offended if the authors prefer to reject these recommendations, as I do not imagine that my role as a reviewer involves telling authors what they ought to present (and how) in lieu of the presentation they wish to make. (Note that I recently added a section of this sort, for similar reasons, to one of my own papers in response a reviewer suggestion. I think it helped a lot.)

The authors are clear about the idea that the "rate of change of surface geometry is solely a function of surface geometry... [thus imposing a] geometric self-constraint" that leads to the Hamiltonian description (Section 3.4 and elsewhere). They then use the familiar erosion model embodied in Eq. (24) to highlight the description and its implications. Meanwhile, the science is moving beyond this simplistic formulation of erosion, except perhaps as a rough indication of large scale landscape behavior. I therefore suggest that the value of this example mostly resides in its familiarity, whereas I would like to imagine that the analyses in the paper are aimed more generally at providing a different perspective on describing motions of eroding surfaces/lines. That said, I also am thinking that the class of such formulations of erosion (satisfying the geometric self-constraint condition) is a small set. Partly for these reasons, I pose a problem concerning river meandering at the end of this review.

**Specific Comments**

Line 34: Is there a compelling reason to "unify" them if they are described with respect to the physics involved? Unified in what sense? There may be merit in pulling key items offered in the discussion (Section 6.1) to clarify what is intended.

Lines 64 and 249, and in reference to Line 515: To "drive" evolution implies physics, whereas a "least time" argument is a geometrical outcome of the variational analysis, as elaborated in the vicinity of Line 950. (The attention given in Section 6.3 to the issue of uncritically appealing to an energy interpretation is appreciated, as summarized in the Abstract.)

Line 70: $h(x, y, t)$

Line 75: Such problems are numerical rather than mathematical, however? The overhang idea, of course, is a restriction.

Lines 108–109: Changes in attributes(?) or effects(?) of vegetation and precipitation are easy to incorporate? Hmm...

Lines 850–855: As the authors describe here, in the Abstract, and elsewhere, the idea of "geometric self-constraint" suggests that the analysis is in fact restricted to those situations in which the formulation of erosion specifically satisfies this constraint, setting aside the added complexities of what the authors are calling "non-locality" with changes in contributing area/length. Given that this is the core premise of the work (Line 906), it probably merits description in the introductory example(s) described above (if the authors decide to offer such a "preview" at the outset). Regarding the final sentences: "...this is the variational principle that governs geomorphic surface erosion. It appears that energy dissipation need not be invoked, and that instead all that matters is geometry," the word 'governs' is strong. After all, physics does the erosion whereas the described outcome arises from the geometric self-constraint — which is only presumed to adequately represent the physics.
* * *
Now I'm going to gently throw a wrench into the works, not because I want the wrench to stop the spinning of the machinery, but rather, because I am curious to see whether the machinery absorbs the wrench and continues to whir along.

The paper focuses on examples in which the surfaces are embedded within a Cartesian coordinate system, where ray trajectories are well defined starting from a set of points on the initial surface/line. Now consider, instead, the problem of a freely meandering river over long time scales. It is now conventional to choose a curvilinear coordinate system in which the primary (intrinsic) coordinate coincides with the sinuous channel centerline. The principle reasons for this choice are to ensure that local channel attributes (direction, curvature, etc.) are single-valued functions for high-amplitude bends, and because the equations of fluid motion can be readily, naturally adapted to this system. In turn, the local rate of channel (centerline) migration is taken to be normal to the centerline. Typically this local rate is expressed as a convolution of upstream (and sometimes downstream) channel geometrical states (i.e., curvature), thus satisfying the geometric self-constraint condition. However, this description of the system means that, with reference to a suitable origin or to two selected initial points on the centerline, the arc length of the deforming coordinate system continuously changes such that the centerline coordinate distance of a local position defined at time $t$ changes at time $t + \mathrm{d}t$, whether the "point" at this position remains fixed, or changes, with respect to a global Cartesian system.

So... setting aside the severe complication arising from meander cutoffs, how might one approach this problem using the variational methods/techniques? Is this even possible? Might it reveal information that we otherwise would not discern from conventional analyses (including numerical simulations of meandering)?

Please know that these questions merely reflect my curiosity. I am not suggesting that they need to be addressed in the paper. Rather, it would be interesting to hear what the authors might have to say about this problem.

djf
01 November 2021

---

## Author Comment (AC1)

**Author comments in response to reviews of esurf-2021-59 "The Direction of Erosion" renamed "The Direction of Landscape Erosion"**

Colin P. Stark & Gavin J. Stark

January 4, 2022

We are really grateful to the referees for their comments and criticisms, both positive and negative, even those with which we disagree. The criticisms have highlighted important shortcomings in our submitted manuscript—in particular, failures to communicate effectively—and they have spurred us to make major revisions. We hope these changes make the paper easier to follow and its conclusions more compelling.

The changes include new text and figures to better explain the motivation for tackling erosion using geometric mechanics, and a substantial reorganization (notably the addition of a non-technical introductory subsection and a shift of non-essential background material into appendices) to bring readers to the main results more quickly. The length of the text remains roughly the same, but the body of the paper has grown slightly (by 3 pages) after adding several new figures and modifying others. To alleviate the concerns of referee #1, we have added new material both on the topic of erosional anisotropy, and on the real-world scales implied by our model; ironically, this drove the need for a new section on non-dimensionalization.

These changes and more are discussed in the "Revisions" section below; first, we address the comments made by the referees.

**Response to Referee #1 (Anonymous)**

The most serious criticisms raised by referee #1 lie in her/his section "Overall impression":

**RR1.1**
**. . . the motivation. . . remain(s) unclear**

We agree, in that we could have explained our basic rationale earlier and more directly. To meet this criticism, we have expanded the introduction (new subsection 1.1 and new figure 1) with a relatively non-technical explanation of the mathematical ideas underpinning our work.

To avoid any further doubt: **the goal of our paper is to understand the direction(s) in which erosion takes place in an evolving landscape**, something we regard as fundamental to the field of geomorphology. We use the tools of geometric mechanics, because they help expose some of the geometric complexity of an apparently simple phenomenon.

The new material in section 1.1 expands on our original introductory remarks, and builds on the referee's comments in their section "About the findings of the paper", paragraph 3, discussing the pairing of points between successive erosion surfaces. The referee and we agree that such a pairing equates to establishing an erosion model, but in the expanded introduction we make the important distinction that this point-pairing $\leftrightarrow$ erosion-model equivalence is only true if the model is **gradient dependent**. And despite finding common ground with the referee on this fundamental geometric premise, we

disagree with two subsequent statements in this part of their review.

First, it is not correct that *"most models in the literature use the* local normal to establish that bijection" (aka: in defining the erosion model). To the best of our knowledge, no published landscape evolution models (LEMs) define erosion as a surface-normal rate; they all write their erosion models as a process acting vertically downwards (we cite multiple reviews of the LEM literature). This is the erosion context to which our work applies. Perhaps the referee is thinking instead of studies of boulder and cobble shape evolution, which they cite at the end of their review? Our work is tailored to erosion processes at the channel and reach scale, and does not apply to erosion of objects at such small (sub-channel) scales and within a flow. We have modified the text to alleviate such possible confusion, emphasizing at several points in the manuscript that our theory applies to catchment evolution. To hammer this point home, we have decided to change the title of the paper to "The Direction of Landscape Erosion."

Second, regarding the comment that *"the sentence in the conclusion... is false and misleading"*, perhaps we could have phrased this sentence better: we maintain that the *intrinsic* direction of erosion-driven motion of a surface—as a 2D geometric entity in 3D space, as opposed to the set of 0D points that define the surface—is in the surface-normal direction: invocation of any other direction entails the provision of supplementary information, such as the direction in which gravity acts. The new introductory subsection 1.1 takes great care to explain this issue and its importance in establishing the dual directions in which erosion acts on a surface. The sentence criticized by the referee has been correspondingly modified to say *"Our foundational premise is that motion of an erosion surface intrinsically acts in the surface-normal direction."*

**RR1.2**

**. . . the significance of the work remain(s) unclear**

There is a studied ambiguity to this comment. Does the referee mean we have not explained the significance of our work sufficiently clearly? Or does she/he mean that our work is insignificant?

If the latter meaning is intended (triviality), this is a value judgement, and we choose to disagree: our paper shows how to transform a classic erosion model into a geomorphic surface Hamiltonian, uses this Hamiltonian to reveal a fundamental anisotropy to the process of surface erosion, quantifies the speed and direction in which boundary change information propagates across a landscape (as a function of gradient scaling), and demonstrates that the variational principle

driving landscape formation is that of least erosion time rather than energy dissipation. All these insights are arguably of fundamental importance to geomorphology. 84 86

If the former meaning is intended (a lack of clarity), we agree to some extent, and we have substantially revised the manuscript to make its significance more clear. These changes are summarized in the section "Revisions" below. 88 90

**RR1.3**

**The erosional model is not new. . .**

This is not entirely true, but insofar as it is true, it is intentional: see our more detailed response RR1.5 below. 92

94

**RR1.4**

**. . . the results seem to be consequences of the chosen solution technique without any physical significance.**

This is fundamentally incorrect.

If a physical phenomenon can be written in Hamiltonian form, the properties of said Hamiltonian, such as the direction in which the system state evolves according to Hamilton's equations, have physical meaning. They are not simply *"consequences of the chosen solution technique"*. 96 98 100

In our case, Hamilton's equations give us two ODEs defining the components of the velocity vector of surface points, $\mathbf{v} = \dot{\mathbf{r}} = \mathrm{d}\mathbf{r}/\mathrm{d}t$, and another two defining the how surface erosion rate (in the form of a slowness covector) changes, $\mathrm{d}\widetilde{\mathbf{p}}/\mathrm{d}t$. The latter has obvious physical meaning; the former, aka the ray velocity, gives the direction and speed at which changes in external conditions imposed on the landscape propagate across it. 102 104 106

In our revision, we have boosted emphasis of this point to help readers avoid the misapprehension shown here—see, for example: the abstract, new section 1.1, now section 3.14, now section 6 first paragraph, new section 6.1, now discussion section 7.1, etc. 108 110

Having addressed the most serious criticisms, we now respond to each of the remaining comments in order.

**RR1.5**

**The application of the Hamiltonian machinery is elegant, however, it strongly relies on the erosion model introduced in Eq.(24), which is identical with the model in Royden and Perron (2013)**

There are three misapprehensions in this comment.

First, our erosion model is *not* identical to the standard "Stream-Power Incision Model" (SPIM, per Lague 2014, more colloquially known as the "stream-power model"), used by Royden and Perron and many, many others (see the literature reviews cited in new section 2.1 of Coulthard, 2001; Pazzaglia, 2003; Tucker and Hancock, 2010; Tucker, 2015; van der Beek, 2013; Willgoose, 2005). Consider equation 73 (revised manuscript), which defines the surface-normal speed of erosion as

$$\xi^{\perp} \;:=\; \varphi_0 \left(L_\mathrm{c} - x\right)^{2\mu} \left(\sin\beta\right)^{\eta}$$

This is not simply an "un"-projection of the vertical rate of erosion given by the SPIM back into a surface-normal rate, although the two models would approximately coincide for small angle $\beta$; the pure SPIM version would be written

$$\xi^{\downarrow} \;:=\; \varphi_0 \left(L_\mathrm{c} - x\right)^{2m} \left(\tan\beta\right)^{n}$$

Our model and SPIM only have *identical* form for linear gradient scaling, $\eta = n = 1$.

Second, and most important, it is not at all our intention, nor would it be a good idea, to introduce a radically new erosion model here. Our goal is to show that the classic (SPIM) way of treating erosion as a separable scaling function can be converted, after only slight modification, into a Hamiltonian form, and then to use this form to reveal properties of geomorphic surface erosion that have long been hidden. We are a bit mystified as to why the referee puts so much store in the introduction of a *new* erosion model, since it would only add unnecessary complexity to our story.

Third, we expect Hamiltonian transformation may be possible for a broader class of erosion models than just the SPIM-like model adopted here. We have not sought to figure out *how* broadly in this paper: that's a task for future work. In any case, the insights to be had from a Hamiltonian based on the simplistic "stream-power model" approach likely remain valid, with some caveats, for more physically realistic erosion models (and for the process of erosion in reality). For example: our conclusion that geomorphic surface erosion is strongly anisotropic, with boundary change information and

surface motion acting in radically different directions, almost certainly transcends the specifics of the Hamiltonian, or even whether a geomorphic Hamiltonian can be written at all. ₁₅₀

₁₅₂

**RR1.6**

**The physical background and interpretation of this formula are missing**

This criticism is fair, and we have made significant changes to the text to fix the problem, as well as to fix related issues to do with ₁₅₄ exposition of our chosen erosion model. We have split the theory section into two parts, and have moved the (old) section 3.20 "For- ₁₅₆ mulation of erosion model" dealing with the specific formulation of an erosion model into the (new) section 4.1 "A modified stream- ₁₅₈ power incision model". We have also improved the explanation here. In addition, we have rewritten section 3.5 (old "A specific form for ₁₆₀ the erosion equation"; new "Separable, gradient-dependent erosion rate model") for greater clarity. ₁₆₂

In passing, we should mention that formula referred to here (in which the model flow component is written as a power function of down- ₁₆₄ stream distance as a proxy for upstream area) has a long history in the literature, including in several of the papers we cite (e.g., Luke, ₁₆₆ Weissel & Seidl, Royden & Perron). We see no need to cover it in our paper. ₁₆₈

**RR1.7**

**Back to Eq. (24): why not assume the surface normal speed in the form $(x)f(sin)$, where $f : [-1, 1]$, a suitably regular function?**

Because we want to build on the long-established SPIM as a reason- ₁₇₀ able scaling approximation of erosion at the reach scale (see above RR1.3). ₁₇₂

**RR1.8**

**Figure (5) shows an example of the ray tracing approach, but also introduces a physically different problem: fault slipping, which was not mentioned before in the text. It is unclear why it is chosen from the possible physical problems listed in the Introduction.**

This is a very good point; ironically, it's a point that the first author ₁₇₄ himself has made in conversation with colleagues over the years: it is entirely unnecessary to invoke any tectonic motion or fault slip ₁₇₆ mechanisms when modeling the evolution of a steady-state channel profile—all that's needed is to have a velocity boundary condition ₁₇₈ and to track motion of the profile in a frame moving with that velocity. Invocation of normal fault slip and uniform uplift is invariably ₁₈₀ made because it puts the erosion model into a real-world geological context that's familiar to everyone in our field. ₁₈₂

So, in the revised manuscript we have made a compromise: we have deleted most of the comments about fault slip etc, and have replaced ₁₈₄

them with mentions of a velocity boundary condition and reference frame. This change is most noticeable in (now) figure 6 (was fig. 5) which illustrates how ray tracing is used to construct a time-invariant ("steady-state") profile. Some links to the tectonic context remain, notably in (now) sections 5.1 and 7.1.

**RR1.9**

**The abstract suggests that one of the conclusions of the work is that "erosion takes the path of least erosion time" but later in Sect. 2.14, the concept is introduced as a proposal. It is unclear whether it was an assumption or an outcome of the model.**

Fair point: we should have clearly distinguished our early scene-setting remarks from presentation of particular outcomes. To be clear: we do not *assume* that erosion takes the path of least time, we *infer* this, on discovering how to convert the erosion model into a (pseudo-)metric and a fundamental function, mapping it into a parametric Hamiltonian, and realizing that Huygens' and Fermat's principles apply.

The core problem with the original manuscript was the weighty background (old) section 2, which was intended to provide as much context and references to related literature as possible, but in so doing created confusion like this. In the revised manuscript, (old) sections 2.5–2.13 have either been moved to (new) appendices A1–A7 or cut. Some of the cut material has been moved to other parts of the manuscript. (Old) section 2.14 introducing the variational principle, which was the source of the referee's confusion here, has been cut entirely.

**RR1.10**

**The manuscript lacks any physical interpretation of the least erosion time, as a variational principle.**

A variational principle is inherently a rather abstract concept (especially if expressed in terms only of "action"), but the phrase "least erosion time" is self-evident. We believe that (now) sections 3.10 and 3.11 on Huygens' and Fermat's principles suffice to provide its physical relevance.

**RR1.11**

**The significance of the rays is also unclear as the proposed equations could be solved with other numerical methods without difficulty**

The significance of the rays lies in the fact that they carry boundary condition information into the interior, and that their speed and direction determine when and how such information reaches different parts of the landscape. It may be that the submitted manuscript failed to make this point as clear as it could. In the revision, we make this point in the abstract and at several points in the body of the paper.

It is worth noting that there is a modest literature (cited papers by Luke, Weissel & Seidl, and Royden & Perron) each of whose main focus was solving erosion using ray characteristics for 1+1D SPIM erosion functions; our work goes much further, using a fully

2D erosion model, developing a body of Hamiltonian theory, tying in aspect of classical mechanics and relatively recent discoveries in (Finsler) geometry, establishing erosional anisotropy, discovering a critical angle, discovering constancy of vertical slowness $p_z$, and so on. Much of this relies on having Hamilton's ray tracing ODEs, regardless of whether they are used in numerical solutions.

**RR1.12**

**The manuscript points out that the slope exponent $\eta$ and the direction of the rays are correlated, but the physical interpretation of this finding and the parameter $\eta$ itself are missing.**

We chose not to pursue this topic in the current paper, but we do make the following comment in (now) section 3.17 "Ray angle" (old section 3.16): "This switch in ray orientation as a function of slope scaling exponent $\eta$, which is illustrated in Fig. 12, echoes the observations in 1+1D of Weissel and Seidl (1998) and Royden and Perron (2013) of a change in upstream propagation with their gradient scaling exponent $n$. As their work has shown, this switch has important consequences for how and when knickpoints form (Stark and Stark, 2022)."

**RR1.13**

**One of the findings of the manuscript is to show the anisotropy of landscape erosion, however, anisotropy is not defined in the manuscript**

This is not entirely correct, but it is a fair criticism. We defined anisotropy as the angular disparity between the ray velocity vector and a vector parallel with the surface, i.e., as $\alpha - \beta + 90°$ (see old section 5.2, old figure 10, old section 5.3, old figure 11, etc), but did not define a symbol to associate with this anisotropy. In the revision we have remedied this mistake in several ways, notably by defining anisotropy as $\psi := \alpha - \beta + 90°$ and by doing so early in the paper (see new figure 1 caption and new section 1.1). See also now figure 8, new figure 9, now section 3.18 "Erosional anisotropy", etc; in particular, see new equation 64.

**RR1.14**

**The introduction suggests that no previous examples of modeling erosion as the evolution of implicit surfaces exist, but there are some papers on the evolution of implicit surfaces under erosion equations (Kraft et.al. (2011), Bencheikh et.al. (2020), The main result stating that the geometry of a surface determines its erosion is also a concept that constitutes the base of many works in the literature (Bloore (1977), Wilson (2009), Sipos et al. (2011), Domokos et al. (2014a), Domokos et al. (2014b)).**

Rereading the original introduction (including old section 1.1), we cannot see why the referee would get this impression. We do not say that erosion has never been modeled using implicit surfaces. In fact, in the (old) section 2.8 "Use of the eikonal equation in geomorphology" we say: *"To our knowledge, only one previous study has attempted to model landscape evolution as an implicit surface moving according to an eikonal equation. Aronsson and Lindé (1982) did so in a treatment of weathering-limited denudation of a rock cliff incised at its base by a river."* Then in (old) section 2.9 we say *"There is a literature on erosion driven by non-geomorphic processes, and much of it is unfamiliar to the geomorphology community. The methods employed in some of these papers provide a partial foundation for our Hamiltonian-based approach. For example, both implicit surface motion and the HJE have been the basis for modeling erosion at microscopic scales in an engineering context."* These comments have been moved to (new) appendices A3 and A4. In addition, in revising (now) discussion section 7.2 "A geomorphic surface Hamiltonian in 3D", we have added more references to seminal work on level sets (papers by Adalsteinsson and Sethian), some of which targeted the topic of surface erosion in the context of micro-scale etching, along with more recent work on this application by Radjenović and others.

Nevertheless, it is true to say that there is almost no literature on treating *landscape* erosion using implicit surfaces.

We are aware of the excellent recent work of Domokos and others, and of the older paper by Bloore (and also by Firey), but as we emphasized earlier, the scale at which our theory applies is radically different: we use a SPIM-type erosion model that parameterizes out any behaviour at or below the scale of the channel width. Shape change of a cobble, boulder, or obstruction (Wilson thesis) submerged in an abrasive flow is a radically different problem; papers on this topic typically emphasize curvature-dependent erosion, which cannot (using the methods we invoke) be converted into Hamiltonian form. Of course, micro-scale etching is also a radically different erosion problem, but we cite those papers because they employ a gradient-dependent erosion model rather like we do, and they demonstrate how to connect such an erosion model to a Hamiltonian via an HJE.

**RR1.15**

**Although the paper presents a simple, two dimensional equation, it misses to compare the computational outcome to available experimental results. For example, the simple flume experiments on cuboid marble blocks in Wilson (2009) might be reproduced.**

Our previous point (RR1.14) makes it clear that such experiments are not germane.

**RR1.16**

**Section 2 mixes topics that are tightly and loosely connected to the main ideas of the paper... There are many explanations and statements that are not used and it would be enough to reference them or move them to the appendix.**

We agree with all the comments in this paragraph, and we have acted on the referee's suggestions. (Old) section 2 has been halved, and the non-essential (but contextually important) material has all been moved into appendices. The prefatory paragraph in section 2 (was "Background") has been cut. Minor changes have been made to several of the remaining subsections 2.x. Old section 2.7 "Landscape as an erosion arrival-time surface" is now section 2.5.

**RR1.17**

**Some textbook concepts (such as covectors in Section 3.1.) are superfluous and should be omitted entirely.**

Strongly disagree!

It is absolutely essential for readers to understand the concept of a normal-slowness covector if they are to tackle our reworking of an erosion equation into a Hamiltonian. Since covectors are an entirely alien concept to almost all geomorphologists, the topic merits a tutorial review here. To suggest that we should refer readers to textbooks on differential calculus is to misunderstand the audience.

There exists no textbook that explains covectors in the context of erosion-surface motion, and none that shows why normal-slowness (erosion pace) covector components are a better way to express erosion rate than normal-velocity vector components. We have a rare opportunity here to introduce geomorphologists to a useful mathematical tool in a way that fits with their experience.

If this does not convince, take a look at "Introduction to Modern Dynamics" (Nolte, 2019), which is the most accessible of the textbooks we cite: his section 9.1.3 provides an excellent introduction to covectors in the context of manifolds and metric tensors, in a chapter on metric spaces and geodesic motion. Here's a snippet: *"In the language of vectors as rank-one tensors, the entity that operates*

*on a vector to yield a real number is known as a dual-vector. These* 320
*dual-vectors also have other names, such as covariant vectors or cov-*
*ectors, and also as one-forms. The name covector comes from the* 322
*fact that components of covectors transform like the basis vectors—*
*they co-transform."* In other words, prerequisites to understanding 324
this textbook explanation of covectors include proficiency with ten-
sors, vector bases, and basis transforms, and the didactic context is 326
entirely mathematical and abstract. Our summary section endeav-
ours to provide a more relatable context for our target audience of 328
geomorphologists and is tailored carefully to help understand how to
derive a geomorphic Hamiltonian. Readers who already understand 330
covectors can simply skip the section.

332

**RR1.18**

**Valid parameter ranges are not discussed in detail and regarded only with the jargon "on a shell"[sic: should be "on-shell"]**

Fair point, and we have addressed it in the following way. In (new)
section 4 "Implementation", (new) subsection 4.2 "Non-dimension- 334
alization", we set appropriate time, length and erosion rate scales,
non-dimensionalize the geomorphic Hamiltonian and the related Ha- 336
milton's equations, and derive a dimensionless number $\mathsf{Ci}$. This di-
mensionless number, which can be thought of as a dimensionless 338
boundary-condition horizontal erosion rate or surface angle, deter-
mines the shape of erosion rays and time-invariant profiles (along 340
with choices of gradient-scaling exponent $\eta$ and area-scaling expo-
nent $\mu$). 342

For the bulk of the figures, we choose a consistent value of $\mathsf{Ci} = 4°$
(along with choices of $\eta = \frac{1}{2}$ or $\eta = \frac{3}{2}$, and with $\mu/\eta = \frac{1}{2}$). New 344
figure 11 is specifically introduced to demonstrate parameter varia-
tion, and it shows how a time-invariant profile varies with choices 346
of $\mathsf{Ci} \in \{0.1°, 1°, 4°\}$. Note that our choices of $\eta$ and $\mu$ generate
profiles whose "slope-area" scaling is consistent with observations 348
(DEM analyses)—see Royden & Perron for discussion of this issue.

In terms of real-world scales, we have added a new discussion sub- 350
section 6.1 "Scales", along with two new tables 1 & 2, which connect
the choices of parameters $\mathsf{Ci}$ and $\eta$ with choices of domain scale $L_\mathrm{c}$ 352
and vertical erosion rate $\xi_\downarrow$ to yield predictions of real-world transit
time and height scales. 354

Regarding the jibe about the use of jargon: we have removed all
use of the phrase "on-shell" since the term is not well-known to the 356
target audience. Although a useful shorthand, it is not necessary.

**Response to Referee #2 (David Jon Furbish)**

We would like to thank David Furbish (DJF) for his review, which we found constructively critical and a pleasure to read. We agree with all his suggested changes and have acted accordingly in revising the manuscript. Addressing his comments in order:

**RR2.1**

**The Abstract reasonably describes the key elements of the paper, whereas the information content of the Introduction (Section 1) is sparse. Following this, I suspect that some if not many will stop reading, with eyes glazed over, somewhere within Sections 2.4 through 2.14. The material in these sections comes fast, and although the authors attempt to make connections with descriptions of Earth surfaces/lines, my reading suggests that these sections risk confusing readers without offering a clear idea of why it is important for readers to grasp these relatively unfamiliar concepts and techniques...**

We find ourselves in total agreement with DJF here, and his comments echo some made by referee#1 (see e.g. RR1.9). In the hope that we can prevent readers' eyes from glazing over (too much), we have cut section 2 (now "Core principles") in half and moved into appendices all the subsections not wholly necessary for setting the scene for section 3 ("Theory"). Section 2 now has only five subsections and they all cover aspects of tracking motion of implicit surfaces.

**RR2.2**

**I suggest the following possibility... Offer [examples], with clear diagram(s), right up front in the Introduction. Show the elements of what the analysis is describing about the motion of a surface/line and what the motion implies. Describe what is happening with reference to qualitative descriptions of the level set(s), normal motion, asymmetry, etc. Such diagrams could be simplified versions of material contained in diagrams that appear later...**

This is a great idea, and it echoes comments by referee#1. So, we have added a new subsection 1.1 "Tracking points on an erosion surface" and a new explanatory figure 1 much as he describes. Refer to RR1.1 for more details.

**RR2.3**

**...In other words, explain at the outset what the specific merits of a mostly unfamiliar style of analysis are. I suspect that such a "preview" might well help readers with the subsequent primer that unfolds the technical material...please know that I will not be offended if the authors prefer to reject these recommendations**

The new section 1.1 is indeed a kind of "preview", as DJF suggests, and we are grateful for his guidance here.

**RR2.4**

**...They then use the familiar erosion model embodied in Eq. (24) to highlight the description and its implications. Meanwhile, the science is moving beyond this simplistic formulation of erosion, except perhaps as a rough indication of large scale landscape behavior. I therefore suggest that the value of this example mostly resides in its familiarity, whereas I would like to imagine that the analyses in the paper are aimed more generally at providing a different perspective on describing motions of eroding surfaces/lines.**

It is true that the field of geomorphology is moving on from the simplistic SPIM or "stream-power model" of erosion. It is also true that we chose this erosion model much as DJF says—for its familiarity and its broad acceptance as, at the very least, a reasonable approximation of how erosion rate scales with gradient and catchment area. Even as more sophisticated models become established, it is important to remember that simpler models such as SPIM may well remain valid—or at least useful—for limited scopes.

Another reason for choosing SPIM was that, as our work demonstrates, building a Hamiltonian version of even a simple erosion model is quite involved, and we wanted to avoid adding more complexity by trying to deal with a more sophisticated model.

We contend, although we don't attempt to prove it in this paper, that key outcomes such as the discoveries of significant anisotropy and critical angles (see now sections 3.17, 3.18), a variational principle of least erosion time, and constancy of the vertical component of erosion slowness, will all transcend the particular choice of erosion model, i.e., that these conclusions reflect aspects of geomorphic reality rather than of a mathematical construct that crudely simulates nature. The degree to which these conclusions are modified, or perhaps falsified, by more realistic modeling is obviously an important task down the road.

**RR2.5**

**Line 34: Is there a compelling reason to "unify" them if they are described with respect to the physics involved? Unified in what sense? There may be merit in pulling key items offered in the discussion. . . to clarify what is intended.**

We have rephrased this part of the introduction, simplifying it into a single paragraph while adding the new prefatory subsection 1.1. Our use of the term "unify" was misleading: the sentence here now reads: *"There are obviously many directions involved in driving the evolution of a landscape, so what can we say about the direction of motion of the erosion surface itself? Our goal here is to answer this question using some concepts and tools of differential geometry and classical mechanics."*.

**RR2.6**

**Lines 64 and 249, and in reference to Line 515: To "drive" evolution implies physics, whereas a "least time" argument is a geometrical outcome of the variational analysis, as elaborated in the vicinity of Line 950.**

True. We have cut the opening paragraph in section 2 (was "Background") containing line 64, and we have rephrased the text elsewhere so that we no longer carelessly state that a variational principle "drives" landscape evolution.

**RR2.7**

**Line 70:** $h(x, y, t)$

Fixed.

**RR2.8**

**Line 75: Such problems are numerical rather than mathematical, however? The overhang idea, of course, is a restriction.**

Here DJF is referring to the following sentence (now at lines 142-143): *"Problems arise when the surface gradient becomes very steep, for example at knickpoints or channel banks, and any development of overhangs is obviously impossible."* The issue is not that an explicit 2+1D (sliced into 1+1D) treatment is inherently unstable numerically for steep-to-vertical cliffs, it's rather that such a treatment creates singularities at steep-to-vertical cliffs (singular in the gradient): this is a mathematical failure with consequences that go beyond the numerical. In any case, our rationale for moving to an implicit 3D (sliced into 2D) approach is not concerned with what happens on steep slopes: we use implicit surfaces so that we can meaningfully track points on successive surfaces (see new section 1.1). The explicit 2+1D approach does not have a vertical *coordinate*, per se, it has an elevation *function*, and so point velocities (ray directions) have no vertical component either. An implicit surface does have a vertical coordinate, so we can meaningfully compute how points move in the vertical direction.

**RR2.9**

**Lines 108–109: Changes in attributes(?) or effects(?) of vegetation and precipitation are easy to incorporate?**

This paragraph at the end of section 2.3, now at lines 174-181, has been rewritten, and we hope the revised version fixes the issue raised here. The pertinent sentence now reads: *"This [level-set] function can readily treat topographic gradient and curvature, and substrate erodibility; suitably provided with coupled process equations, it could also incorporate water flow depth and velocity, intermittent sediment cover, development of a vegetation layer, spatiotemporal precipitation, tectonic displacement, and so on."* In other words, the level-set method is sufficiently general to allow all such complexities to be incorporated if one so desired.

**RR2.10**

**Lines 850–855: As the authors describe here, in the Abstract, and elsewhere, the idea of "geometric self-constraint" suggests that the analysis is in fact restricted to those situations in which the formulation of erosion specifically satisfies this constraint, setting aside the added complexities of what the authors are calling "non-locality" with changes in contributing area/length. Given that this is the core premise of the work (Line 906), it probably merits description in the introductory example(s) described above (if the authors decide to offer such a "preview" at the outset).**

We don't explicitly mention the idea of a "geometric self-constraint" in the new preview section 1.1, because it is not needed to explain how a gradient-dependent erosion model converts into a Hamiltonian. However, we have revised sections 3.3–3.5 to make this idea more transparent.

To be clear, however, our statement in the discussion section (was line 906, now line 919) is this: *"our core premise [is] that what matters is the geometric self-constraint imposed by geomorphic processes, not the details of those processes."* We do not turn this premise into a mathematical constraint, so our theoretical development does not rely on it. Instead, it is a fair summary of why we believe the properties revealed by our geomorphic Hamiltonian will be valid even as the details of the erosion model changes.

**RR2.11**

**Regarding the final sentences: "…this is the variational principle that governs geomorphic surface erosion. It appears that energy dissipation need not be invoked, and that instead all that matters is geometry" the word 'governs' is strong**

Agreed. The rewritten sentence (now line 1011) now uses the term "guides". 470

472

**RR2.12**

**Now I'm going to gently throw a wrench into the works…
Consider…the problem of a freely meandering river over long time scales …the local rate of channel (centerline) migration is taken to be normal to the centerline … [and] is expressed as a convolution of upstream (and sometimes downstream) channel geometrical states …However…the arc length of the deforming coordinate system continuously changes …how might one approach this problem using the variational methods/techniques? Is this even possible? Might it reveal information that we otherwise would not discern from conventional analyses (including numerical simulations of meandering)?**

This is a very thought-provoking question. The first author (CPS) is deeply familiar with the theory of river meander- 474 ing, and with the work of DJF on this topic, so we can see the appeal of wanting to experiment with a variational 476 version.

However, we need to make clear that the particular frame- 478 work we have employed, which is to cast the equation of motion (erosion) into a metric function and thence into 480 a Hamiltonian, only works because the erosion function is equivalent to a first-order Hamilton-Jacobi equation (HJE). 482 HJEs in their classic form are (1) local and (2) functions of surface gradient but not curvature. There is some literature 484 on non-local HJEs and eikonal equations (in the context of dislocation dynamics in crystals; see papers by Alvarez et 486 al, Barles & Ley, Da Lio et al), but it is quite niche and theoretical. We were able to handle the non-locality of ero- 488 sion (in the sense that it is driven by flow that accumulates over an area upstream) relatively easily in a 2D vertical slice 490 with a fixed drainage divide, but this trick won't transfer to the meander problem. We were also able to disregard 492 surface curvature, because erosion driven by topographic curvature is not thought to be a relevant to erosion at the 494 scales to which our work applies. Meandering models, how- ever, rely heavily on tracking curvature up and downstream 496 to compute local perturbations in bend migration rate.

So, at first sight a Hamiltonian for channel meandering 498 would have to be both non-local and second order. If so, we have no road map for how to proceed. But we could easily 500 be mistaken, and there may well be a more easy route to treating meandering in a variational fashion. More thought 502 is needed.

**Revisions**

Below we run section-by-section through the changes we have made to the original manuscript.

**Title: The Direction of Landscape Erosion**

Referee#1 expressed some understandable confusion about the scope of our theory, given that the original title was very generally phrased "The Direction of Erosion". To alleviate the risk of readers having the same confusion, we have chosen to qualify this phrase to emphasize that we are dealing with erosion-driven evolution at the landscape scale, not at the boulder scale or to the in-situ erosion of objects in a flow.

**Abstract**

Several sentences have been reworded for clarity. The main changes are at the end where we have boosted communication of our results, with comments about erosional anisotropy, ray propagation of boundary-condition information, critical surface tilt, and non-dimensionalization. The sentences discussing the variational principle have been cut to prevent the abstract growing too long.

**1 Introduction**

The opener has been cut to a single paragraph and adjusted per the advice of DJF (referee #2). The cut material is now better covered in 1.1.

**1.1 Tracking points on an erosion surface**

Following the advice of DJF, we have inserted a new subsection here that previews our geomorphic Hamiltonian theory. It is relatively non-technical and introduces some of the mathematical ideas (notably how to relate successive erosion surface and points on them) from first principles. An companion explanatory figure 1.1 is introduced here too.

**1.2 Structure of the paper**

This subsection "summary of contents" has been extensively revised to reflect the changes made in the    526
manuscript.

**2 Core principles    528**

The opening paragraph has been cut for brevity.

**2.1 Landscape as an implicit surface – 2.4 Motion described by the    530**
Hamilton-Jacobi equation

Modest changes here, notably to 2.3 (The level-set equation) to clarify (per DJF's question) what we    532
meant about the flexibility of the level-set approach.

Note that throughout the revised text we now use calligraphic letters to denote the Hamiltonian,    534
fundamental function, and Lagrangian $(\mathcal{H}, \mathcal{F}, \mathcal{L})$ so as to avoid confusion with the use of $L_\mathrm{c}$ for the
domain length scale (replacing $x_1$).    536

**(was 2.5) Geoscience applications of the HJE – (was 2.6) Applications of**
the HJE to geomorphology    538

Moved to appendices A1-A2, per the advice of both referees.

**2.5 Landscape as an erosion arrival-time surface    540**

Moved up from 2.7 and modified slightly for mathematical clarity and brevity.

**(was 2.6) Use of the eikonal equation in geomorphology – (was 2.13) Ray    542**
tracing the motion of a front

Moved to appendices A3–A7, per the advice of both referees.    544

**(was 2.14) Variational principle governing erosion patterns**

Cut entirely to address the concerns of referee#1.    546

**3 Theory**

The opener has been modified to fit with changes to this section and to improve exposition. ⁵⁴⁸

**3.1 Tracking erosion with covectors**

Minor improvements to equations. ⁵⁵⁰

**3.2 Gradient is a covector**

Minor improvements to maths explanation. ⁵⁵²

**3.3 Modelling erosion in the surface-normal direction**

Renamed subsection. ⁵⁵⁴

**3.4 Erosion imposes a geometric self-constraint**

Trivial mods. ⁵⁵⁶

**3.5 Separable, gradient-dependent erosion rate model**

This renamed subsection is heavily reworded to meet the criticisms of referee#1. ⁵⁵⁸

**3.6 The erosion equation in Hamiltonian coordinates**

Trivial mods. ⁵⁶⁰

**3.7 The fundamental function**

Minor edits (e.g., use of $\mathcal{F}_*$ instead of $F^*$ for co-metric fundamental function). Reference to "on shell"/"off shell" concept removed to satisfy referee#1. ⁵⁶²

**3.8 The geomorphic surface Hamiltonian**

Minor edits.

**3.9 The geomorphic surface Lagrangian**

Minor edits.

**3.10 Erosional wavelets and Huygens' Principle**

Extensively reworded, along with mods to figure 4 (colour changes for consistency with other figs).

**3.11 Fermat's principle as a least action integral**

Extensively reworded, along with mods to figure 5 (colour changes for consistency with other figs).

**Figure 6. Ray tracing of erosion using Hamilton's equations**

Modified along with caption to satisfy referee#1's criticism that invocation of a fault-slip b.c. is not needed.

**3.12 Derivation of Hamilton's ray tracing equations**

Minor edits.

**3.14 Constancy of the vertical erosion rate along a ray**

Added paragraph discussing how this constancy is an example of Noether's theorem on continuous  symmetry.

**3.15 Conjugacy of point velocity and front slowness**

Minor edits.

**Figure 7. Variation of ray dip**

This figure is a revision of old fig. 12: the curves here are now generated from theory rather than numerical simulation, and they are annotated with the location of the extremal ray angle $\alpha_{\text{ext}}$ and corresponding critical surface tilt angle $\beta_{\text{c}}$—these discoveries that are new to this revision.

**3.16 Constancy of the Lagrangian**

Moved up from 3.17. Minor edits.

**Figure 8. Erosional anisotropy**

This figure is a revision of old fig. 13: the curves here are now generated from theory rather than numerical simulation, and they are annotated with the location of the extremal ray angle $\alpha_{\text{ext}}$

**3.17 Ray angle**

Moved down from 3.16. Several changes and additions, notably on our discovery of an extremal ray angle $\alpha_{\text{ext}}$ and corresponding critical surface tilt angle $\beta_{\text{c}}$, and the consequences of these threshold angles for properties of the Hamiltonian etc.

592
594

**Figure 9. Ray anisotropy**

Entirely new figure introduced to address several concerns raised by referee#1. This fig provides graphic explanation of how ray-surface anisotropy varies with gradient-scaling exponent $\eta$.

**(was 3.18): HJE and Hamilton action**

Moved to appendix F, per DJF's suggestion to reduce the burden on the reader.

**3.18 Erosional anisotropy**

New subsection to address criticism of referee#1 that anisotropy $\psi$ is not defined and its importance not sufficiently well explained.

**3.19 Measuring slope along the erosion front**

Trivial mod.

**Figure10. Estimation of the surface-normal angle**

As with all other remaining figures in the manuscript, this figure is now annotated with the value of Ci (dimensionless horizontal erosion rate imposed at boundary) used to generate the curves in it.

**3.20 Formulation of erosion model**

Material here has been moved to section 4.

**4 Implementation**

New section split off from the theory section 3 to respond to questions/misunderstandings of the referees regarding the specific formulation of an SPIM-like erosion model. The theory section demonstrates how a geomorphic Hamiltonian can be constructed from a fairly general (separable, scaling) erosion model, and this new section shows that settling on a SPIM form allows for further analysis, specific solution, and non-dimensionalization.

**4.1 A modified stream-power incision model**

Adapted from old subsection 3.20.

**4.2 Non-dimensionalization**

Entirely new subsection added to address concerns raised by referee#1 about parameter choices etc.

**Figure 11. Time-invariant profiles**

Entirely new figure added to help readers understand how model river profile shape is affected by choices of Ci.

**4.3 Direct integration**

Completely rewritten to take advantage of having a non-dimensional form of the model and its generality. 624

**5 Ray tracing solutions**

Renamed and renumbered section 4 Solution. The opener is somewhat modified; the section and allied 626 figures a whole have been heavily revised to incorporate the use of a non-dimensional form of Hamilton's equations. 628

**Figure 12. Tracing of a reference ray**

Revised version of old fig.6. Middle subfig (for $\eta = 1$) has been cut for simplicity and consistency 630 with other figs. Ray trajectories are slightly different now because of a change in the choice of model parameters, notably a standardization on the choice of $\mathsf{Ci} = 4°$. 632

**5.1 Model domain and boundary conditions**

Rewritten, simplified version of old 4.1 Model domain merged with old 4.2 Boundary and initial condi- 634 tions. Now invokes a constant vertical erosion rate at the boundary and discusses the relationship to a tectonically driven domain in a different way, per the suggestion of referee#1. Simplification made 636 possible by use of non-dimensionalized equations and by shift of some material into the next section.

**5.2 Ray equations 638**

New subsection split from old 4.1 Model domain. Some rewording.

**(was 4.3) Direct integration 640**

Moved to section 4 Implementation.

**Figure 13. Comparison of ray-traced solutions of time-invariant profiles 642**

Revised version of old fig.8. Profiles and ray trajectories are slightly different now because of a change in the choice of model parameters, notably a standardization on the choice of $\mathsf{Ci} = 4°$. Dimensionless 644 times used instead.

**5.3 Numerical integration method**

Minor changes to what was subsection 4.4.

**5.4 Reference ray construction**

Minor edits.

**5.5 Synthesis of a time-invariant profile**

Changes made to make more clear choices of parameters and use of dimensionless form of Hamilton's equations.

**Figure 14. Ray tracing construction of erosion surfaces**

Modified to match use of non-dimensionalized equations.

**6 Results**

Opening paragraph heavily revised to take account of the use of non-dimensionalized equations and other changes. Was section 6.

**6.1 Scales**

New subsection, with new tables 1 and 2, summarizing the relationships between model parameters $\eta$ and $\mathsf{Ci}$, real-world numbers such as domain size $L_c$ and vertical erosion rate at the boundary, and consequent time, rate and length scales (both dimensionless and dimensional).

**6.2 Time-invariant solutions**

Was subsection 5.1. Minor edits.

**6.3 Erosion rates**

Was subsection 5.2. Heavily rewritten to match changes discussed above, and because of changes to figures 15 and 16 (old figs 10 and 11). The figures now illustrate along-profile (non-dimensionalized)

behaviour (for $\text{Ci} = 4°$) of (a) anisotropy $\psi$, (b) ray velocity vector $\mathbf{v}$ components, (c) normal erosion rate $\xi^{\perp}$, (d) horizontal erosion rate $\xi^{\rightarrow}$, and (e) vertical erosion rate $\xi^{\rightarrow}$. 668

**Figure 17. Anisotropy of erosion**

Was fig. 14. Modified to use solutions for $\text{Ci} = 4°$. 670

**6.4 Anisotropy**

Was subsection 5.3. Reworded and shortened since some of this material is now covered elsewhere. 672

**7 Discussion**

Was section 6. 674

**7.1 Geometry controls (almost) everything**

Was 6.1. Some improvements to the wording here. Also to meet change in how we describe the boundary 676 condition as an erosion rate not a slip rate.

**7.2 A geomorphic surface Hamiltonian in 3D 678**

Was 6.2. Heavy revision towards the end of this section to describe better, and cite relevant literature more extensively, how we anticipate solving the geomorphic Hamiltonian using non-ray-tracing methods. 680

**7.3 The variational principle is not energy minimization**

Was 6.3. Trivial mods. 682

**8 Conclusions**

Was 7. Some rewording to improve clarity of exposition and to meet criticisms of referees. 684

**Appendices**

New appendices generated from material removed from main text. Otherwise largely unchanged. 686

**References**

Some additions, particularly regarding solution of HJEs using methods other than ray-tracing. Some 688 corrections.